# Revenue maximization via machine learning with noisy data

**Ellen Vitercik**
Department of Electrical Engineering and Computer Sciences
University of California, Berkeley
`vitercik@berkeley.edu`

**Tom Yan**
Department of Machine Learning
Carnegie Mellon University
`tyyan@cmu.edu`

## Abstract

Increasingly, copious amounts of consumer data are used to learn high-revenue mechanisms via machine learning. Existing research on mechanism design via machine learning assumes that there is a distribution over the buyers' values for the items for sale and that the learning algorithm's input is a *training set* sampled from this distribution. This setup makes the strong assumption that no noise is introduced during data collection. In order to help place mechanism design via machine learning on firm foundations, we investigate the extent to which this learning process is robust to noise. Optimizing revenue using noisy data is challenging because revenue functions are extremely volatile: an infinitesimal change in the buyers' values can cause a steep drop in revenue. Nonetheless, we provide guarantees when arbitrarily correlated noise is added to the training set; we only require that the noise has bounded magnitude or is sub-Gaussian. We conclude with an application of our guarantees to *multi-task* mechanism design, where there are multiple distributions over buyers' values and the goal is to learn a high-revenue mechanism per distribution. To our knowledge, we are the first to study mechanism design via machine learning with noisy data as well as multi-task mechanism design.

## 1 Introduction

Revenue maximization in multi-item settings is one of the most important, long-standing open problems in mechanism design. In Bayesian Mechanism Design, there is a set of items for sale and an underlying distribution defining a set of agents' values for the items. A mechanism determines which buyers receive which items and what they pay. For decades, research in economics has assumed that the mechanism designer must know the exact distribution over buyers' values. An explosion of research [e.g., 1, 4, 19, 20, 25, 26, 28, 30, 31, 33, 37, 40, 44–48, 51] has relaxed this strong assumption: instead, the distribution is unknown and the mechanism designer only has a training set of i.i.d. samples. Using the training set, the goal is to learn a mechanism with high expected revenue. *Learning-based* mechanism design is on the verge of taking over as the main tool for designing high-revenue mechanisms for selling items—a cornerstone of many modern enterprises.

Motivated by recent literature on the brittleness of deep learning models in the face of *imperceptible* noise [32, 41, 52], an important question is whether adversarial noise has the same effect in learning-based mechanism design. Learning methods deployed in real-world settings to design mechanisms

35th Conference on Neural Information Processing Systems (NeurIPS 2021).

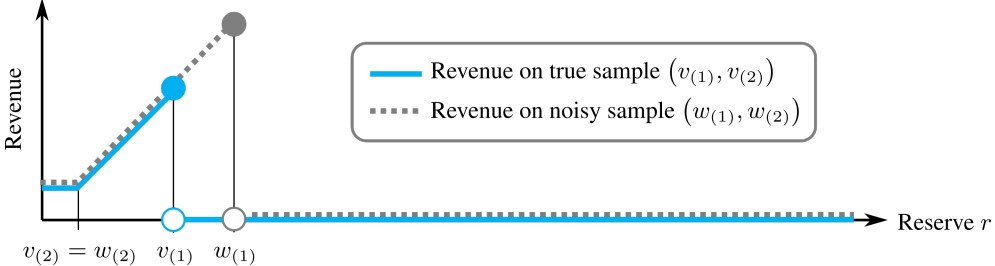

Figure 1: We illustrate revenue function volatility using the single-item second-price auction with a reserve $r$. Let $v_{(1)}$ be the highest bid and $v_{(2)}$ be the second-highest bid. Revenue as a function of the reserve $r$ equals $v_{(2)}$ if $r \leq v_{(2)}$, $r$ if $v_{(2)} < r \leq v_{(1)}$, and $0$ if $r > v_{(1)}$, as illustrated by the blue solid line. Suppose we receive a noisy sample with highest bid $w_{(1)} > v_{(1)}$ and second-highest bid $w_{(2)} = v_{(2)}$. Revenue is illustrated by the dotted grey line. Setting the reserve equal to $w_{(1)}$ maximizes revenue on the noisy sample, but leads to zero revenue on the true values $(v_{(1)}, v_{(2)})$.

must be robust to minute noise in the data. We provide guarantees for learning mechanisms with high expected revenue in the face of an adversary that can add arbitrarily correlated noise to the training set; we only require that the noise has bounded magnitude or is sub-Gaussian. In contrast, classic learning-theoretic results typically rely on the stronger assumption that the training samples are independent.

## 1.1   Summary of main contributions and overview of techniques

We set out to determine whether imperceptible adversarial noise in the training set can cause a catastrophic loss in the learned mechanism's expected revenue. Our main contribution is a *sensitivity analysis* of revenue with respect to the noise's magnitude, which answers this question in the negative. Our results apply to *empirical revenue maximization*, the canonical approach to learning-based mechanism design. To our knowledge, we are the first to study learning-based mechanism design under adversarial noise. We provide guarantees for optimizing revenue over several classic mechanism classes: second-price auctions with non-anonymous reserves under additive buyers, single-priced lottery mechanisms under unit-demand buyers, and item-pricing mechanisms under unit-demand buyers. These classes have been studied extensively [e.g., 6, 18, 24, 46, 51] and can be viewed as *hypothesis classes*, just as DNNs correspond to a particular (brittle) type of hypothesis class. In some settings, mechanisms from these classes have been shown to provide approximately optimal revenue [22, 23, 35, 39].

The key challenge we face is that revenue functions are extremely sensitive to small perturbations of the buyers' values. In a second-price auction, for example, slightly shifting the highest bid from below to above the reserve price can cause an arbitrarily large drop in revenue, as illustrated in Figure 1. The set of bidders whose bids are above their reserve prices can completely change when even an infinitesimal amount of noise is added, radically altering the analytical form of the revenue function. This is unlike most functions that we understand well from a learning-theoretic perspective, which generally are smooth, continuous, or—more broadly—exhibit a straightforward connection between parameters and output value. Despite the volatility of these revenue functions, we nonetheless are able to prove bounds on the revenue loss incurred by optimizing over a noisy training set.

Our second main contribution is an application of our guarantees to *multi-task mechanism design*. To our knowledge, we are the first to study this problem. The existing literature on mechanism design via machine learning assumes that there is a *single* distribution defining the buyers' values. Often, however, the mechanism designer may be interested in designing high-revenue mechanisms for multiple related distributions. Each distribution thus defines a distinct learning *task*. The goal is to leverage the similarity between the distributions to learn a high-revenue mechanism per distribution. Multi-task learning has proven useful in fields such as Computer Vision and Natural Language Processing [53], and we demonstrate its value in mechanism design as well. Our main technical insight is that we can transform training instances from one task into slightly noisy training instances for another task and make use of our near-optimality guarantees under noised samples.

## 1.2  Related research

**Adversarial Machine Learning:** A prominent line of research on machine learning in the presence of noise studies the case where some fraction of the training data may be noisy (data poisoning). Research in this vein includes settings where every instance can be corrupted with probability $\eta < 1$ [e.g., 3] and where an $\eta$-fraction of the training data can be corrupted [e.g., 16, 27]. Meanwhile, in our setting, every sample may be adversarially perturbed with probability 1, not just an $\eta$-fraction, though we require that the perturbation be bounded. Another line of research, especially prominent in Computer Vision, studies the case where the test data can be adversarially perturbed within, for example, an $\ell_p$-ball [32, 41, 52]. We, on the other hand, are concerned with noisy training data and face the unique challenges imposed by mechanism design.

**Mechanism Design with Noised Distributions:** While the robustness of mechanism design in the face of model uncertainty has been well studied (see for instance work by Bergemann and Morris [14] and Bergemann and Schlag [15]), to our knowledge, there has been relatively less work done on mechanism design via machine learning in the particular setting where samples are not necessarily drawn i.i.d. from the true distribution. The paper most closely related to ours is that by Cai and Daskalakis [20], who show that given access to a noisy distribution, it is possible to learn a mechanism with high expected revenue over the true distribution if the Kolmogorov distance between the noisy and true distributions are small. One might hope that the uniform distribution over the noisy training data could constitute the noisy distribution, but the Kolmogorov distance between this empirical distribution and the true distribution could be as large as 1, so these results do not apply to our setting. Moreover, we note that Cai and Daskalakis [20] makes the stronger assumption that the buyers' values follow a product distribution, whereas our assumed noise model allows for arbitrarily correlated distributions.

**Multi- vs Single-item Mechanism Design:** Further afield, Huang et al. [38] and Guo et al. [33] have also studied mechanism design under noisy settings, albeit under *single-item* settings. By contrast, we study a wider and different set of *multi-item* mechanism classes including multi-item lotteries and multi-item item-pricing mechanisms.

The past few decades of research on revenue-maximizing multi-item mechanism design has demonstrated that results and intuition from the single-item setting do not always carry over to the multi-item setting. And we see that this is the case in our paper as well. For example, an item-pricing mechanism for a single item would have a straight-forward stability analysis (similar to the analysis of a single-item anonymous second-price auction that we include for intuition in Appendix B.1). With multiple items and unit-demand buyers, however, stability does not even hold. Therefore, intuition from the single-item case does not carry over to the multi-item case, and we must use completely different analysis techniques.

**Dispersion:** The technique needed is that of *dispersion* (section 3.4). Our bounds improve based on how "nice" the distribution over buyers' values is, quantified by dispersion [5]. Dispersion has primarily been used to provide regret bounds [5, 8]. Balcan et al. [7] also use dispersion in mechanism design, though for a very different problem: estimating how much utility an agent can gain by misreporting their value in a manipulable mechanism.

The appendix summarizes additional related work on mechanism design with side information, multi-task learning, transfer learning, and approximate incentive compatibility.

## 2  Notation

There are $n$ buyers and $m$ items for sale. Each buyer $i \in [n]$ has a value for each item $j \in [m]$, denoted as $v_{ij} \geq 0$. We analyze unit-demand and additive buyers. If buyer $i$ is unit-demand, he is only interested in obtaining one item, so his value for a bundle $b \subseteq [m]$ of goods is equal to the maximum value he has for any item in $b$, $\max_{j \in b} v_{ij}$. If buyer $i$ is additive, his value for a bundle $b \subseteq [m]$ is $\sum_{j \in b} v_{ij}$. We use the notation $\boldsymbol{v}_i = (v_{i1}, \ldots, v_{im})$ to denote buyer $i$'s values for all $m$ items and $\boldsymbol{v} = (\boldsymbol{v}_1, \ldots, \boldsymbol{v}_n) \in \mathbb{R}^{nm}$ to denote all $n$ buyers' values. When there is only one item, we use the notation $\boldsymbol{v} = (v_1, \ldots, v_n) \in \mathbb{R}^n$, where $v_i \in \mathbb{R}$ is buyer $i$'s value for the item.

We study mechanism classes parameterized by vectors $\boldsymbol{r} \in \mathbb{R}^d$ for some $d$ (for example, the class of second-price auctions parameterized by non-anonymous reserves $\boldsymbol{r}$). The mechanisms we analyze

are dominant-strategy incentive compatible, so we assume the bids equal the buyers' true values. We denote the revenue of the mechanism defined by $r$ when the buyers' values equal $v$ by $\text{rev}_r(v)$. For simplicity of notation, we assume $\text{rev}_r(v)$ is in $[0, 1]$ though our results can be extended to the case where $\text{rev}_r(v) \in [0, H]$ for some $H$ (all bounds must simply be multiplied by $H$).

# 3 Learning under adversarial noise

In this section, we study mechanism design via machine learning when the data is corrupted. Let $\mathcal{S} = \left\{ v^{(1)}, \ldots, v^{(L)} \right\} \subset \mathbb{R}_{\geq 0}^{nm}$ be a set of valuation vectors drawn from an unknown distribution $\mathcal{D}$. Our learning algorithm receives a poisoned training set $\mathcal{S}' = \left\{ w^{(1)}, \ldots, w^{(L)} \right\} \subset \mathbb{R}_{\geq 0}^{nm}$.

**Bounded Noise Model:** To model the *imperceptible* noise that corrupts the bids, we assume the bounded noise model that is commonly assumed in adversarial defense literature. That is, for some known $\epsilon > 0$, all samples $\ell \in [L]$, all bidders $i \in [n]$, and all items $j \in [m]$, $w_{ij}^{(\ell)} \in \left[ v_{ij}^{(\ell)} - \epsilon, v_{ij}^{(\ell)} \right]$, or more succinctly, $v^{(\ell)} - \epsilon \leq w^{(\ell)} \leq v^{(\ell)}$. (If we only know that $\left\| v^{(\ell)} - w^{(\ell)} \right\|_\infty \leq \epsilon$, we can shift all bids in each vector $w^{(\ell)}$ down by $\epsilon$, in which case $v^{(\ell)} - 2\epsilon \leq w^{(\ell)} \leq v^{(\ell)}$.). Our goal is to use the noisy set $\mathcal{S}'$ to learn a mechanism with high expected revenue over $\mathcal{D}$.

We provide some more motivating factors for our choice of the noise model in the context of mechanism design. Besides its prevalence in adversarial learning literature, (1) Such noise may arise when one is using estimated bids to learn auction parameters. We give a concrete example in Section 4 where we study multi-task mechanism design. (2) The noise may result from bounded rationality on the part of the bidders, leading to small, seemingly-innocuous differences between the buyers' true values and reported values. (3) Lastly, we note that our results can immediately be extended to cover unbounded standard noise models such as *sub-Gaussian* noise. In this case, suppose that each element of each vector $v^{(i)} \in \mathcal{S}$ is perturbed by $\text{subG}(\sigma^2)$ to obtain the noisy vector $w^{(i)}$. Then with probability $1 - \delta$, for all $i \in [L]$, $\|w^{(i)} - v^{(i)}\|_\infty \leq \sigma \sqrt{2 \log \frac{2Lnm}{\delta}}$. The noise need not be independent. Therefore, all of our results hold with (high) probability $1 - 2\delta$ over the draw of the true values $\mathcal{S}$ and the noise for $\epsilon = 2\sigma \sqrt{2 \log \frac{2Lnm}{\delta}}$.

**Our Approach:** We begin by fixing a mechanism class parameterized by vectors $r \in \mathbb{R}^d$ for some $d$ (for example, the class of second-price auctions parameterized by non-anonymous reserves $r$). Given the training set $\mathcal{S}'$, the most widely used learning algorithm is *empirical revenue maximization (ERM)*, which returns the parameter setting $\hat{r}'$ that maximizes average empirical revenue over $\mathcal{S}'$. This algorithm will be the subject of our study. Throughout this section, we analyze the following key question: what is the difference between the expected revenue of the mechanism defined by $\hat{r}'$, $\mathbb{E}_{v \sim \mathcal{D}} [\text{rev}_{\hat{r}'}(v)]$, and that of the optimal mechanism in the class, $\max_{r \in \mathbb{R}^d} \mathbb{E} [\text{rev}_r(v)]$?

Another key aspect of learning besides optimality is sample complexity. In machine learning beyond the context of mechanism design, prior research [e.g., 49] has shown that a large increase in sample complexity is needed to handle noise with bounded $\ell_\infty$-norm. By contrast, under the same noise assumption, we show that empirical revenue maximization can achieve near-optimal revenue *without significantly higher sample complexity*. This exposes a notable contrast between the two learning tasks. We provide lower bounds showing that ERM's loss has an optimal dependence on the noise $\epsilon$.

## 3.1 Robustness

We begin by providing guarantees for any mechanism class that satisfies a notion of robustness, which helps us isolate exactly the properties we need to prove that robust revenue optimization is possible. A class is robust if it satisfies two properties. The first is a *stability* property. Let $\hat{r}$ be the *empirically optimal* parameter setting over the set $\mathcal{S}$ of true samples, which means that of all $r \in \mathbb{R}^d$, average revenue over $\mathcal{S}$ is maximized when $r = \hat{r}$. Let $\hat{r}'$ be the empirically optimal parameter vector over the noisy set $\mathcal{S}'$. Stability is satisfied when the average revenues of $\hat{r}$ and $\hat{r}'$ over $\mathcal{S}$ are close.

Our second property relies on a classic learning-theoretic notion of convergence. It is satisfied when, for every parameter vector $r$, average revenue over the set $\mathcal{S}$ is close to expected revenue. Although convergence bounds have been derived in prior research for the mechanism classes we analyze [6, 46], a convergence bound alone *does not imply any guarantees* whatsoever for optimization with noisy

| Mechanism class | Buyers' values | $p$-stable | Error bound |
|---|---|---|---|
| Single-item anonymous second-price auction | Additive | $p(\epsilon, n, m) = 2\epsilon$ (Theorem B.2) | $\tilde{O}(\epsilon + \sqrt{1/L})$ (Corollary B.6) |
| Multi-item non-anonymous second-price auction | Additive | $p(\epsilon, n, m) = 2m\epsilon$ (Theorem 3.3) | $\tilde{O}(m\epsilon + \sqrt{nm/L})$ (Corollary 3.7) |
| Multi-item non-anonymous lotteries | Unit-demand | $p(\epsilon, n, m) = n\epsilon$ (Theorem 3.9) | $\tilde{O}(n\epsilon + \sqrt{nm/L})$ (Corollary 3.10) |
| Item-pricing mechanisms | Unit-demand | Does not satisfy (Lemma B.11) | $\tilde{O}(nm^2(\kappa\epsilon + \sqrt{1/L}))^*$ (Theorem 3.13, Lemma 3.15) |

$^*$ The distribution over buyers' values is $\kappa$-bounded (Definition 3.14).
Table 1: Our $p$-stability guarantees together with the resulting error bounds. Item-pricing mechanisms do not satisfy $p$-stability, so we use alternative techniques to provide an error bound (Section 3.4).

data. This is why we introduce the separate notion of stability, which has not been previously studied in the automated mechanism design literature. The challenge then lies in proving that a variety of mechanism classes satisfy stability. To begin, we define the two properties formally as follows:

**Definition 3.1.** Given two functions $p : [0,1] \times \mathbb{Z}^2 \to \mathbb{R}$ and $q : [0,1] \times \mathbb{Z}^3 \to \mathbb{R}$, we say that a mechanism class is $p$-*stable and $q$-convergent* if the following conditions hold:

1. $p$-*stable.* For any $L \geq 1$, let $\mathcal{S} = \left\{ \boldsymbol{v}^{(1)}, \ldots, \boldsymbol{v}^{(L)} \right\} \subset \mathbb{R}^{nm}_{\geq 0}$ and $\mathcal{S}' = \left\{ \boldsymbol{w}^{(1)}, \ldots, \boldsymbol{w}^{(L)} \right\} \subset \mathbb{R}^{nm}_{\geq 0}$ be two arbitrary sets of vectors such that $\boldsymbol{v}^{(\ell)} - \boldsymbol{\epsilon} \leq \boldsymbol{w}^{(\ell)} \leq \boldsymbol{v}^{(\ell)}$ for all $\ell \in [L]$. Let $\hat{\boldsymbol{r}}$ and $\hat{\boldsymbol{r}}'$ be the empirically optimal parameter vectors over $\mathcal{S}$ and $\mathcal{S}'$: $\hat{\boldsymbol{r}} = \operatorname{argmax}_{\boldsymbol{r} \in \mathbb{R}^d} \sum_{\ell=1}^L \operatorname{rev}_{\boldsymbol{r}} \left( \boldsymbol{v}^{(\ell)} \right)$ and $\hat{\boldsymbol{r}}' = \operatorname{argmax}_{\boldsymbol{r} \in \mathbb{R}^d} \sum_{\ell=1}^L \operatorname{rev}_{\boldsymbol{r}} \left( \boldsymbol{w}^{(\ell)} \right)$. We require that on average over $\mathcal{S}$, the revenues of $\hat{\boldsymbol{r}}$ and $\hat{\boldsymbol{r}}'$ are close: $\frac{1}{L} \sum_{\ell=1}^L \operatorname{rev}_{\hat{\boldsymbol{r}}} \left( \boldsymbol{v}^{(\ell)} \right) - \operatorname{rev}_{\hat{\boldsymbol{r}}'} \left( \boldsymbol{v}^{(\ell)} \right) \leq p(\epsilon, n, m)$.

2. $q$-*convergent.* For any $L \geq 1$ and $\delta \in (0, 1)$, with probability $1 - \delta$ over the draw $\mathcal{S} \sim \mathcal{D}^L$, for every vector $\boldsymbol{r} \in \mathbb{R}^d$, the difference between the average revenue over $\mathcal{S}$ and the expected revenue is at most $q(\delta, L, n, m)$. In other words, $\left| \frac{1}{L} \sum_{\boldsymbol{v} \in \mathcal{S}} \operatorname{rev}_{\boldsymbol{r}}(\boldsymbol{v}) - \mathbb{E}\left[\operatorname{rev}_{\boldsymbol{r}}(\boldsymbol{v})\right] \right| \leq q(\delta, L, n, m)$.

If a mechanism class is $p$-stable and $q$-convergent, then the expected revenue of the empirically optimal mechanism over the noisy samples $\mathcal{S}'$ is close to the expected revenue of the optimal mechanism in the class. For completeness, the proof is in Appendix B.

**Fact 3.2.** *Fix a $p$-stable and $q$-convergent mechanism class. Let $\mathcal{S} = \left\{ \boldsymbol{v}^{(1)}, \ldots, \boldsymbol{v}^{(L)} \right\}$ and $\mathcal{S}' = \left\{ \boldsymbol{w}^{(1)}, \ldots, \boldsymbol{w}^{(L)} \right\}$ be two sets of valuation vectors such that for all $\ell \in [L]$, $\boldsymbol{v}^{(\ell)} - \boldsymbol{\epsilon} \leq \boldsymbol{w}^{(\ell)} \leq \boldsymbol{v}^{(\ell)}$. Let $\hat{\boldsymbol{r}}'$ be empirically optimal over $\mathcal{S}'$: $\hat{\boldsymbol{r}}' = \operatorname{argmax}_{\boldsymbol{r} \in \mathbb{R}^d} \sum_{\ell=1}^L \operatorname{rev}_{\boldsymbol{r}} \left( \boldsymbol{w}^{(\ell)} \right)$. With probability $1 - \delta$ over $\mathcal{S} \sim \mathcal{D}^L$, $\max_{\boldsymbol{r} \in \mathbb{R}^d} \mathbb{E}_{\boldsymbol{v} \sim \mathcal{D}} \left[ \operatorname{rev}_{\boldsymbol{r}}(\boldsymbol{v}) \right] - \mathbb{E}_{\boldsymbol{v} \sim \mathcal{D}} \left[ \operatorname{rev}_{\hat{\boldsymbol{r}}'}(\boldsymbol{v}) \right] \leq p(\epsilon, n, m) + 2q(\delta, L, n, m)$.*

We now prove that several mechanism classes are stable: second-price auctions with non-anonymous reserves and lotteries. (For intuition, we also analyze the simpler class of second-price auctions with anonymous reserves in Appendix B.1.) Table 1 summarizes our results.

### 3.2 Non-anonymous second-price auctions

We prove that *second-price auctions with non-anonymous reserves* and additive bidders are robust. In the single-item setting, this auction is defined by a vector $\boldsymbol{r} \in \mathbb{R}^n$, where $r_i$ is bidder $i$'s *reserve price*. Each bidder submits a bid and the mechanism discards all bidders whose bids are smaller than their reserves. If there are bidders remaining, the highest bidder, say bidder $i$, wins and pays the maximum of the second-highest remaining bid and $r_i$ (or $r_i$ if there are no other remaining bids). In the multi-item case, there is only one copy of each item and there is a separate auction per item. The mechanism is defined by $\boldsymbol{r} = (\boldsymbol{r}_1, \ldots, \boldsymbol{r}_m) \in \mathbb{R}^{nm}$, where $\boldsymbol{r}_j \in \mathbb{R}^n$ is the reserve vector for item $j$.

We begin by analyzing single-item auctions, which then implies guarantees for multiple items. The key challenge is that the set of bidders whose bids are above their reserves can completely change when even infinitesimal noise is added, radically altering the analytical form of the revenue function.

**Theorem 3.3.** *Single-item non-anonymous second-price auctions are $p$-stable with $p(\epsilon, n, m) = 2\epsilon$.*

*Proof.* For any $L \geq 1$, let $\mathcal{S} = \left\{ \boldsymbol{v}^{(1)}, \ldots, \boldsymbol{v}^{(L)} \right\} \subset \mathbb{R}^n_{\geq 0}$ and $\mathcal{S}' = \left\{ \boldsymbol{w}^{(1)}, \ldots, \boldsymbol{w}^{(L)} \right\} \subset \mathbb{R}^n_{\geq 0}$ be two sets of valuation vectors such that $\boldsymbol{v}^{(\ell)} - \boldsymbol{\epsilon} \leq \boldsymbol{w}^{(\ell)} \leq \boldsymbol{v}^{(\ell)}$ for all $\ell \in [L]$. Let $\hat{\boldsymbol{r}}'$ be the empirically optimal reserve vector over $\mathcal{S}'$ and let $\hat{\boldsymbol{r}}$ be empirically optimal over $\mathcal{S}$. This proof relies on two key lemmas. The first states that if we shift the reserve vector $\hat{\boldsymbol{r}}$—which is empirically optimal over $\mathcal{S}$—down by an additive factor of $\boldsymbol{\epsilon} = (\epsilon, \ldots, \epsilon) \in \mathbb{R}^n$ and apply it to the valuations in $\mathcal{S}'$, little revenue is lost. We use the standard notation $\langle x \rangle = \max\{x, 0\}$. The full proof is in Appendix B.2.

**Lemma 3.4.** *Let $\boldsymbol{r}_\epsilon = \left( \langle \hat{r}_1 - \epsilon \rangle, \ldots, \langle \hat{r}_n - \epsilon \rangle \right)$. If $\mathrm{rev}_{\hat{\boldsymbol{r}}} \left( \boldsymbol{v}^{(\ell)} \right) > 0$, then $\mathrm{rev}_{\boldsymbol{r}_\epsilon} \left( \boldsymbol{w}^{(\ell)} \right) \geq \mathrm{rev}_{\hat{\boldsymbol{r}}} \left( \boldsymbol{v}^{(\ell)} \right) - 2\epsilon$.*

*Proof sketch of Lemma 3.4.* Suppose that $\mathrm{rev}_{\hat{\boldsymbol{r}}} \left( \boldsymbol{v}^{(\ell)} \right) > 0$. Let $i$ be the winner under the values $\boldsymbol{v}^{(\ell)}$ and reserve $\hat{\boldsymbol{r}}$. Since $v_i^{(\ell)} \geq \hat{r}_i$, we know that $w_i^{(\ell)} \geq \langle v_i^{(\ell)} - \epsilon \rangle \geq \langle \hat{r}_i - \epsilon \rangle$, so under the values $\boldsymbol{w}^{(\ell)}$ and reserve $\boldsymbol{r}_\epsilon$, there is at least one bidder whose bid is at least his reserve. Let $i'$ be the winner under the valuation vector $\boldsymbol{w}^{(\ell)}$ and reserve $\boldsymbol{r}_\epsilon$. We split this proof into four cases that depend on:

1. Whether or not $i = i'$, and
2. Whether—under $\boldsymbol{v}^{(\ell)}$ and $\hat{\boldsymbol{r}}$—there is another bidder besides $i$ whose bid is at least his reserve (in other words, whether or not $\{t : v_t^{(\ell)} \geq \hat{r}_t, t \neq i\} = \emptyset$).

In the first case, $i = i'$ and $\{t : v_t^{(\ell)} \geq \hat{r}_t, t \neq i\} \neq \emptyset$. Let $k$ be the index of the second-highest bidder in $\boldsymbol{v}^{(\ell)}$ whose bid is above his reserve in $\hat{\boldsymbol{r}}$: $k = \mathrm{argmax}_{t \neq i}\{v_t^{(\ell)} : v_t^{(\ell)} \geq \hat{r}_t\}$. This means that $\mathrm{rev}_{\hat{\boldsymbol{r}}}(\boldsymbol{v}^{(\ell)}) = \max\{\hat{r}_i, v_k^{(\ell)}\}$. Since $v_k^{(\ell)} \geq \hat{r}_k$, we have that $w_k^{(\ell)} \geq \langle v_k^{(\ell)} - \epsilon \rangle \geq \langle \hat{r}_k - \epsilon \rangle$. Since $k \neq i$ and $i = i'$, it must be that $k \neq i'$, so $\{t : w_t^{(\ell)} \geq \langle \hat{r}_t - \epsilon \rangle, t \neq i'\} \neq \emptyset$ (in particular, the set contains $k$). Let $k'$ be the index of the second-highest bidder in $\boldsymbol{w}^{(\ell)}$ whose bid is above his reserve in $\boldsymbol{r}_\epsilon$: $k' = \mathrm{argmax}_{t \neq i'}\{w_t^{(\ell)} : w_t^{(\ell)} \geq \langle \hat{r}_t - \epsilon \rangle\}$, which means that $\mathrm{rev}_{\boldsymbol{r}_\epsilon}(\boldsymbol{w}^{(\ell)}) = \max\{\hat{r}_{i'} - \epsilon, w_{k'}^{(\ell)}\}$. Since $k \in \{t : w_t^{(\ell)} \geq \langle \hat{r}_t - \epsilon \rangle, t \neq i'\}$, we know that $w_{k'}^{(\ell)} \geq w_k^{(\ell)} \geq v_k^{(\ell)} - \epsilon$. Putting this all together, we prove that $\mathrm{rev}_{\boldsymbol{r}_\epsilon}(\boldsymbol{w}^{(\ell)}) \geq \max\{\hat{r}_i - \epsilon, w_{k'}^{(\ell)}\} \geq \max\{\hat{r}_i, v_k^{(\ell)}\} - 2\epsilon = \mathrm{rev}_{\hat{\boldsymbol{r}}}(\boldsymbol{v}^{(\ell)}) - 2\epsilon$. We prove the other three cases in the appendix. $\square$

We use Lemma 3.4 to continue Theorem 3.3's proof. Let $I$ be the set of indices $\ell$ such that $\mathrm{rev}_{\hat{\boldsymbol{r}}} \left( \boldsymbol{v}^{(\ell)} \right) > 0$. By Lemma 3.4, $\sum_{\ell=1}^L \mathrm{rev}_{\hat{\boldsymbol{r}}} \left( \boldsymbol{v}^{(\ell)} \right) = \sum_{\ell \in I} \mathrm{rev}_{\hat{\boldsymbol{r}}} \left( \boldsymbol{v}^{(\ell)} \right) \leq \sum_{\ell \in I} \mathrm{rev}_{\boldsymbol{r}_\epsilon} \left( \boldsymbol{w}^{(\ell)} \right) + 2L\epsilon \leq \sum_{\ell=1}^L \mathrm{rev}_{\boldsymbol{r}_\epsilon} \left( \boldsymbol{w}^{(\ell)} \right) + 2L\epsilon \leq \sum_{\ell=1}^L \mathrm{rev}_{\hat{\boldsymbol{r}}'} \left( \boldsymbol{w}^{(\ell)} \right) + 2L\epsilon$.

Next, we prove that for any reserve vector $\boldsymbol{r}$, revenue under the samples $\boldsymbol{v}^{(\ell)}$ will only be higher than revenue under the samples $\boldsymbol{w}^{(\ell)}$, which intuitively makes sense since $\boldsymbol{w}^{(\ell)} \leq \boldsymbol{v}^{(\ell)}$. The proof, which has a similar structure as the proof of Lemma 3.4, is in Appendix B.2.

**Lemma 3.5.** *For all samples $\ell \in [L]$ and reserve vectors $\boldsymbol{r} \in \mathbb{R}^n$, $\mathrm{rev}_{\boldsymbol{r}} \left( \boldsymbol{w}^{(\ell)} \right) \leq \mathrm{rev}_{\boldsymbol{r}} \left( \boldsymbol{v}^{(\ell)} \right)$.*

Since $\sum_{\ell=1}^L \mathrm{rev}_{\hat{\boldsymbol{r}}} \left( \boldsymbol{v}^{(\ell)} \right) \leq \sum \mathrm{rev}_{\hat{\boldsymbol{r}}'} \left( \boldsymbol{w}^{(\ell)} \right) + 2L\epsilon$, Lemma 3.5 implies that the theorem holds. $\square$

We next use Theorem 3.3 to prove stability guarantees in the case where there are multiple items and $n$ additive buyers. It follows from the fact that the mechanism's revenue decomposes additively over the items. The proof is in Appendix B.2.

**Theorem 3.6.** *The set of multi-item non-anonymous second-price auctions under $n$ additive buyers is $p$-stable with $p(\epsilon, n, m) = 2m\epsilon$.*

Theorem 3.6 and the fact that the class is $q$-convergent with $q(\delta, L, n, m) = O\left( \sqrt{\frac{1}{L} \left( nm \log(nm) + \log \frac{1}{\delta} \right)} \right)$ [46] implies that optimization under noise is possible:

**Corollary 3.7.** *Let $\mathcal{S} = \left\{ \boldsymbol{v}^{(1)}, \dots, \boldsymbol{v}^{(L)} \right\} \subset \mathbb{R}_{\geq 0}^{nm}$ and $\mathcal{S}' = \left\{ \boldsymbol{w}^{(1)}, \dots, \boldsymbol{w}^{(L)} \right\} \subset \mathbb{R}_{\geq 0}^{nm}$ be two sets such that for all $\ell \in [L]$, $\boldsymbol{v}^{(\ell)} - \boldsymbol{\epsilon} \leq \boldsymbol{w}^{(\ell)} \leq \boldsymbol{v}^{(\ell)}$. Let $\hat{\boldsymbol{r}}' = \arg\max_{\boldsymbol{r} \in \mathbb{R}^{nm}} \sum_{\ell=1}^{L} \mathrm{rev}_{\boldsymbol{r}} \left( \boldsymbol{w}^{(\ell)} \right)$. With high probability over $\mathcal{S} \sim \mathcal{D}^L$, $\max_{\boldsymbol{r} \in \mathbb{R}^{nm}} \mathbb{E}_{\boldsymbol{v} \sim \mathcal{D}} \left[ \mathrm{rev}_{\boldsymbol{r}}(\boldsymbol{v}) \right] - \mathbb{E}_{\boldsymbol{v} \sim \mathcal{D}} \left[ \mathrm{rev}_{\hat{\boldsymbol{r}}'}(\boldsymbol{v}) \right] = \tilde{O} \left( m\epsilon + \sqrt{\frac{nm}{L}} \right).$*

This dependence on $m\epsilon$ is tight: no algorithm has better error than $m\epsilon$. Therefore, empirical revenue maximization provides an optimal dependence on the error term $\epsilon$. The proof is in Appendix B.2.

**Proposition 3.8.** *Fix an arbitrary error term $\epsilon$. For any deterministic algorithm $\mathcal{A}$ that takes as input a training set $\mathcal{S} \subseteq \mathbb{R}^{nm}$ and returns a vector of non-anonymous reserves $\mathcal{A}(\mathcal{S}) \in \mathbb{R}^{nm}$, there exists a distribution $\mathcal{D}$ such that with probability 1 over the draw $\mathcal{S} = \left\{ \boldsymbol{v}^{(1)}, \dots, \boldsymbol{v}^{(L)} \right\} \sim \mathcal{D}^L$, $\max_{\boldsymbol{r} \in \mathbb{R}^{nm}} \mathbb{E} \left[ \mathrm{rev}_{\boldsymbol{r}}(\boldsymbol{v}) \right] - \mathbb{E} \left[ \mathrm{rev}_{\mathcal{A}(\mathcal{S}')}(\boldsymbol{v}) \right] = \Omega(m\epsilon)$ for some noisy training set $\mathcal{S}' = \left\{ \boldsymbol{w}^{(1)}, \dots, \boldsymbol{w}^{(L)} \right\}$ such that $\left\| \boldsymbol{v}^{(\ell)} - \boldsymbol{w}^{(\ell)} \right\|_\infty \leq \epsilon$ for all $\ell \in [L]$.*

### 3.3 Lotteries

We next prove that single-priced lottery mechanisms satisfy our stability and convergence conditions. We analyze a setting where there are $n$ unit-demand buyers with values for $m$ items. Unlike the previous section, we assume there are at least $n$ units of each good available. A single-priced lottery is defined by $n(m+1)$ parameters: for each buyer $i$, there is a price $r_{i0} \in \mathbb{R}_{\geq 0}$ and a set of probabilities $r_{i1}, \dots, r_{im} \in [0, 1]$ with $\sum_{j=1}^{m} r_{ij} = 1$. If the buyer chooses to pay $r_{i0}$, she will receive one item $J \in [m]$, and $\Pr[J = j] = r_{ij}$. Therefore, her expected utility is $\sum_{j=1}^{m} v_{ij} r_{ij} - r_{i0}$. She will choose to participate in the lottery so long as her expected utility is at least 0. We prove that this mechanism class satisfies our stability and convergence conditions. The full proof is in Appendix B.3.

**Theorem 3.9.** *The set of lotteries with $n$ unit-demand buyers is p-stable with $p(\epsilon, n, m) = n\epsilon$.*

*Proof sketch.* We sketch the proof that $p(\epsilon, n, m) = \epsilon$ in the single-buyer setting ($n = 1$) and for the sake of generality, we prove the guarantee for $n$ buyers in the appendix. For any $L \geq 1$, let $\mathcal{S} = \left\{ \boldsymbol{v}^{(1)}, \dots, \boldsymbol{v}^{(L)} \right\} \subset \mathbb{R}_{\geq 0}^{m}$ and $\mathcal{S}' = \left\{ \boldsymbol{w}^{(1)}, \dots, \boldsymbol{w}^{(L)} \right\} \subset \mathbb{R}_{\geq 0}^{m}$ be two arbitrary sets of valuation vectors such that $\boldsymbol{v}^{(\ell)} - \boldsymbol{\epsilon} \leq \boldsymbol{w}^{(\ell)} \leq \boldsymbol{v}^{(\ell)}$ for all $\ell \in [L]$. Let $\hat{\boldsymbol{r}} = (\hat{r}_0, \hat{r}_1, \dots, \hat{r}_m)$ be the empirically optimal parameter vector over the set $\mathcal{S}$ and let $\hat{\boldsymbol{r}}' = (\hat{r}'_0, \hat{r}'_1, \dots, \hat{r}'_m)$ be the empirically optimal parameter vector over the set $\mathcal{S}'$. We prove that $\frac{1}{L} \sum_{\ell=1}^{L} \mathrm{rev}_{\hat{\boldsymbol{r}}} \left( \boldsymbol{v}^{(\ell)} \right) - \mathrm{rev}_{\hat{\boldsymbol{r}}'} \left( \boldsymbol{v}^{(\ell)} \right) \leq \epsilon$.

We first prove that if we shift the price $\hat{r}_0$ down by $\epsilon$ and evaluate the resulting lottery over $\mathcal{S}'$, little revenue is lost. Specifically, letting $\boldsymbol{r}_\epsilon = (\max\{\hat{r}_0 - \epsilon, 0\}, \hat{r}_1, \dots, \hat{r}_n)$ we prove that if $\mathrm{rev}_{\hat{\boldsymbol{r}}} \left( \boldsymbol{v}^{(\ell)} \right) > 0$, then $\mathrm{rev}_{\boldsymbol{r}_\epsilon} \left( \boldsymbol{w}^{(\ell)} \right) \geq \mathrm{rev}_{\hat{\boldsymbol{r}}} \left( \boldsymbol{v}^{(\ell)} \right) - \epsilon$. This implies that $\sum_{\ell=1}^{L} \mathrm{rev}_{\hat{\boldsymbol{r}}} \left( \boldsymbol{v}^{(\ell)} \right) \leq \sum_{\ell=1}^{L} \mathrm{rev}_{\boldsymbol{r}_\epsilon} \left( \boldsymbol{w}^{(\ell)} \right) + L\epsilon$. Since $\hat{\boldsymbol{r}}'$ is empirically optimal under $\mathcal{S}'$, $\sum_{\ell=1}^{L} \mathrm{rev}_{\hat{\boldsymbol{r}}} \left( \boldsymbol{v}^{(\ell)} \right) \leq \sum_{\ell=1}^{L} \mathrm{rev}_{\hat{\boldsymbol{r}}'} \left( \boldsymbol{w}^{(\ell)} \right) + L\epsilon$.

Next, we prove that for any parameter vector $\boldsymbol{r}$, revenue under the samples $\boldsymbol{v}^{(\ell)}$ will only be higher than revenue under the samples $\boldsymbol{w}^{(\ell)}$. Specifically, for every $\ell \in [L]$ and any parameter vector $\boldsymbol{r} \in \mathbb{R}^{m+1}$, $\mathrm{rev}_{\boldsymbol{r}} \left( \boldsymbol{w}^{(\ell)} \right) \leq \mathrm{rev}_{\boldsymbol{r}} \left( \boldsymbol{v}^{(\ell)} \right)$. This implies that $\sum_{\ell=1}^{L} \mathrm{rev}_{\hat{\boldsymbol{r}}} \left( \boldsymbol{v}^{(\ell)} \right) \leq L\epsilon + \sum_{\ell=1}^{L} \mathrm{rev}_{\hat{\boldsymbol{r}}'} \left( \boldsymbol{v}^{(\ell)} \right)$. $\qquad \square$

By a natural generalization of prior research [6], $q(\delta, L, n, m) = O\left( \sqrt{\frac{1}{L} \left( nm \log(nm) + \log \frac{1}{\delta} \right)} \right)$. Fact 3.2 and Theorem 3.9 imply our main result for this section—that the class of lotteries is robust:

**Corollary 3.10.** *Let $\mathcal{S} = \left\{ \boldsymbol{v}^{(1)}, \dots, \boldsymbol{v}^{(L)} \right\} \subset \mathbb{R}_{\geq 0}^{nm}$ and $\mathcal{S}' = \left\{ \boldsymbol{w}^{(1)}, \dots, \boldsymbol{w}^{(L)} \right\} \subset \mathbb{R}_{\geq 0}^{nm}$ be two sets such that for all $\ell \in [L]$, $\boldsymbol{v}^{(\ell)} - \boldsymbol{\epsilon} \leq \boldsymbol{w}^{(\ell)} \leq \boldsymbol{v}^{(\ell)}$. Let $\hat{\boldsymbol{r}}' = \arg\max_{\boldsymbol{r} \in \mathbb{R}^{n(m+1)}} \sum_{\ell=1}^{L} \mathrm{rev}_{\boldsymbol{r}} \left( \boldsymbol{w}^{(\ell)} \right)$. With high probability over $\mathcal{S} \sim \mathcal{D}^L$, $\max_{\boldsymbol{r} \in \mathbb{R}^{n(m+1)}} \mathbb{E}_{\boldsymbol{v} \sim \mathcal{D}} \left[ \mathrm{rev}_{\boldsymbol{r}}(\boldsymbol{v}) \right] - \mathbb{E} \left[ \mathrm{rev}_{\hat{\boldsymbol{r}}'}(\boldsymbol{v}) \right] = \tilde{O} \left( n\epsilon + \sqrt{\frac{nm}{L}} \right).$*

This dependence on $n\epsilon$ is tight: no algorithm has better error than $n\epsilon$. The proof is in Appendix B.3.

**Proposition 3.11.** *Fix an arbitrary error term $\epsilon$. For any deterministic algorithm $\mathcal{A}$ that takes as input a training set $\mathcal{S} \subseteq \mathbb{R}^{nm}$ and returns a vector of lottery parameters $\mathcal{A}(\mathcal{S}) \in \mathbb{R}^{n(m+1)}$, there exists a distribution $\mathcal{D}$ such that with probability 1 over the draw $\mathcal{S} = \left\{ \boldsymbol{v}^{(1)}, \dots, \boldsymbol{v}^{(L)} \right\} \sim \mathcal{D}^L$, $\max_{\boldsymbol{r} \in \mathbb{R}^{n(m+1)}} \mathbb{E} \left[ \mathrm{rev}_{\boldsymbol{r}}(\boldsymbol{v}) \right] - \mathbb{E} \left[ \mathrm{rev}_{\mathcal{A}(\mathcal{S}')}(\boldsymbol{v}) \right] = \Omega(n\epsilon)$ for some noisy training set $\mathcal{S}' = \left\{ \boldsymbol{w}^{(1)}, \dots, \boldsymbol{w}^{(L)} \right\}$ such that $\left\| \boldsymbol{v}^{(\ell)} - \boldsymbol{w}^{(\ell)} \right\|_\infty \leq \epsilon$ for all $\ell \in [L]$.*

## 3.4 Guarantees via dispersion

We now provide guarantees for the class of item-pricing mechanisms under unit-demand buyers, under which revenue is a particularly volatile function. In fact, the $p$-stability property does not hold for any non-trivial stability function $p$. We show that we can use another tool—called *dispersion* [5]—to obtain guarantees for learning with a noisy dataset. Therefore, we illustrate that $p$-stability is a sufficient but not necessary condition for obtaining robustness guarantees for a wide range of mechanism classes. We showcase this for item-pricing mechanisms.

**Item-pricing mechanisms:** Let there be $n$ buyers and $m$ items and as in Section 3.2, we assume there is only one unit of each item for sale. There is a non-anonymous price $r_{ij} \in \mathbb{R}$ for each buyer $i \in [n]$ and each item $j \in [m]$. First, buyer 1 arrives and buys the item $j \in [m]$ that maximizes his utility $v_{1j} - r_{1j}$ (or chooses not to buy if $v_{1j} < r_{1j}$ for all items $j \in [m]$). Next, buyer 2 arrives and selects the item among the remaining that maximizes his utility $v_{2j} - r_{2j}$ (or chooses not to buy). This process continues for each buyer $i \in [n]$. As in the previous section, we use the notation $\text{rev}_{\boldsymbol{r}}(\boldsymbol{v})$ to denote the revenue of the mechanism with prices $\boldsymbol{r} \in \mathbb{R}^{nm}$ when the buyers have values $\boldsymbol{v} \in \mathbb{R}^{nm}$.

Under item-pricing mechanisms, we face an immediate obstacle which is that $p$-stability does not necessarily hold for any non-trivial choice of the function $p$: the empirically optimal prices over the noisy set $\mathcal{S}'$ might have terrible revenue on average over the true set $\mathcal{S}$ (Lemma B.11 in Appendix B.4).

The primary challenge in proving a stability guarantee is that an agent may have very different values for two items but very similar utilities given a set of prices $\boldsymbol{r}$. If we add a little noise to the buyer's values, their preference ordering over the items may flip. In turn, this may cause the buyer to select a low-cost item instead of a high-cost item, thus triggering a sharp drop in revenue. As a result, average revenue over a noisy training set may be completely different from average revenue over the uncorrupted training set. If, however, given any price vector $\boldsymbol{r}$, the buyers' utilities across items are not too similar (they are "dispersed"), their preferences will not change if a bit of noise is added to their values. Below, we formally capture this notion of dispersion.

**Definition 3.12** (($\epsilon, k$)-dispersion). Let $\mathcal{S} = \left\{\boldsymbol{v}^{(1)}, \ldots, \boldsymbol{v}^{(L)}\right\} \subset \mathbb{R}^{nm}_{\geq 0}$ be a set of valuation vectors. We say that $\mathcal{S}$ is ($\epsilon, k$)-dispersed if for any price vector $\boldsymbol{r} \in \mathbb{R}^m$, there are at most $k$ valuation vectors in $\mathcal{S}$ such that for some buyer $i \in [n]$, either:
1. For some item pair $j, j' \in [m]$, buyer $i$'s utility for item $j$ is within $\epsilon$ of her utility for item $j'$, or
2. Buyer $i$'s utility for some item $j$ is between 0 and $\epsilon$.

The parameter $k$ allows for some slack: for some—but not all—of the valuation vectors in $\mathcal{S}$, the buyers' utilities can concentrate. Later in this section, we demonstrate that dispersion holds under mild assumptions. First, we provide a guarantee based on dispersion for optimizing prices using a noisy training set. The full proof is in Appendix B.4.

**Theorem 3.13.** *Let $\mathcal{S} = \left\{\boldsymbol{v}^{(1)}, \ldots, \boldsymbol{v}^{(L)}\right\} \sim \mathcal{D}^L$ be a set of $(2\epsilon, k)$-dispersed vectors. Let $\mathcal{S}' = \left\{\boldsymbol{w}^{(1)}, \ldots, \boldsymbol{w}^{(L)}\right\} \subset \mathbb{R}^{nm}_{\geq 0}$ be another set such that for all $\ell \in [L]$, $\left\|\boldsymbol{v}^{(\ell)} - \boldsymbol{w}^{(\ell)}\right\|_{\infty} \leq \epsilon$. Let $\hat{\boldsymbol{r}}'$ be empirically optimal over $\mathcal{S}'$: $\hat{\boldsymbol{r}}' = \operatorname{argmax}_{\boldsymbol{r} \in \mathbb{R}^{nm}} \sum_{\ell=1}^{L} \text{rev}_{\boldsymbol{r}}\left(\boldsymbol{w}^{(\ell)}\right)$. With probability $1 - \delta$ over the draw of $\mathcal{S}$, $\max_{\boldsymbol{r} \in \mathbb{R}^{nm}} \mathbb{E}_{\boldsymbol{v} \sim \mathcal{D}}\left[\text{rev}_{\boldsymbol{r}}(\boldsymbol{v})\right] - \mathbb{E}\left[\text{rev}_{\hat{\boldsymbol{r}}'}(\boldsymbol{v})\right] = O\left(\frac{k}{L} + \sqrt{\frac{1}{L}\left(nm \log(nm) + \log\frac{1}{\delta}\right)}\right)$.*

*Proof sketch.* We first prove that for any price vector $\boldsymbol{r}$, average revenue over $\mathcal{S}$ is within $\frac{k}{L}$ of average revenue over $\mathcal{S}'$, for the following reason. Due to dispersion, for most of the valuation vectors $\boldsymbol{v}^{(\ell)} \in \mathcal{S}$, the buyers will choose to buy the same set of goods regardless of whether their values are defined by $\boldsymbol{v}^{(\ell)}$ or $\boldsymbol{w}^{(\ell)}$, so the revenue remains constant. For at most $k$ valuation vectors in $\mathcal{S}$, the allocation may change arbitrarily, but revenue is always bounded in the interval $[0, 1]$, which is why the bound contains the term $\frac{k}{L}$. Finally, to relate average and expected revenue, we use a generalization bound from prior research [46] which equals the second summand of our bound. $\square$

This result raises an important question: when will a set of valuations be dispersed? One example, also observed in prior research [5], is when the distribution over buyers' values is relatively "smooth" (*a la* smoothed analysis [50]), formalized as follows.

**Definition 3.14.** For any distribution over an abstract set $\mathcal{X}$ with probability density function $f : \mathcal{X} \to \mathbb{R}_{\geq 0}$, the density function is $\kappa$-*bounded* if $\max_{x \in \mathcal{X}} f(x) \leq \kappa$.

The proof of the following lemma is in Appendix B.4.

**Lemma 3.15.** *Suppose that for every buyer $i \in [n]$ and every pair of items $j, j' \in [m]$, buyer $i$'s values for items $j$ and $j'$ have a $\kappa$-bounded joint density function. Then for any $\epsilon > 0$, with probability $1 - \delta$ over the draw $\mathcal{S} = \{v^{(1)}, \ldots, v^{(L)}\} \sim \mathcal{D}^L$, the set $\mathcal{S}$ is $(2\epsilon, k)$-dispersed with $k = 4Lnm^2\kappa\epsilon + O\left(nm^2\sqrt{L\log\frac{nm}{\delta}}\right)$.*

Theorem 3.13 and Lemma 3.15 imply learning guarantees for optimizing with noisy data when dispersion holds, even for these extremely volatile revenue functions.

# 4 Multi-task learning

We now show how our results from the previous section apply to multi-task mechanism design.

**Setup:** In this setting, there are $T$ tasks, and each task $t \in [T]$ is defined by a distribution $\mathcal{D}^{(t)}$ over buyers' values. Each distribution could represent, for example, buyers from different regions or market segments. The learning algorithm receives a training set sampled from each task's distribution. Given a mechanism class parameterized by vectors $r \in \mathbb{R}^d$ (which equal, for example, reserve prices) and the $T$ training sets, our goal is to learn a parameter setting $r^{(t)}$ for each task $t$ with high expected revenue over $\mathcal{D}^{(t)}$. In particular, $\max_{r \in \mathbb{R}^d} \mathbb{E}_{v \sim \mathcal{D}^{(t)}}[\text{rev}_r(v)] - \mathbb{E}_{v \sim \mathcal{D}^{(t)}}[\text{rev}_{r^{(t)}}(v)]$ should be small.

**Task relatedness:** If the distributions are completely unrelated, there is no hope that sharing information across tasks could improve learning. Thus, we require some notion of "task-relatedness". Under our model of task-relatedness, buyers' values across tasks are similar up to additive shifts which represent, for example, differences in their income brackets. More formally, for each buyer $i \in [n]$ and item $j \in [m]$, there is an underlying distribution $\mathcal{D}_{ij}$ over $\mathbb{R}_{\geq 0}$. Let $\mathcal{D} = \times_{i,j}\mathcal{D}_{ij}$ denote the cross product of these $nm$ distributions. Each task $t \in [T]$ is defined by an unknown vector of *common-base values* $b^{(t)} \in \mathbb{R}^n$. Intuitively, when $b_i^{(t)}$ is large, buyer $i$ tends to be willing to pay more for items, and when $b_i^{(t)}$ is small, buyer $i$ tends to be frugal. This model is inspired by the well-studied *common-base-value* model [9, 24]. For each task $t \in [T]$, our learning algorithm receives $L$ samples $v^{(1,t)}, \ldots, v^{(L,t)} \in \mathbb{R}^{nm}$. Each sample $v^{(\ell,t)}$ is generated by sampling $z^{(\ell,t)} \sim \mathcal{D}$ and defining buyer $i$'s value for item $j$ to be $v_{ij}^{(\ell,t)} = b_i^{(t)} + z_{ij}^{(\ell,t)}$. This defines a distribution $\mathcal{D}_{ij}^{(t)}$ over buyer $i$'s value for item $j$. We use the notation $\mathcal{D}^{(t)} = \times_{i,j}\mathcal{D}_{ij}^{(t)}$ to denote the cross product of these $nm$ distributions for task $t$. The learning algorithm does not know the vectors $b^{(t)}$ or $z^{(\ell,t)}$; it only observes the valuation vectors $v^{(\ell,t)}$. Throughout this section, we use the (slight abuse of) notation $b^{(t)} + z^{(\ell,t)}$ to denote $v^{(\ell,t)}$. In Appendix D, we supply empirical evidence that demonstrates that this is a reasonable model of task-relatedness on real-world auction data.

**Sample bootstrapping:** Our multi-task approach follows from the observation that for each task $t \in [T]$, $\mathcal{S}_t = \{b^{(t)} + z^{(\ell,\tau)} : \tau \in [T], \ell \in [L]\}$ is a training set of $LT$ samples from the $t^{th}$ task's distribution $\mathcal{D}^{(t)}$ since each vector $z^{(\ell,\tau)}$ is sampled from $\mathcal{D}$. However, we cannot compile this training set since we do not know the base values $b^{(t)}$. Nonetheless, in Appendix C, we show that it is straightforward to compute a set of vectors $\mathcal{S}_t' = \{w^{(\ell,t,\tau)} : \ell \in [L], \tau \in [T]\}$ that closely approximate* the vectors in $\mathcal{S}_t$: with high probability, for all pairs of tasks $t, \tau \in [T]$ and indices $\ell \in [L]$, $\left\|b^{(t)} + z^{(\ell,\tau)} - w^{(\ell,t,\tau)}\right\|_\infty = \tilde{O}\left(\sqrt{\frac{1}{Lm}}\right)$. From our results in Section 3, we know that it is possible to learn a mechanism with high expected revenue over the distribution $\mathcal{D}^{(t)}$ using the noisy training set $\mathcal{S}_t'$. We summarize the resulting guarantees below. All proofs are in Appendix C.

**Multi-item non-anonymous second-price auctions.** Our results from Section 3.2 imply the following guarantee for learning a high-revenue multi-item second-price auction with non-anonymous reserves under additive bidders.

**Theorem 4.1.** *For each task $t \in [T]$, let $\hat{r}_t'$ be the empirically optimal reserve vector over the set $\mathcal{S}_t'$: $\hat{r}_t' = \text{argmax}_{r \in \mathbb{R}^{nm}} \sum_{\ell=1}^{L} \sum_{\tau=1}^{T} \text{rev}_r\left(w^{(\ell,t,\tau)}\right)$. With probability $1 - \delta$, for every task $t \in [T]$,*
$$\max_{r \in \mathbb{R}^{nm}} \mathbb{E}_{v \sim \mathcal{D}^{(t)}}\left[\text{rev}_r(v) - \text{rev}_{\hat{r}_t'}(v)\right] = \tilde{O}\left(\sqrt{\frac{1}{LT}\left(nm + \log\frac{T}{\delta}\right) + \frac{m}{L}\log\frac{nT}{\delta}}\right).$$

---

*In some cases, we show that it is also possible to exactly recover the vectors in $\mathcal{S}_t$ (Appendix C.2).

This theorem implies that for any $\gamma \in (0,1)$, when $L = \tilde{\Omega}\left(m/\gamma^2\right)$ and $T = \tilde{\Omega}(n)$, the revenue of the auction defined by the reserves $\hat{r}'_t$ is within $\gamma$ of optimal: $\max \mathbb{E}_{\boldsymbol{v}\sim\mathcal{D}^{(t)}}\left[\mathrm{rev}_{\boldsymbol{r}}(\boldsymbol{v})\right] - \mathbb{E}_{\boldsymbol{v}\sim\mathcal{D}^{(t)}}\left[\mathrm{rev}_{\hat{r}'_t}(\boldsymbol{v})\right] \leq \gamma$. In contrast, the best-known single-task sample complexity guarantee requires $L = \tilde{\Omega}\left(nm/\gamma^2\right)$ [46][†]. Our per-task sample complexity is smaller by a multiplicative factor equalling the number of buyers.

**Lotteries.** Our results from Section 3.3 imply that under unit-demand buyers, optimizing lottery parameters using the noisy training sets $\mathcal{S}'_t$ results in nearly optimal revenue.

**Theorem 4.2.** *For each task $t$, let $\hat{r}'_t$ be the empirically optimal lottery parameter vector over the set $\mathcal{S}'_t$: $\hat{r}'_t = \mathrm{argmax}_{\boldsymbol{r}\in\mathbb{R}^{n(m+1)}} \sum_{\ell=1}^{L} \sum_{\tau=1}^{T} \mathrm{rev}_{\boldsymbol{r}}\left(\boldsymbol{w}^{(\ell,t,\tau)}\right)$. With probability $1-\delta$, for every $t \in [T]$,*
$$\max_{\boldsymbol{r}\in\mathbb{R}^{n(m+1)}} \mathbb{E}_{\boldsymbol{v}\sim\mathcal{D}^{(t)}}\left[\mathrm{rev}_{\boldsymbol{r}}(\boldsymbol{v}) - \mathrm{rev}_{\hat{r}'_t}(\boldsymbol{v})\right] = \tilde{O}\left(\sqrt{\frac{1}{LT}\left(nm + \log\frac{T}{\delta}\right) + \frac{n^2}{Lm}\log\frac{nT}{\delta}}\right).$$

Theorem 4.2 implies that for any $\gamma \in (0,1)$, when $L = \tilde{\Omega}\left(n/\gamma^2\right)$, $T = \tilde{\Omega}(m)$, and there are more items than buyers $(m > n)$, the revenue of the lottery defined by $\hat{r}'_t$ is within $\gamma$ of optimal: $\max \mathbb{E}_{\boldsymbol{v}\sim\mathcal{D}^{(t)}}\left[\mathrm{rev}_{\boldsymbol{r}}(\boldsymbol{v})\right] - \mathbb{E}_{\boldsymbol{v}\sim\mathcal{D}^{(t)}}\left[\mathrm{rev}_{\hat{r}'_t}(\boldsymbol{v})\right] \leq \gamma$. Meanwhile, the best-known single-task sample complexity bound requires $L = \tilde{\Omega}\left(nm/\gamma^2\right)$ [6]. Our approach's per-task sample complexity is better by a multiplicative factor of $m$.

# 5    Conclusions

We provided guarantees for learning high-revenue mechanisms with noisy data. Learning in the presence of noise is particularly challenging in mechanism design because revenue functions are volatile, exhibiting many jump discontinuities. We were nonetheless able to provide revenue guarantees only under the assumption that the magnitude of the noise is bounded or sub-Gaussian. Our guarantees apply to the dominant approach to learning-based mechanism design: empirical revenue maximization (ERM). We demonstrated the application of our guarantees to multi-task mechanism design. We thus initiated the study of both learning with noisy data and multi-task learning in mechanism design.

**Future directions:** (1) We focused on learning with bounded noise rather than noise of arbitrary magnitude (the full adversarial game). This is an exciting future direction if any guarantees are indeed possible when bids may be arbitrarily altered, and may require the development of new learning algorithms beyond ERM. (2) A major bottleneck for empirical evaluations in learning-based mechanism design is the lack of public datasets—we are only aware of one, from eBay [44]. We focused on theoretical guarantees, but applied research on robust and multi-task learning in mechanism design is a great future direction. (3) Building on our results for multi-task learning, what other notions of task-relatedness permit strong multi-task sample complexity bounds?

**Negative societal impacts:** This paper falls into the broader line of research on using machine learning for mechanism design. In this direction, collecting individuals' data and using it to fine-tune prices could have negative privacy implications. The development of privacy-preserving approaches to mechanism design via machine learning is a great direction for future research.

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
