*Proof sketch of Lemma 3.4.* Suppose that $\mathrm{rev}_{\hat{r}}(v^{(\ell)}) > 0$. Let $i$ be the winner under the values $v^{(\ell)}$ and reserve $\hat{r}$. Since $v_i^{(\ell)} \geq \hat{r}_i$, we know that $w_i^{(\ell)} \geq \langle v_i^{(\ell)} - \epsilon \rangle \geq \langle \hat{r}_i - \epsilon \rangle$, so under the values $w^{(\ell)}$ and reserve $r_\epsilon$, there is at least one bidder whose bid is at least his reserve. Let $i'$ be the winner under the valuation vector $w^{(\ell)}$ and reserve $r_\epsilon$. We split this proof into four cases that depend on:

1. Whether or not $i = i'$, and
2. Whether—under $v^{(\ell)}$ and $\hat{r}$—there is another bidder besides $i$ whose bid is at least his reserve (in other words, whether or not $\{t : v_t^{(\ell)} \geq \hat{r}_t, t \neq i\} = \emptyset$).

In the first case, $i = i'$ and $\{t : v_t^{(\ell)} \geq \hat{r}_t, t \neq i\} \neq \emptyset$. Let $k$ be the index of the second-highest bidder in $v^{(\ell)}$ whose bid is above his reserve in $\hat{r}$: $k = \mathrm{argmax}_{t \neq i}\{v_t^{(\ell)} : v_t^{(\ell)} \geq \hat{r}_t\}$. This means that $\mathrm{rev}_{\hat{r}}(v^{(\ell)}) = \max\{\hat{r}_i, v_k^{(\ell)}\}$. Since $v_k^{(\ell)} \geq \hat{r}_k$, we have that $w_k^{(\ell)} \geq \langle v_k^{(\ell)} - \epsilon \rangle \geq \langle \hat{r}_k - \epsilon \rangle$. Since $k \neq i$ and $i = i'$, it must be that $k \neq i'$, so $\{t : w_t^{(\ell)} \geq \langle \hat{r}_t - \epsilon \rangle, t \neq i'\} \neq \emptyset$ (in particular, the set contains $k$). Let $k'$ be the index of the second-highest bidder in $w^{(\ell)}$ whose bid is above his reserve in $r_\epsilon$: $k' = \mathrm{argmax}_{t \neq i'}\{w_t^{(\ell)} : w_t^{(\ell)} \geq \langle \hat{r}_t - \epsilon \rangle\}$, which means that $\mathrm{rev}_{r_\epsilon}(w^{(\ell)}) = \max\{\hat{r}_{i'} - \epsilon, w_{k'}^{(\ell)}\}$. Since $k \in \{t : w_t^{(\ell)} \geq \langle \hat{r}_t - \epsilon \rangle, t \neq i'\}$, we know that $w_{k'}^{(\ell)} \geq w_k^{(\ell)} \geq v_k^{(\ell)} - \epsilon$. Putting this all together, we prove that $\mathrm{rev}_{r_\epsilon}(w^{(\ell)}) \geq \max\{\hat{r}_i - \epsilon, w_{k'}^{(\ell)}\} \geq \max\{\hat{r}_i, v_k^{(\ell)}\} - 2\epsilon = \mathrm{rev}_{\hat{r}}(v^{(\ell)}) - 2\epsilon$. We prove the other three cases in the appendix. $\square$

We use Lemma 3.4 to continue Theorem 3.3's proof. Let $I$ be the set of indices $\ell$ such that $\mathrm{rev}_{\hat{r}}(v^{(\ell)}) > 0$. By Lemma 3.4, $\sum_{\ell=1}^{L} \mathrm{rev}_{\hat{r}}(v^{(\ell)}) = \sum_{\ell \in I} \mathrm{rev}_{\hat{r}}(v^{(\ell)}) \leq \sum_{\ell \in I} \mathrm{rev}_{r_\epsilon}(w^{(\ell)}) + 2L\epsilon \leq \sum_{\ell=1}^{L} \mathrm{rev}_{r_\epsilon}(w^{(\ell)}) + 2L\epsilon \leq \sum_{\ell=1}^{L} \mathrm{rev}_{\hat{r}'}(w^{(\ell)}) + 2L\epsilon$.

Next, we prove that for any reserve vector $r$, revenue under the samples $v^{(\ell)}$ will only be higher than revenue under the samples $w^{(\ell)}$, which intuitively makes sense since $w^{(\ell)} \leq v^{(\ell)}$. The proof, which has a similar structure as the proof of Lemma 3.4, is in Appendix B.2.

**Lemma 3.5.** *For all samples* $\ell \in [L]$ *and reserve vectors* $r \in \mathbb{R}^n$, $\mathrm{rev}_r(w^{(\ell)}) \leq \mathrm{rev}_r(v^{(\ell)})$.

Since $\sum_{\ell=1}^{L} \mathrm{rev}_{\hat{r}}(v^{(\ell)}) \leq \sum \mathrm{rev}_{\hat{r}'}(w^{(\ell)}) +

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

# A  Additional related research

**Multi-task learning.**   Learning algorithms often require large training sets to find accurate models. *Multi-task learning* aims to leverage information across multiple related tasks to improve the learning outcomes per task. The majority of the existing literature on multi-task learning is empirical, whereas we focus on theoretical guarantees. Two early works on the sample complexity of multi-task learning [10, 12] study binary classification, whereas we study real-valued revenue functions. Blum et al. [17] also study multi-task binary classification. In their case, one can pool non-noisy data across different tasks to train the classifiers, which is the not the case in our setting. They also assume that there is a common optimal or close-to-optimal target function that performs well with respect to each task's distribution. We, on the other hand, do not assume the existence of a common set of parameters that yield nearly optimal revenue on each task. Moving beyond binary classification, Ando and Zhang [2] provide generalization guarantees that hold on average over the tasks, whereas we bound the error for each individual task.

**Mechanism design with side information.**   Devanur et al. [26] study single-item mechanism design via machine learning when the seller can see some public information—or *signal*—about each buyer. The signal indicates, for example, their annual income. Each signal defines a distinct marginal distribution over buyers' values, thus characterizing—in essence—a distinct learning task. Devanur et al. [26] show how to learn a mechanism that has high revenue in expectation over the draw of a fresh task (or in other words, the draw of a fresh signal). For us, there is no distribution over tasks: the tasks are fixed up front and we aim to learn a high-revenue mechanism for every task. Moreover, we study the multi-item, not single-item, setting.

**Transfer learning and domain adaptation.**   Transfer learning and domain adaptation are closely related to multi-task learning. In this setting, there is a *source domain* and a *target domain*. The learning algorithm receives training examples from the source domain (and perhaps a few training examples from the target domain), and aims to learn a hypothesis with low loss on the target domain. Research with provable guarantees on this topic include papers by Ben-David et al. [13], Mansour et al. [42], and McNamara and Balcan [43]. In our multi-task mechanism design setting, for any one task, we do not have enough data to provide strong generalization guarantees, let alone to provide strong guarantees if we were to apply the learned mechanism to another task, no matter how related it is. Galanti et al. [29] study a setting where there are a number of *source tasks* together with a *target task*. The source tasks are used to help in the target learning task, but optimizing the loss of the source tasks is not part of the learner's goal. This is in contrast to our goal, which is to learn a high-revenue mechanism per task.

**Approximate incentive compatibility.**   Prior research on approximate *incentive compatibility (IC)* has studied the conversion of $\eta$-IC mechanisms into IC mechanisms [e.g., 11, 21, 36]. A manipulable mechanism is $\eta$-IC if each agent can improve his utility by at most $\eta$ when he misreports his values by any arbitrary amount. In contrast, the mechanism classes we study in this paper are dominant strategy incentive compatible (DSIC). Our overall goal is to use the noisy training data to learn a mechanism that has high revenue in expectation over the fresh draw from the true distribution over buyers' values. Since the resulting mechanism is DSIC, we may assume that with respect to this fresh draw, the buyers' bids equal their true values, so no notion of approximate incentive compatibility is relevant. In our model, the adversary has an $\ell_\infty$-norm budget of $\epsilon$. This $\epsilon$ should not be confused with $\eta$ in the literature on $\eta$-IC mechanism design, which captures the maximum amount the buyer's utility can change when he misreports his bid.

# B  Additional proofs about learning under adversarial noise (Section 3)

**Notation.**   Throughout this appendix, we use the notation $\langle x \rangle = \max\{x, 0\}$.

**Fact 3.2.** *Fix a $p$-stable and $q$-convergent mechanism class. Let $\mathcal{S} = \{\boldsymbol{v}^{(1)}, \ldots, \boldsymbol{v}^{(L)}\}$ and $\mathcal{S}' = \{\boldsymbol{w}^{(1)}, \ldots, \boldsymbol{w}^{(L)}\}$ be two sets of valuation vectors such that for all $\ell \in [L]$, $\boldsymbol{v}^{(\ell)} - \boldsymbol{\epsilon} \leq \boldsymbol{w}^{(\ell)} \leq \boldsymbol{v}^{(\ell)}$. Let $\hat{\boldsymbol{r}}'$ be empirically optimal over $\mathcal{S}'$: $\hat{\boldsymbol{r}}' = \mathrm{argmax}_{\boldsymbol{r} \in \mathbb{R}^d} \sum_{\ell=1}^{L} \mathrm{rev}_{\boldsymbol{r}}\left(\boldsymbol{w}^{(\ell)}\right)$. With probability $1 - \delta$ over $\mathcal{S} \sim \mathcal{D}^L$, $\max_{\boldsymbol{r} \in \mathbb{R}^d} \mathbb{E}_{\boldsymbol{v} \sim \mathcal{D}}\left[\mathrm{rev}_{\boldsymbol{r}}(\boldsymbol{v})\right] - \mathbb{E}_{\boldsymbol{v} \sim \mathcal{D}}\left[\mathrm{rev}_{\hat{\boldsymbol{r}}'}(\boldsymbol{v})\right] \leq p(\epsilon, n, m) + 2q(\delta, L, n, m).$*

*Proof.* Let $\hat{r}$ be the parameter vector of the empirically optimal mechanism over the true samples $\mathcal{S}$: $\hat{r} = \operatorname{argmax}_{\boldsymbol{r} \in \mathbb{R}^d} \sum_{\ell=1}^{L} \operatorname{rev}_{\boldsymbol{r}} \left( \boldsymbol{v}^{(\ell)} \right)$. Next, let $\boldsymbol{r}^*$ be the parameter vector of the mechanism with the highest expected revenue over the distribution $\mathcal{D}$: $\boldsymbol{r}^* = \operatorname{argmax}_{\boldsymbol{r} \in \mathbb{R}^d} \mathbb{E}_{\boldsymbol{v} \sim \mathcal{D}} \left[ \operatorname{rev}_{\boldsymbol{r}}(\boldsymbol{v}) \right]$.

Since the mechanism class is $(p, q)$-robust, we know that with probability $1 - \delta$, the average revenue of any parameter $\boldsymbol{r}$ over the true samples $\mathcal{S}$ is close to its expected revenue:

$$\left| \mathbb{E}_{\boldsymbol{v} \sim \mathcal{D}} \left[ \operatorname{rev}_{\boldsymbol{r}}(\boldsymbol{v}) \right] - \frac{1}{L} \sum_{\ell=1}^{L} \operatorname{rev}_{\boldsymbol{r}} \left( \boldsymbol{v}^{(\ell)} \right) \right| \leq q(\delta, L, n, m). \tag{1}$$

Since $\hat{r}$ is the empirically optimal parameter vector over the true samples $\mathcal{S}$, the average revenue of $\boldsymbol{r}^*$ over $\mathcal{S}$ is at most the average revenue of $\hat{r}$ over $\mathcal{S}$. Combined with Equation (1) with $\boldsymbol{r} = \boldsymbol{r}^*$, this means that

$$\mathbb{E}_{\boldsymbol{v} \sim \mathcal{D}} \left[ \operatorname{rev}_{\boldsymbol{r}^*}(\boldsymbol{v}) \right] \leq q(\delta, L, n, m) + \frac{1}{L} \sum_{\ell=1}^{L} \operatorname{rev}_{\boldsymbol{r}^*} \left( \boldsymbol{v}^{(\ell)} \right)$$

$$- \frac{1}{L} \sum_{\ell=1}^{L} \operatorname{rev}_{\hat{r}} \left( \boldsymbol{v}^{(\ell)} \right) + \frac{1}{L} \sum_{\ell=1}^{L} \operatorname{rev}_{\hat{r}} \left( \boldsymbol{v}^{(\ell)} \right)$$

$$\leq q(\delta, L, n, m) + \frac{1}{L} \sum_{\ell=1}^{L} \operatorname{rev}_{\hat{r}} \left( \boldsymbol{v}^{(\ell)} \right).$$

Next, we combine this inequality and Equation (1) with $\boldsymbol{r} = \hat{r}'$:

$$\mathbb{E}_{\boldsymbol{v} \sim \mathcal{D}} \left[ \operatorname{rev}_{\boldsymbol{r}^*}(\boldsymbol{v}) \right] - \mathbb{E}_{\boldsymbol{v} \sim \mathcal{D}} \left[ \operatorname{rev}_{\hat{r}'}(\boldsymbol{v}) \right]$$

$$\leq \frac{1}{L} \sum_{\ell=1}^{L} \operatorname{rev}_{\hat{r}} \left( \boldsymbol{v}^{(\ell)} \right) - \operatorname{rev}_{\hat{r}'} \left( \boldsymbol{v}^{(\ell)} \right) + 2q(\delta, L, n, m)$$

Since the class is $(p, q)$-robust, $\frac{1}{L} \sum \operatorname{rev}_{\hat{r}} \left( \boldsymbol{v}^{(\ell)} \right) - \operatorname{rev}_{\hat{r}'} \left( \boldsymbol{v}^{(\ell)} \right) \leq p(\epsilon, n, m)$, so the fact holds. $\square$

**Lemma B.1.** *Let $f_1, \ldots, f_m$ be $m$ functions mapping $\mathbb{R}^n$ to $\mathbb{R}$. Then*

$$\max_{\boldsymbol{r}_1, \ldots, \boldsymbol{r}_m \in \mathbb{R}^n} \sum_{j=1}^{m} f_j(\boldsymbol{r}_j) = \sum_{j=1}^{m} \max_{\boldsymbol{r}_j \in \mathbb{R}^n} f_j(\boldsymbol{r}_j).$$

*Proof.* Clearly,

$$\max_{\boldsymbol{r}_1, \ldots, \boldsymbol{r}_m \in \mathbb{R}^n} \sum_{j=1}^{m} f_j(\boldsymbol{r}_j) \leq \sum_{j=1}^{m} \max_{\boldsymbol{r}_j \in \mathbb{R}^n} f_j(\boldsymbol{r}_j).$$

Suppose

$$\max_{\boldsymbol{r}_1, \ldots, \boldsymbol{r}_m \in \mathbb{R}^n} \sum_{j=1}^{m} f_j(\boldsymbol{r}_j) < \sum_{j=1}^{m} \max_{\boldsymbol{r}_j \in \mathbb{R}^n} f_j(\boldsymbol{r}_j).$$

Let $(\boldsymbol{q}_1, \ldots, \boldsymbol{q}_m)$ be a set of vectors in $\operatorname{argmax}_{\boldsymbol{r}_1, \ldots, \boldsymbol{r}_m \in \mathbb{R}^n} \sum_{j=1}^{m} f_j(\boldsymbol{r}_j)$ and for each $j \in [m]$, let $\boldsymbol{q}_j'$ be a vector in $\operatorname{argmax}_{\boldsymbol{r}_j \in \mathbb{R}^n} f_j(\boldsymbol{r}_j)$. By assumption $\sum_{j=1}^{m} f_j(\boldsymbol{q}_j) < \sum_{j=1}^{m} f_j(\boldsymbol{q}_j')$. However, this contradicts the fact that

$$\sum_{j=1}^{m} f_j(\boldsymbol{q}_j) = \max_{\boldsymbol{r}_1, \ldots, \boldsymbol{r}_m \in \mathbb{R}^n} \sum_{j=1}^{m} f_j(\boldsymbol{r}_j).$$

Therefore, the lemma statement holds. $\square$

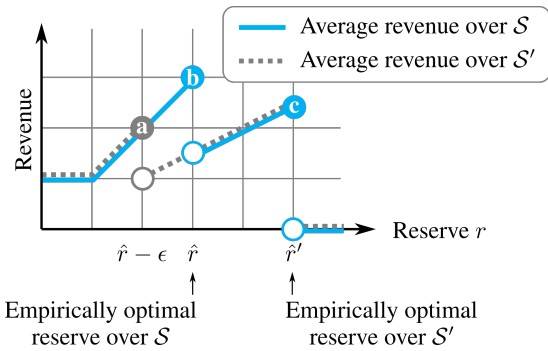

Figure 2: Illustration of Example B.3, which exemplifies Theorem B.2 when $\epsilon = 1$, $\boldsymbol{v}^{(1)} = (1,3)$, $\boldsymbol{w}^{(1)} = (1,2)$, and $\boldsymbol{v}^{(2)} = \boldsymbol{w}^{(2)} = (1,5)$. The blue solid line is average revenue over $\mathcal{S} = \{\boldsymbol{v}^{(1)}, \boldsymbol{v}^{(2)}\}$ as a function of the reserve $r$ and the grey dotted line is average revenue over $\mathcal{S}' = \{\boldsymbol{w}^{(1)}, \boldsymbol{w}^{(2)}\}$. The lemma shows that the difference between points (b) and (c) is small.

## B.1 Second-price auctions with anonymous reserves

In this section, we analyze second-price auctions with anonymous reserves, which is a bit less involved than our analysis of multi-item *non-anonymous* reserves. We assume there is only one unit of the item for sale. A single-item second-price auction with an anonymous reserve is defined by a reserve price $r \in \mathbb{R}$. Each bidder submits a real-valued bid to the auctioneer. The highest bidder wins so long as her bid is above the reserve and pays the maximum of the second-highest bid and the reserve. We prove that this mechanism class is $p$-stable.

**Theorem B.2.** *The class of single-item second-price auctions with anonymous reserves is $p$-stable with $p(\epsilon, n, m) = 2\epsilon$.*

We first provide an example of Theorem B.2's implications and a proof overview.

**Example B.3.** Figure 2 llustrates Theorem B.2 when $\epsilon = 1$, $\boldsymbol{v}^{(1)} = (1,3)$, $\boldsymbol{w}^{(1)} = (1,2)$, and $\boldsymbol{v}^{(2)} = \boldsymbol{w}^{(2)} = (1,5)$. The blue solid line is average revenue over $\mathcal{S} = \{\boldsymbol{v}^{(1)}, \boldsymbol{v}^{(2)}\}$, which equals $\frac{1}{2}\sum_{\ell=1}^{2} \text{rev}_r(\boldsymbol{v}^{(\ell)})$, as a function of the reserve $r$. The grey dotted line is average revenue over $\mathcal{S}' = \{\boldsymbol{w}^{(1)}, \boldsymbol{w}^{(2)}\}$. The reserves that maximize average revenue over $\mathcal{S}$ and $\mathcal{S}'$ are $\hat{r} = 3$ and $\hat{r}' = 5$, respectively. Theorem B.2 states that the difference between average revenue over $\mathcal{S}$ when $r = \hat{r}'$ and $r = \hat{r}$ is small. One might hope that $\boldsymbol{v}^{(\ell)} - \boldsymbol{\epsilon} \leq \boldsymbol{w}^{(\ell)} \leq \boldsymbol{v}^{(\ell)}$ would imply that $\hat{r} - \epsilon \leq \hat{r}' \leq \hat{r}$, which might imply the lemma holds. Figure 2 shows this is not the case. We first show that the difference between points (a) and (b) is small, where (a) is the average revenue of $\hat{r} - \epsilon$ over $\mathcal{S}'$ and (b) is the average revenue of $\hat{r}$ over $\mathcal{S}$. We then show that this implies that Theorem B.2 holds: the difference between points (b) and (c) is small, where (c) is the average revenue of $\hat{r}'$ over $\mathcal{S}$.

*Proof of Theorem B.2.* For any $L \geq 1$, let $\mathcal{S} = \{\boldsymbol{v}^{(1)}, \ldots, \boldsymbol{v}^{(L)}\} \subset [0,1]^n$ and $\mathcal{S}' = \{\boldsymbol{w}^{(1)}, \ldots, \boldsymbol{w}^{(L)}\} \subset [0,1]^n$ be two arbitrary sets of valuation vectors such that $\boldsymbol{v}^{(\ell)} - \boldsymbol{\epsilon} \leq \boldsymbol{w}^{(\ell)} \leq \boldsymbol{v}^{(\ell)}$ for all $\ell \in [L]$. Let $\hat{r}'$ be the empirically optimal reserve over the set $\mathcal{S}'$ and $\hat{r}$ be the empirically optimal reserve over the set $\mathcal{S}$. We prove that $\frac{1}{L}\sum_{\ell=1}^{L} \text{rev}_{\hat{r}}(\boldsymbol{v}^{(\ell)}) - \text{rev}_{\hat{r}'}(\boldsymbol{v}^{(\ell)}) \leq \epsilon$.

This proof relies on two key claims. The first, Claim B.4 states that if we shift the reserve $\hat{r}$—which is empirically optimal for the samples in $\mathcal{S}$—down by an additive factor of $\epsilon$ and apply this reserve to the samples in $\mathcal{S}'$, little revenue is lost. Specifically, Claim B.4 guarantees that for all $\ell \in [L]$, if $\text{rev}_{\hat{r}}(\boldsymbol{v}^{(\ell)}) > 0$, then $\text{rev}_{\langle \hat{r}-\epsilon \rangle}(\boldsymbol{w}^{(\ell)}) \geq \text{rev}_{\hat{r}}(\boldsymbol{v}^{(\ell)}) - 2\epsilon$.

**Claim B.4.** *For all $\ell \in [L]$, if $\text{rev}_{\hat{r}}(\boldsymbol{v}^{(\ell)}) > 0$, then $\text{rev}_{\langle \hat{r}-\epsilon \rangle}(\boldsymbol{w}^{(\ell)}) \geq \text{rev}_{\hat{r}}(\boldsymbol{v}^{(\ell)}) - 2\epsilon$.*

*Proof of Claim B.4.* Fix an arbitrary sample $\ell \in [L]$ such that $\text{rev}_{\hat{r}}(\boldsymbol{v}^{(\ell)}) > 0$. Let $i$ be the bidder with the highest valuation in $\boldsymbol{v}^{(\ell)}$. Since $\text{rev}_{\hat{r}}(\boldsymbol{v}^{(\ell)}) > 0$, it must be that bidder $i$'s bid is at least the reserve $\hat{r}$. This implies that $w_i^{(\ell)} \geq \langle v_i^{(\ell)} - \epsilon \rangle \geq \langle \hat{r} - \epsilon \rangle$, so under the valuation vector $\boldsymbol{w}^{(\ell)}$ and

reserve $\langle \hat{r} - \epsilon \rangle$, there is at least one bidder whose bid is at least the reserve. Let $i'$ be the highest bidder (or in other words, the winner) under the valuation vector $\boldsymbol{w}^{(\ell)}$. There are two cases, depending on whether or not $i = i'$. In both cases, we show that $\text{rev}_{\langle \hat{r} - \epsilon \rangle}\left(\boldsymbol{w}^{(\ell)}\right) \geq \text{rev}_{\hat{r}}\left(\boldsymbol{v}^{(\ell)}\right) - 2\epsilon$, so the claim holds. We use the notation $k$ to denote the second-highest bidder under $\boldsymbol{v}^{(\ell)}$ $\left(k = \text{argmax}_{t \neq i} v_t^{(\ell)}\right)$ and $k'$ to denote the second-highest bidder under $\boldsymbol{w}^{(\ell)}$ $\left(k' = \text{argmax}_{t \neq i'} w_t^{(\ell)}\right)$. Using this notation, $\text{rev}_{\hat{r}}\left(\boldsymbol{v}^{(\ell)}\right) = \max\left\{\hat{r}, v_k^{(\ell)}\right\}$ and $\text{rev}_{\langle \hat{r} - \epsilon \rangle}\left(\boldsymbol{w}^{(\ell)}\right) = \max\left\{\langle \hat{r} - \epsilon \rangle, w_{k'}^{(\ell)}\right\} = \max\left\{\hat{r} - \epsilon, w_{k'}^{(\ell)}\right\}$.

In the first case, $i = i'$. Since $k \neq i$, it must be that $k \neq i'$. Bidder $i'$ is the highest bidder under $\boldsymbol{w}^{(\ell)}$, so it must be that bidder $k$ is the $t^{th}$ highest bidder under $\boldsymbol{w}^{(\ell)}$ for some $t \geq 2$. Since bidder $k'$ is the second-highest bidder under $\boldsymbol{w}^{(\ell)}$, this means that $w_{k'}^{(\ell)} \geq w_k^{(\ell)} \geq v_k^{(\ell)} - \epsilon$. Therefore, $\text{rev}_{\langle \hat{r} - \epsilon \rangle}\left(\boldsymbol{w}^{(\ell)}\right) = \max\left\{\hat{r} - \epsilon, w_{k'}^{(\ell)}\right\} \geq \max\left\{\hat{r}, w_{k'}^{(\ell)}\right\} - \epsilon \geq \max\left\{\hat{r}, v_k^{(\ell)} - \epsilon\right\} - \epsilon \geq \max\left\{\hat{r}, v_k^{(\ell)}\right\} - 2\epsilon = \text{rev}_{\hat{r}}\left(\boldsymbol{v}^{(\ell)}\right) - 2\epsilon$.

In the second case, $i \neq i'$. Bidder $i'$ is the highest bidder under $\boldsymbol{w}^{(\ell)}$, so it must be that bidder $i$ is the $t^{th}$ highest bidder under $\boldsymbol{w}^{(\ell)}$ for some $t \geq 2$. Since bidder $k'$ is the second-highest bidder under $\boldsymbol{w}^{(\ell)}$, this means that $w_{k'}^{(\ell)} \geq w_i^{(\ell)} \geq v_i^{(\ell)} - \epsilon$. Therefore, $\text{rev}_{\langle \hat{r} - \epsilon \rangle}\left(\boldsymbol{w}^{(\ell)}\right) \geq w_{k'}^{(\ell)} \geq v_i^{(\ell)} - \epsilon \geq \text{rev}_{\hat{r}}\left(\boldsymbol{v}^{(\ell)}\right) - \epsilon$, where the final inequality holds because the auction's revenue is always at most the highest bid. $\qquad \square$

Let $I \subseteq [L]$ be the set of indices $\ell$ such that $\text{rev}_{\hat{r}}\left(\boldsymbol{v}^{(\ell)}\right) > 0$ (for all other indices, $\text{rev}_{\hat{r}}\left(\boldsymbol{v}^{(\ell)}\right) = 0$). By Claim B.4, for all $\ell \in I$, $\text{rev}_{\hat{r}}\left(\boldsymbol{v}^{(\ell)}\right) \leq \text{rev}_{\langle \hat{r} - \epsilon \rangle}\left(\boldsymbol{w}^{(\ell)}\right) + 2\epsilon$. Therefore,

$$\sum_{\ell=1}^{L} \text{rev}_{\hat{r}}\left(\boldsymbol{v}^{(\ell)}\right) = \sum_{\ell \in I} \text{rev}_{\hat{r}}\left(\boldsymbol{v}^{(\ell)}\right) \leq \sum_{\ell \in I} \text{rev}_{\langle \hat{r} - \epsilon \rangle}\left(\boldsymbol{w}^{(\ell)}\right) + 2L\epsilon \leq \sum_{\ell=1}^{L} \text{rev}_{\langle \hat{r} - \epsilon \rangle}\left(\boldsymbol{w}^{(\ell)}\right) + 2L\epsilon. \tag{2}$$

(In Figure 2, this proves the claim that the difference between points (a) and (b) is small.) Since $\hat{r}'$ is the empirically optimal reserve under the perturbed training instances $\mathcal{S}'$ (mathematically, $\hat{r}' = \text{argmax}_{r \in [0,1]} \sum_{\ell=1}^{L} \text{rev}_r\left(\boldsymbol{w}^{(\ell)}\right)$), Equation (2) implies that

$$\sum_{\ell=1}^{L} \text{rev}_{\hat{r}}\left(\boldsymbol{v}^{(\ell)}\right) \leq 2L\epsilon + \sum_{\ell=1}^{L} \text{rev}_{\hat{r}'}\left(\boldsymbol{w}^{(\ell)}\right). \tag{3}$$

Next, we prove that for any reserve $r$, revenue under the samples $\boldsymbol{v}^{(\ell)}$ will only be higher than revenue under the samples $\boldsymbol{w}^{(\ell)}$. This claim intuitively makes sense since $\boldsymbol{w}^{(\ell)} \leq \boldsymbol{v}^{(\ell)}$.

**Claim B.5.** *For every sample $\ell \in [L]$ and any reserve $r \in [0, 1]$, $\text{rev}_r\left(\boldsymbol{w}^{(\ell)}\right) \leq \text{rev}_r\left(\boldsymbol{v}^{(\ell)}\right)$.*

*Proof of Claim B.5.* Fix an arbitrary index $\ell \in [L]$ such that $\text{rev}_r\left(\boldsymbol{w}^{(\ell)}\right) > 0$ (if $\text{rev}_r\left(\boldsymbol{w}^{(\ell)}\right) = 0$, then the claim clearly holds). Let $i'$ be the highest bidder under the valuation vector $\boldsymbol{w}^{(\ell)}$. Since $\text{rev}_r\left(\boldsymbol{w}^{(\ell)}\right) > 0$, it must be that the bid of bidder $i'$ is at least the reserve $r$. Since $v_{i'}^{(\ell)} \geq w_{i'}^{(\ell)}$, we know that $v_{i'}^{(\ell)} \geq r$ as well. Therefore, under the valuation vector $\boldsymbol{v}^{(\ell)}$ and reserve $r$, there is at least one bidder whose bid is at least the reserve. Let $i$ be the highest bidder (or in other words, the winner) under the valuation vector $\boldsymbol{v}^{(\ell)}$. As in the proof of Claim B.4, there are two cases, depending on whether or not $i = i'$. In both cases, we show that $\text{rev}_r\left(\boldsymbol{w}^{(\ell)}\right) \leq \text{rev}_r\left(\boldsymbol{v}^{(\ell)}\right)$, so the claim holds. Again, we use the notation $k$ to denote the second-highest bidder under $\boldsymbol{v}^{(\ell)}$ $\left(k = \text{argmax}_{t \neq i} v_t^{(\ell)}\right)$ and $k'$ to denote the second-highest bidder under $\boldsymbol{w}^{(\ell)}$ $\left(k' = \text{argmax}_{t \neq i'} w_t^{(\ell)}\right)$. Using this notation, $\text{rev}_{\hat{r}}\left(\boldsymbol{v}^{(\ell)}\right) = \max\left\{\hat{r}, v_k^{(\ell)}\right\}$ and $\text{rev}_{\hat{r}}\left(\boldsymbol{w}^{(\ell)}\right) = \max\left\{\hat{r}, w_{k'}^{(\ell)}\right\}$.

In the first case, $i = i'$. Since $k' \neq i'$, it must be that $k' \neq i$. Therefore, bidder $k'$ is the $t^{th}$ highest bidder under the valuation vector $\boldsymbol{v}^{(\ell)}$ for some $t \geq 2$. Since bidder $k$ is the second-highest bidder under $\boldsymbol{v}^{(\ell)}$, we know that $v_k^{(\ell)} \geq v_{k'}^{(\ell)} \geq w_{k'}^{(\ell)}$. Therefore, $\text{rev}_r\left(\boldsymbol{w}^{(\ell)}\right) = \max\left\{r, w_{k'}^{(\ell)}\right\} \leq \max\left\{r, v_k^{(\ell)}\right\} = \text{rev}_r\left(\boldsymbol{v}^{(\ell)}\right)$.

In the second case, $i \neq i'$, so it must be that bidder $i'$ is the $t^{th}$ highest bidder under the valuation vector $\boldsymbol{v}^{(\ell)}$ for some $t \geq 2$. Since bidder $k$ is the second-highest bidder under $\boldsymbol{v}^{(\ell)}$, we know that $v_k^{(\ell)} \geq v_{i'}^{(\ell)}$. Therefore, $\text{rev}_r\left(\boldsymbol{v}^{(\ell)}\right) \geq v_k^{(\ell)} \geq v_{i'}^{(\ell)} \geq w_{i'}^{(\ell)} \geq \text{rev}_r\left(\boldsymbol{w}^{(\ell)}\right)$, where the final inequality follows from the fact that the revenue of the auction is always at most the valuation of the highest bidder. $\qquad\square$

Equation (3) and Claim B.5 imply $\sum_{\ell=1}^{L} \text{rev}_{\hat{r}}\left(\boldsymbol{v}^{(\ell)}\right) \leq 2L\epsilon + \sum_{\ell=1}^{L} \text{rev}_{\hat{r}'}\left(\boldsymbol{v}^{(\ell)}\right)$, as claimed. $\qquad\square$

Morgenstern and Roughgarden [46] prove the mechanism class is $q$-convergent with $q(\delta, L, n, m) = O\left(\sqrt{\frac{1}{L}\log\frac{1}{\delta}}\right)$. This fact together with Fact 3.2 and Theorem B.2 implies the following guarantee:

**Corollary B.6.** *Let* $\mathcal{S} = \left\{\boldsymbol{v}^{(1)}, \ldots, \boldsymbol{v}^{(L)}\right\} \subset \mathbb{R}_{\geq 0}^{n}$ *and* $\mathcal{S}' = \left\{\boldsymbol{w}^{(1)}, \ldots, \boldsymbol{w}^{(L)}\right\} \subset \mathbb{R}_{\geq 0}^{n}$ *be two sets such that for all* $\ell \in [L]$, $\boldsymbol{v}^{(\ell)} - \boldsymbol{\epsilon} \leq \boldsymbol{w}^{(\ell)} \leq \boldsymbol{v}^{(\ell)}$. *Let* $\hat{r}' = \text{argmax} \sum_{\ell=1}^{L} \text{rev}_r\left(\boldsymbol{w}^{(\ell)}\right)$. *With high probability,* $\max \mathbb{E}_{\boldsymbol{v}\sim\mathcal{D}}[\text{rev}_r(\boldsymbol{v})] - \mathbb{E}_{\boldsymbol{v}\sim\mathcal{D}}[\text{rev}_{\hat{r}'}(\boldsymbol{v})] = \tilde{O}\left(\epsilon + \sqrt{\frac{1}{L}}\right)$.

### B.2 Additional proofs about second-price auctions with non-anonymous reserves

**Lemma 3.4.** *Let* $\boldsymbol{r}_{\epsilon} = (\langle\hat{r}_1 - \epsilon\rangle, \ldots, \langle\hat{r}_n - \epsilon\rangle)$. *If* $\text{rev}_{\hat{r}}\left(\boldsymbol{v}^{(\ell)}\right) > 0$, *then* $\text{rev}_{\boldsymbol{r}_{\epsilon}}\left(\boldsymbol{w}^{(\ell)}\right) \geq \text{rev}_{\hat{r}}\left(\boldsymbol{v}^{(\ell)}\right) - 2\epsilon$.

*Proof.* Fix an arbitrary index $\ell \in [L]$ such that $\text{rev}_{\hat{r}}\left(\boldsymbol{v}^{(\ell)}\right) > 0$. Let $i$ be the winner under the valuation vector $\boldsymbol{v}^{(\ell)}$ and reserve vector $\hat{r}$. Since $v_i^{(\ell)} \geq \hat{r}_i$, we know that $w_i^{(\ell)} \geq \left\langle v_i^{(\ell)} - \epsilon\right\rangle \geq \langle\hat{r}_i - \epsilon\rangle$, so under the valuation vector $\boldsymbol{w}^{(\ell)}$ and reserve vector $\boldsymbol{r}_{\epsilon}$, there is at least one bidder whose bid is at least his reserve. Let $i'$ be the winner under the valuation vector $\boldsymbol{w}^{(\ell)}$ and reserve vector $\boldsymbol{r}_{\epsilon}$.

As in the proof of Claim B.4, we split this proof into cases. This proof, however, is a bit more involved because there are four cases, instead of two. These four cases depend on whether or not $i = i'$ (as in Claim B.4) and whether or not revenue depends on a second-highest bidder under the valuation vector $\boldsymbol{v}^{(\ell)}$ and reserve vector $\hat{r}$ (in other words, whether or not $\left\{t : v_t^{(\ell)} \geq \hat{r}_t, t \neq i\right\} = \emptyset$). In all four cases, we show that $\text{rev}_{\boldsymbol{r}_{\epsilon}}\left(\boldsymbol{w}^{(\ell)}\right) \geq \text{rev}_{\hat{r}}\left(\boldsymbol{v}^{(\ell)}\right) - 2\epsilon$, so the claim holds.

**Case 1:** $i = i'$ **and** $\left\{t : v_t^{(\ell)} \geq \hat{r}_t, t \neq i\right\} \neq \emptyset$. Let $k$ be the index of the second-highest bidder in $\boldsymbol{v}^{(\ell)}$ whose bid is above his reserve in $\hat{r}$: $k = \text{argmax}_{t\neq i}\left\{v_t^{(\ell)} : v_t^{(\ell)} \geq \hat{r}_t\right\}$. This means that $\text{rev}_{\hat{r}}\left(\boldsymbol{v}^{(\ell)}\right) = \max\left\{\hat{r}_i, v_k^{(\ell)}\right\}$. Since $v_k^{(\ell)} \geq \hat{r}_k$, we have that $w_k^{(\ell)} \geq \left\langle v_k^{(\ell)} - \epsilon\right\rangle \geq \langle\hat{r}_k - \epsilon\rangle$. Since $k \neq i$ and $i = i'$, it must be that $k \neq i'$, so $\left\{t : w_t^{(\ell)} \geq \langle\hat{r}_t - \epsilon\rangle, t \neq i'\right\} \neq \emptyset$ (in particular, the set contains $k$). Let $k'$ be the index of the second-highest bidder in $\boldsymbol{w}^{(\ell)}$ whose bid is above his reserve in $\boldsymbol{r}_{\epsilon}$: $k' = \text{argmax}_{t\neq i'}\left\{w_t^{(\ell)} : w_t^{(\ell)} \geq \langle\hat{r}_t - \epsilon\rangle\right\}$, which means that $\text{rev}_{\boldsymbol{r}_{\epsilon}}\left(\boldsymbol{w}^{(\ell)}\right) = \max\left\{\langle\hat{r}_{i'} - \epsilon\rangle, w_{k'}^{(\ell)}\right\} = \max\left\{\hat{r}_{i'} - \epsilon, w_{k'}^{(\ell)}\right\}$. Since $k \in \left\{t : w_t^{(\ell)} \geq \langle\hat{r}_t - \epsilon\rangle, t \neq i'\right\}$, we know that $w_{k'}^{(\ell)} \geq w_k^{(\ell)} \geq v_k^{(\ell)} - \epsilon$, which means that $\text{rev}_{\boldsymbol{r}_{\epsilon}}\left(\boldsymbol{w}^{(\ell)}\right) = \max\left\{\hat{r}_{i'} - \epsilon, w_{k'}^{(\ell)}\right\} = \max\left\{\hat{r}_i - \epsilon, w_{k'}^{(\ell)}\right\} \geq \max\left\{\hat{r}_i, w_{k'}^{(\ell)}\right\} - \epsilon \geq \max\left\{\hat{r}_i, v_k^{(\ell)} - \epsilon\right\} - \epsilon \geq \max\left\{\hat{r}_i, v_k^{(\ell)}\right\} - 2\epsilon = \text{rev}_{\hat{r}}\left(\boldsymbol{v}^{(\ell)}\right) - 2\epsilon$.

**Case 2:** $i = i'$ **and** $\left\{t : v_t^{(\ell)} \geq \hat{r}_t, t \neq i\right\} = \emptyset.$ In this case, $\text{rev}_{\hat{r}}\left(v^{(\ell)}\right) = \hat{r}_i$, so $\text{rev}_{r_\epsilon}\left(w^{(\ell)}\right) \geq \hat{r}_{i'} - \epsilon = \hat{r}_i - \epsilon = \text{rev}_{\hat{r}}\left(v^{(\ell)}\right) - \epsilon.$

**Case 3:** $i \neq i'$ **and** $\left\{t : v_t^{(\ell)} \geq \hat{r}_t, t \neq i\right\} \neq \emptyset.$ We know that $v_i^{(\ell)} \geq \hat{r}_i$, which means that $w_i^{(\ell)} \geq \left\langle v_i^{(\ell)} - \epsilon \right\rangle \geq \left\langle \hat{r}_i - \epsilon \right\rangle$. Since $i \neq i'$, it must be that $\left\{t : w_t^{(\ell)} \geq \left\langle \hat{r}_t - \epsilon \right\rangle, t \neq i'\right\} \neq \emptyset$ (in particular, the set contains $i$). Let $k'$ be the index of the second-highest bidder in $w^{(\ell)}$ whose bid is above his reserve in $r_\epsilon$: $k' = \text{argmax}_{t \neq i'}\left\{w_t^{(\ell)} : w_t^{(\ell)} \geq \left\langle \hat{r}_t - \epsilon \right\rangle\right\}$, which means that $\text{rev}_{r_\epsilon}\left(w^{(\ell)}\right) = \max\left\{\left\langle \hat{r}_{i'} - \epsilon \right\rangle, w_{k'}^{(\ell)}\right\}$. Since $i \in \left\{t : w_t^{(\ell)} \geq \left\langle \hat{r}_t - \epsilon \right\rangle, t \neq i'\right\}$, it must be that $w_{k'}^{(\ell)} \geq w_i^{(\ell)} \geq v_i^{(\ell)} - \epsilon$. Therefore, $\text{rev}_{r_\epsilon}\left(w^{(\ell)}\right) \geq w_{k'}^{(\ell)} \geq v_i^{(\ell)} - \epsilon \geq \text{rev}_{\hat{r}}\left(v^{(\ell)}\right) - \epsilon$, where the final inequality holds because the revenue of the auction is never more than the highest bid.

**Case 4:** $i \neq i'$ **and** $\left\{t : v_t^{(\ell)} \geq \hat{r}_t, t \neq i\right\} = \emptyset.$ Suppose that $\left\{t : w_t^{(\ell)} \geq \left\langle \hat{r}_t - \epsilon \right\rangle, t \neq i'\right\} = \emptyset$. Since $i \neq i'$, this means that $\left\langle \hat{r}_i - \epsilon \right\rangle > w_i^{(\ell)}$. Since $w_i^{(\ell)} \geq 0$, this inequality implies that $\hat{r}_i - \epsilon > w_i^{(\ell)} \geq v_i^{(\ell)} - \epsilon$. However, this means that $\hat{r}_i > v_i^{(\ell)}$, which is a contradiction. Therefore, $\left\{t : w_t^{(\ell)} \geq \left\langle \hat{r}_t - \epsilon \right\rangle, t \neq i'\right\} \neq \emptyset$. Let $k'$ be the index of the second-highest bidder in $w^{(\ell)}$ whose bid is above his reserve in $r_\epsilon$, or in other words, $k' = \text{argmax}_{t \neq i'}\left\{w_t^{(\ell)} : w_t^{(\ell)} \geq \left\langle \hat{r}_t - \epsilon \right\rangle\right\}$. Since $i \in \left\{t : w_t^{(\ell)} \geq \left\langle \hat{r}_t - \epsilon \right\rangle, t \neq i'\right\}$, we know that $w_{k'}^{(\ell)} \geq w_i^{(\ell)} \geq v_i^{(\ell)} - \epsilon$, so

$$\text{rev}_{r_\epsilon}\left(w^{(\ell)}\right) = \max\left\{\left\langle \hat{r}_{i'} - \epsilon \right\rangle, w_{k'}^{(\ell)}\right\} \geq w_{k'}^{(\ell)} \geq v_i^{(\ell)} - \epsilon \geq \text{rev}_{\hat{r}}\left(v^{(\ell)}\right) - \epsilon,$$

where the final inequality holds because the revenue of the auction is never more than the highest bid. □

**Lemma 3.5.** *For all samples $\ell \in [L]$ and reserve vectors $r \in \mathbb{R}^n$, $\text{rev}_r\left(w^{(\ell)}\right) \leq \text{rev}_r\left(v^{(\ell)}\right)$.*

*Proof.* Fix an arbitrary index $\ell \in [L]$ such that $\text{rev}_r\left(w^{(\ell)}\right) > 0$ (if $\text{rev}_r\left(w^{(\ell)}\right) = 0$, then the claim clearly holds). Let $i'$ be the winner under the valuation vector $w^{(\ell)}$ and reserve vector $r$. Since $w_{i'}^{(\ell)} \geq r_{i'}$ and $v_{i'}^{(\ell)} \geq w_{i'}^{(\ell)}$, we know that $v_{i'}^{(\ell)} \geq r_{i'}$. Therefore, under the valuation vector $v^{(\ell)}$ and reserve vector $r$, there is at least one bidder whose bid is at least his reserve. Let $i$ be the winner under the valuation vector $v^{(\ell)}$ and reserve vector $r$.

There are four cases, depending on whether or not $i = i'$ and whether or not revenue depends on a second-highest bidder under the valuation vector $w^{(\ell)}$ and reserve vector $r$ (in other words, whether or not $\left\{t : w_t^{(\ell)} \geq r_t, t \neq i'\right\} = \emptyset$). As in the proof of Lemma 3.4, splitting the analysis into these four cases helps us deal with the challenge summarized by the proof sketch in the main body. In all four cases, we show that $\text{rev}_r\left(w^{(\ell)}\right) \leq \text{rev}_r\left(v^{(\ell)}\right)$, so the claim holds.

**Case 1:** $i = i'$ **and** $\left\{t : w_t^{(\ell)} \geq r_t, t \neq i'\right\} \neq \emptyset.$ Let $k'$ be the index of the second-highest bidder in $w^{(\ell)}$ whose bid is above his reserve: $k' = \text{argmax}_{t \neq i'}\left\{w_t^{(\ell)} : w_t^{(\ell)} \geq r_t\right\}$, so $\text{rev}_r\left(w^{(\ell)}\right) = \max\left\{r_{i'}, w_{k'}^{(\ell)}\right\}$. Since $r_{k'} \leq w_{k'}^{(\ell)} \leq v_{k'}^{(\ell)}$, $k' \neq i'$, and $i = i'$, we know that $\left\{t : v_t^{(\ell)} \geq r_t, t \neq i\right\} \neq \emptyset$ (in particular, the set contains $k'$). Let $k$ be the index of the second-highest bidder in $v^{(\ell)}$ whose bid is above his reserve: $k = \text{argmax}_{t \neq i}\left\{v_t^{(\ell)} : v_t^{(\ell)} \geq r_t\right\}$. Since $k' \in \left\{t : v_t^{(\ell)} \geq r_t, t \neq i\right\}$, it must be that $v_k^{(\ell)} \geq v_{k'}^{(\ell)} \geq w_{k'}^{(\ell)}$. Therefore, $\text{rev}_r\left(w^{(\ell)}\right) = \max\left\{r_{i'}, w_{k'}^{(\ell)}\right\} = \max\left\{r_i, w_{k'}^{(\ell)}\right\} \leq \max\left\{r_i, v_k^{(\ell)}\right\} = \text{rev}_r\left(v^{(\ell)}\right).$

**Case 2:** $i = i'$ **and** $\left\{t : w_t^{(\ell)} \geq r_t, t \neq i'\right\} = \emptyset.$ In this case, $\text{rev}_{\boldsymbol{r}}\left(\boldsymbol{w}^{(\ell)}\right) = r_{i'} = r_i \leq \text{rev}_{\boldsymbol{r}}\left(\boldsymbol{v}^{(\ell)}\right).$

**Case 3:** $i \neq i'$ **and** $\left\{t : w_t^{(\ell)} \geq r_t, t \neq i'\right\} \neq \emptyset.$ Since $r_{i'} \leq w_{i'}^{(\ell)} \leq v_{i'}^{(\ell)}$ and $i' \neq i$, we know that $\left\{t : v_t^{(\ell)} \geq r_t, t \neq i\right\} \neq \emptyset$ (in particular, the set contains $i'$). Let $k$ be the index of the second-highest bidder in $\boldsymbol{v}^{(\ell)}$ whose bid is above his reserve: $k = \text{argmax}_{t \neq i}\left\{v_t^{(\ell)} : v_t^{(\ell)} \geq r_t\right\}$, so $\text{rev}_{\boldsymbol{r}}\left(\boldsymbol{v}^{(\ell)}\right) = \max\left\{r_i, v_k^{(\ell)}\right\}$. Since $i' \in \left\{t : v_t^{(\ell)} \geq r_t, t \neq i\right\}$, it must be that $v_k^{(\ell)} \geq v_{i'}^{(\ell)}$. Moreover, by assumption, $v_{i'}^{(\ell)} \geq w_{i'}^{(\ell)}$. Therefore, $\text{rev}_{\boldsymbol{r}}\left(\boldsymbol{v}^{(\ell)}\right) \geq v_k^{(\ell)} \geq v_{i'}^{(\ell)} \geq w_{i'}^{(\ell)} \geq \text{rev}_{\boldsymbol{r}}\left(\boldsymbol{w}^{(\ell)}\right)$, where the final inequality holds because the revenue of a second-price auction is never more than the highest bid.

**Case 4:** $i \neq i'$ **and** $\left\{t : w_t^{(\ell)} \geq r_t, t \neq i'\right\} = \emptyset.$ Suppose that $\left\{t : v_t^{(\ell)} \geq r_t, t \neq i\right\} = \emptyset.$ Since $i \neq i'$, this means that $r_{i'} > v_{i'}^{(\ell)} \geq w_{i'}^{(\ell)}$, which is a contradiction. Therefore, $\left\{t : v_t^{(\ell)} \geq r_t, t \neq i\right\} \neq \emptyset$. Let $k$ be the index of the second-highest bidder in $\boldsymbol{v}^{(\ell)}$ whose bid is above his reserve: $k = \text{argmax}_{t \neq i}\left\{v_t^{(\ell)} : v_t^{(\ell)} \geq r_t\right\}$, so $\text{rev}_{\boldsymbol{r}}\left(\boldsymbol{v}^{(\ell)}\right) = \max\left\{r_i, v_k^{(\ell)}\right\}$. Since $i' \in \left\{t : v_t^{(\ell)} \geq r_t, t \neq i\right\}$, it must be that $v_k^{(\ell)} \geq v_{i'}^{(\ell)}$. Moreover, by assumption, $v_{i'}^{(\ell)} \geq w_{i'}^{(\ell)}$. Therefore, $\text{rev}_{\boldsymbol{r}}\left(\boldsymbol{v}^{(\ell)}\right) \geq v_k^{(\ell)} \geq v_{i'}^{(\ell)} \geq w_{i'}^{(\ell)} \geq \text{rev}_{\boldsymbol{r}}\left(\boldsymbol{w}^{(\ell)}\right).$ $\qquad\square$

**Theorem 3.6.** *The set of multi-item non-anonymous second-price auctions under $n$ additive buyers is p-stable with $p(\epsilon, n, m) = 2m\epsilon$.*

*Proof.* Let $\mathcal{S} = \left\{\boldsymbol{v}^{(1)}, \ldots, \boldsymbol{v}^{(L)}\right\} \subset [0,1]^{nm}$ and $\mathcal{S}' = \left\{\boldsymbol{w}^{(1)}, \ldots, \boldsymbol{w}^{(L)}\right\} \subset [0,1]^{nm}$ be two arbitrary sets of valuation vectors such that $\boldsymbol{v}^{(\ell)} - \boldsymbol{\epsilon} \leq \boldsymbol{w}^{(\ell)} \leq \boldsymbol{v}^{(\ell)}$ for all $\ell \in [L]$. Let $\hat{\boldsymbol{r}}'$ be the empirically optimal reserve vector over the set $\mathcal{S}'$ and $\hat{\boldsymbol{r}}$ be the empirically optimal reserve vector over the set $\mathcal{S}$.

For a valuation vector $\boldsymbol{v}^{(\ell)}$, let $\boldsymbol{v}^{(\ell)}(j) \in \mathbb{R}^n$ denote all bidders' bids for item $j$. Given a reserve vector $\boldsymbol{r} = (\boldsymbol{r}_1, \ldots, \boldsymbol{r}_m) \in [0,1]^{nm}$, the revenue obtained from selling item $j$ is not a function of $\boldsymbol{v}^{(\ell)}(j')$ or $\boldsymbol{r}_{j'}$ for any $j' \neq j$, so we can define $\text{rev}_{\boldsymbol{r}_j}\left(\boldsymbol{v}^{(\ell)}(j)\right)$ to be the revenue obtained from selling item $j$ using the reserves $\boldsymbol{r}_j$. Of course, the overall revenue equals the sum of the revenues obtained from selling each item, so $\text{rev}_{\boldsymbol{r}}\left(\boldsymbol{v}^{(\ell)}\right) = \sum_{j=1}^m \text{rev}_{\boldsymbol{r}_j}\left(\boldsymbol{v}^{(\ell)}(j)\right)$. Using this notation, we can write

$$\sum_{\ell=1}^L \text{rev}_{\hat{\boldsymbol{r}}}\left(\boldsymbol{v}^{(\ell)}\right) = \max_{\boldsymbol{r} \in [0,1]^{nm}} \sum_{\ell=1}^L \sum_{j=1}^m \text{rev}_{\boldsymbol{r}_j}\left(\boldsymbol{v}^{(\ell)}(j)\right) = \max_{\boldsymbol{r} \in [0,1]^{nm}} \sum_{j=1}^m \sum_{\ell=1}^L \text{rev}_{\boldsymbol{r}_j}\left(\boldsymbol{v}^{(\ell)}(j)\right). \quad (4)$$

In Lemma B.1, we prove that we can flip the order of the max and the sum in Equation (4):

$$\sum_{\ell=1}^L \text{rev}_{\hat{\boldsymbol{r}}}\left(\boldsymbol{v}^{(\ell)}\right) = \sum_{j=1}^m \max_{\boldsymbol{r}_j \in [0,1]^n} \sum_{\ell=1}^L \text{rev}_{\boldsymbol{r}_j}\left(\boldsymbol{v}^{(\ell)}(j)\right) \quad (5)$$

and

$$\sum_{\ell=1}^L \text{rev}_{\hat{\boldsymbol{r}}'}\left(\boldsymbol{w}^{(\ell)}\right) = \sum_{j=1}^m \max_{\boldsymbol{r}_j \in [0,1]^n} \sum_{\ell=1}^L \text{rev}_{\boldsymbol{r}_j}\left(\boldsymbol{w}^{(\ell)}(j)\right).$$

Fix an arbitrary item $j \in [m]$. Let $\hat{\boldsymbol{q}}_j \in \mathbb{R}^n$ be the vector of non-anonymous reserves that maximizes average revenue over $\boldsymbol{v}^{(1)}(j), \ldots, \boldsymbol{v}^{(L)}(j)$:

$$\hat{\boldsymbol{q}}_j = \text{argmax}_{\boldsymbol{r}_j \in [0,1]^n} \sum_{\ell=1}^L \text{rev}_{\boldsymbol{r}_j}\left(\boldsymbol{v}^{(\ell)}(j)\right).$$

Similarly, let $\hat{q}'_j \in \mathbb{R}^n$ be the vector of non-anonymous reserves that maximizes average revenue over $w^{(1)}(j), \ldots, w^{(L)}(j)$:

$$\hat{q}'_j = \mathrm{argmax}_{r_j \in [0,1]^n} \sum_{\ell=1}^{L} \mathrm{rev}_{r_j}\left(w^{(\ell)}(j)\right).$$

Equation (5) implies that the empirically optimal reserve vector $\hat{r}$ over $\mathcal{S}$ is simply the concatenation of the empirically optimal reserves per item: $\hat{r} = (\hat{q}_1, \ldots, \hat{q}_m)$. Similarly, $\hat{r}' = (\hat{q}'_1, \ldots, \hat{q}'_m)$. From Theorem 3.3, we know that the revenues of $\hat{q}_j$ and $\hat{q}'_j$ are close on average over $\mathcal{S}$. In other words, $\sum_{\ell=1}^{L} \mathrm{rev}_{\hat{q}_j}\left(v^{(\ell)}(j)\right) - \mathrm{rev}_{\hat{q}'_j}\left(v^{(\ell)}(j)\right) \leq 2L\epsilon$. Therefore,

$$\sum_{\ell=1}^{L} \mathrm{rev}_{\hat{r}}\left(v^{(\ell)}\right) - \mathrm{rev}_{\hat{r}'}\left(v^{(\ell)}\right) = \sum_{j=1}^{m} \sum_{\ell=1}^{L} \mathrm{rev}_{\hat{q}_j}\left(v^{(\ell)}(j)\right) - \mathrm{rev}_{\hat{q}'_j}\left(v^{(\ell)}(j)\right) \leq 2Lm\epsilon,$$

so the lemma statement holds. $\qquad\square$

**Proposition 3.8.** *Fix an arbitrary error term $\epsilon$. For any deterministic algorithm $\mathcal{A}$ that takes as input a training set $\mathcal{S} \subseteq \mathbb{R}^{nm}$ and returns a vector of non-anonymous reserves $\mathcal{A}(\mathcal{S}) \in \mathbb{R}^{nm}$, there exists a distribution $\mathcal{D}$ such that with probability 1 over the draw $\mathcal{S} = \left\{v^{(1)}, \ldots, v^{(L)}\right\} \sim \mathcal{D}^L$, $\max_{r \in \mathbb{R}^{nm}} \mathbb{E}\left[\mathrm{rev}_r(v)\right] - \mathbb{E}\left[\mathrm{rev}_{\mathcal{A}(\mathcal{S}')}(v)\right] = \Omega(m\epsilon)$ for some noisy training set $\mathcal{S}' = \left\{w^{(1)}, \ldots, w^{(L)}\right\}$ such that $\left\|v^{(\ell)} - w^{(\ell)}\right\|_\infty \leq \epsilon$ for all $\ell \in [L]$.*

*Proof.* Fix a particular algorithm $\mathcal{A}$ and consider the $\mathcal{A}$'s choice of mechanism for a particular item $i$:

**Case 1** There exists constant $\bar{s} > 3\epsilon$ such that for $\mathcal{S}' = \{\bar{s}\mathbf{1}, \ldots, \bar{s}\mathbf{1}\}$, $\mathcal{A}(\mathcal{S}')_i > \bar{s} - \epsilon$, where $\mathcal{A}(\mathcal{S}')_i$ denotes the mechanism parameter for the $i$th item. We will argue that there exists a distribution $\mathcal{D}$ such that algorithm $\mathcal{A}$ attains zero revenue in the worst case.

Consider buyer distribution $\mathcal{D}$ where each buyer-item distribution is a point mass distribution at $\bar{s} - \epsilon$. Then, with probability 1, $\mathcal{S} \sim \mathcal{D}$ is such that $\mathcal{S} = \{(\bar{s} - \epsilon)\mathbf{1}, \ldots, \{(\bar{s} - \epsilon)\mathbf{1}\}$ by definition. And so, for the $i$th item, since $\mathcal{A}(\mathcal{S}')_i > \bar{s} - \epsilon$:

$$\mathbb{E}_{v \sim \mathcal{D}}[\mathrm{rev}_{\mathcal{A}(\mathcal{S}')}(v)] = 0$$

And the constructed $\mathcal{S}'$ satisfies $|\mathcal{S} - \mathcal{S}'| \leq \epsilon$.

To conclude, it remains to note that since $\mathcal{D}$ is a point mass at the same point, the optimal revenue for the $i$th item is:

$$\max_{M \in \mathcal{M}} \mathbb{E}_{v \sim \mathcal{D}}[\mathrm{rev}_M(v)] = \bar{s} - \epsilon$$

And the statement follows from applying the same reasoning for all $m$ items and summing.

**Case 2** In the other case, $\mathcal{A}$ is such that for all constant $\bar{s} > 3\epsilon$ and $\mathcal{S}' = \{\bar{s}\mathbf{1}, \ldots, \bar{s}\mathbf{1}\}$, $\mathcal{A}(\mathcal{S}')_i \leq \bar{s} - \epsilon$.

To prove the statement, we may construct such a $\mathcal{D}$ by again setting it to be the same distribution as in Case 1 where each buyer-item distribution is a point mass at a particular value $\rho$. If $\mathcal{S} \sim \mathcal{D}$, then $\mathcal{S} = \{\rho\mathbf{1}, \ldots, \rho\mathbf{1}\}$.

Now, set $\mathcal{S}' = \{(\rho - \epsilon)\mathbf{1}, \ldots, (\rho - \epsilon)\mathbf{1}\}$, which satisfies $|\mathcal{S}' - \mathcal{S}| \leq \epsilon$. Then, $\mathcal{A}(\mathcal{S}')_i \leq (\rho - \epsilon) - \epsilon$. Hence, for the $i$th item:

$$\mathbb{E}_{v \sim \mathcal{D}}[\mathrm{rev}_{\mathcal{A}(\hat{S})}(v)] \leq \rho - 2\epsilon$$

As argued before, the optimal revenue attainable for the $i$th item is:

$$\max_{M \in \mathcal{M}} \mathbb{E}_{v \sim \mathcal{D}}[\mathrm{rev}_M(v)] = \rho$$

Summing across all $m$ items yield the result.

$\square$

## B.3 Additional proofs about lottery mechanisms

**Single buyer.** We begin by studying lotteries in the case where there is a single unit-demand buyer with values $(v_{11}, \ldots, v_{1m}) \in [0,1]^m$ for $m$ items. Lotteries are defined by a price $r_0 \in [0,1]$ and a set of probabilities $r_1, \ldots, r_m \in [0,1]$ with $\sum_{j=1}^{m} r_j = 1$. If the buyer chooses to pay the price $r_0$, she will receive one item $J \in [m]$, and $\Pr[J = j] = r_j$. Therefore, her expected utility is $\sum_{j=1}^{m} v_{1j} r_j - r_0$. She will choose to participate in the lottery so long as her expected utility is at least 0. We now prove that this class of mechanisms satisfies our robustness condition (Definition 3.1).

**Lemma B.7.** *The set of lottery mechanisms with a single unit-demand buyer is $(p, q)$-robust with*
$$p(\epsilon, n, m) = \epsilon \text{ and } q(\delta, L, n, m) = O\left( \sqrt{\frac{m \log m}{L}} + \sqrt{\frac{1}{L} \log \frac{1}{\delta}} \right).$$

*Proof.* For any $L \geq 1$, let $\mathcal{S} = \{v^{(1)}, \ldots, v^{(L)}\} \subset [0,1]^m$ and $\mathcal{S}' = \{w^{(1)}, \ldots, w^{(L)}\} \subset [0,1]^m$ be two arbitrary sets of valuation vectors such that $v^{(\ell)} - \epsilon \leq w^{(\ell)} \leq v^{(\ell)}$ for all $\ell \in [L]$. Let $\hat{r} = (\hat{r}_0, \hat{r}_1, \ldots, \hat{r}_m)$ be the empirically optimal parameter vector over the set $\mathcal{S}$ and let $\hat{r}' = (\hat{r}'_0, \hat{r}'_1, \ldots, \hat{r}'_m)$ be the empirically optimal parameter vector over the set $\mathcal{S}'$. We prove that $\frac{1}{L} \sum_{\ell=1}^{L} \mathrm{rev}_{\hat{r}}\left(v^{(\ell)}\right) - \mathrm{rev}_{\hat{r}'}\left(v^{(\ell)}\right) \leq \epsilon$.

This proof relies on two claims. The first states that if we shift the price $\hat{r}_0$ down by $\epsilon$ and evaluate the resulting lottery over $\mathcal{S}'$, little revenue is lost. Again, we use the notation $\langle x \rangle = \max\{x, 0\}$.

**Claim B.8.** *Let $r_\epsilon = (\langle \hat{r}_0 - \epsilon \rangle, \hat{r}_1, \ldots, \hat{r}_n)$. If $\mathrm{rev}_{\hat{r}}\left(v^{(\ell)}\right) > 0$, then $\mathrm{rev}_{r_\epsilon}\left(w^{(\ell)}\right) \geq \mathrm{rev}_{\hat{r}}\left(v^{(\ell)}\right) - \epsilon$.*

*Proof of Claim B.8.* Fix an arbitrary $v^{(\ell)}$ such that $\mathrm{rev}_{\hat{r}}\left(v^{(\ell)}\right) > 0$, which means the buyer's utility must be at least 0: $\sum_{j=1}^{m} v_{1j}^{(\ell)} \hat{r}_j - \hat{r}_0 \geq 0$. As we show, this implies the buyer with values $w^{(\ell)}$ has non-negative utility for the lottery defined by $r_\epsilon$. To see why, first suppose that $\hat{r}_0 < \epsilon$. Since $\mathrm{rev}_{r_\epsilon}\left(w^{(\ell)}\right) \geq 0$, it must be that $\mathrm{rev}_{r_\epsilon}\left(w^{(\ell)}\right) \geq \mathrm{rev}_{\hat{r}}\left(v^{(\ell)}\right) - \epsilon = \hat{r}_0 - \epsilon$. Otherwise, $\hat{r}_0 \geq \epsilon$, so $\langle \hat{r}_0 - \epsilon \rangle = \hat{r}_0 - \epsilon$, which means that

$$0 \leq \sum_{j=1}^{m} v_{1j}^{(\ell)} \hat{r}_j - \hat{r}_0 = \sum_{j=1}^{m} v_{1j}^{(\ell)} \hat{r}_j - \epsilon - (\hat{r}_0 - \epsilon) = \sum_{j=1}^{m} v_{1j}^{(\ell)} \hat{r}_j - \sum_{j=1}^{m} \hat{r}_j \epsilon - (\hat{r}_0 - \epsilon) \qquad \left( \sum_{j=1}^{m} \hat{r}_j = 1 \right)$$

$$= \sum_{j=1}^{m} \left( v_{1j}^{(\ell)} - \epsilon \right) \hat{r}_j - (\hat{r}_0 - \epsilon) \leq \sum_{j=1}^{m} w_{1j}^{(\ell)} \hat{r}_j - (\hat{r}_0 - \epsilon). \qquad \left( v^{(\ell)} - \epsilon \leq w^{(\ell)} \right)$$

As a result, the buyer with valuations $w^{(\ell)}$ will participate in the lottery defined by $r_\epsilon$ and pay a price of $\hat{r}_0 - \epsilon$. Therefore, $\mathrm{rev}_{r_\epsilon}\left(w^{(\ell)}\right) = \hat{r}_0 - \epsilon = \mathrm{rev}_{\hat{r}}\left(v^{(\ell)}\right) - \epsilon$. $\square$

Let $I \subseteq [L]$ be the set of indices $\ell$ such that $\mathrm{rev}_{\hat{r}}\left(v^{(\ell)}\right) > 0$ (for all other indices, $\mathrm{rev}_{\hat{r}}\left(v^{(\ell)}\right) = 0$). By Claim B.8, for all $\ell \in I$, $\mathrm{rev}_{\hat{r}}\left(v^{(\ell)}\right) \leq \mathrm{rev}_{r_\epsilon}\left(w^{(\ell)}\right) + \epsilon$. Therefore,

$$\sum_{\ell=1}^{L} \mathrm{rev}_{\hat{r}}\left(v^{(\ell)}\right) = \sum_{\ell \in I} \mathrm{rev}_{\hat{r}}\left(v^{(\ell)}\right) \leq \sum_{\ell \in I} \mathrm{rev}_{r_\epsilon}\left(w^{(\ell)}\right) + L\epsilon \leq \sum_{\ell=1}^{L} \mathrm{rev}_{r_\epsilon}\left(w^{(\ell)}\right) + L\epsilon. \qquad (6)$$

Since $\hat{r}'$ is the empirically optimal reserve under the perturbed training instances $\mathcal{S}'$ (meaning that $\hat{r}' = \mathrm{argmax}_{r \in [0,1]^{m+1}} \sum_{\ell=1}^{L} \mathrm{rev}_r\left(w^{(\ell)}\right)$), Equation (6) implies that

$$\sum_{\ell=1}^{L} \mathrm{rev}_{\hat{r}}\left(v^{(\ell)}\right) \leq \sum_{\ell=1}^{L} \mathrm{rev}_{\hat{r}'}\left(w^{(\ell)}\right) + L\epsilon. \qquad (7)$$

Next, we prove that for any parameter vector $r$, revenue under the samples $v^{(\ell)}$ will only be higher than revenue under the samples $w^{(\ell)}$, which intuitively makes sense since $w^{(\ell)} \leq v^{(\ell)}$.

**Claim B.9.** *For every $\ell \in [L]$ and any parameter vector $\boldsymbol{r} \in [0,1]^{m+1}$, $\mathrm{rev}_{\boldsymbol{r}} \left( \boldsymbol{w}^{(\ell)} \right) \leq \mathrm{rev}_{\boldsymbol{r}} \left( \boldsymbol{v}^{(\ell)} \right)$.*

*Proof of Claim B.9.* Fix an arbitrary index $\ell \in [L]$ such that $\mathrm{rev}_{\boldsymbol{r}} \left( \boldsymbol{w}^{(\ell)} \right) > 0$ (if $\mathrm{rev}_{\boldsymbol{r}} \left( \boldsymbol{w}^{(\ell)} \right) = 0$, then the claim clearly holds). Since the revenue is non-zero, it must be that the buyer's utility for the lottery is at least 0: $\sum_{j=1}^{m} w_{1j}^{(\ell)} r_j - r_0 \geq 0$. Because $\boldsymbol{v}^{(\ell)} \geq \boldsymbol{w}^{(\ell)}$, it must be that $\sum_{j=1}^{m} v_{1j}^{(\ell)} r_j - r_0 \geq \sum_{j=1}^{m} w_{1j}^{(\ell)} r_j - r_0 \geq 0$, so the buyer with valuations $\boldsymbol{v}^{(\ell)}$ will also participate in the lottery and pay a price of $r_0$. Therefore, $\mathrm{rev}_{\boldsymbol{r}} \left( \boldsymbol{w}^{(\ell)} \right) = \mathrm{rev}_{\boldsymbol{r}} \left( \boldsymbol{v}^{(\ell)} \right) = r_0$. $\qquad\square$

Finally, Equation (7) and Claim B.9 imply that $\sum_{\ell=1}^{L} \mathrm{rev}_{\hat{\boldsymbol{r}}} \left( \boldsymbol{v}^{(\ell)} \right) \leq L\epsilon + \sum_{\ell=1}^{L} \mathrm{rev}_{\hat{\boldsymbol{r}}'} \left( \boldsymbol{v}^{(\ell)} \right).$ $\qquad\square$

### B.3.1 Multiple buyers.

Next, we move on to the case where there are $n$ unit-demand buyers.

**Theorem 3.9.** *The set of lotteries with $n$ unit-demand buyers is p-stable with $p(\epsilon, n, m) = n\epsilon$.*

*Proof.* Let $\mathcal{S} = \left\{ \boldsymbol{v}^{(1)}, \ldots, \boldsymbol{v}^{(L)} \right\} \subset [0,1]^{nm}$ and $\mathcal{S}' = \left\{ \boldsymbol{w}^{(1)}, \ldots, \boldsymbol{w}^{(L)} \right\} \subset [0,1]^{nm}$ be two arbitrary sets of valuation vectors such that $\boldsymbol{v}^{(\ell)} - \boldsymbol{\epsilon} \leq \boldsymbol{w}^{(\ell)} \leq \boldsymbol{v}^{(\ell)}$ for all $\ell \in [L]$. Let $\hat{\boldsymbol{r}}'$ be the empirically optimal reserve vector over the set $\mathcal{S}'$ and $\hat{\boldsymbol{r}}$ be the empirically optimal reserve vector over the set $\mathcal{S}$.

For a valuation vector $\boldsymbol{v}^{(\ell)}$, let $\boldsymbol{v}_i^{(\ell)} \in \mathbb{R}^m$ denote buyer $i$'s values for all $m$ items. Let $\boldsymbol{r} = (\boldsymbol{r}_1, \ldots, \boldsymbol{r}_n) \in \mathbb{R}^{n(m+1)}$ be a lottery parameter vector, where $\boldsymbol{r}_i = (r_{i0}, r_{i1}, \ldots, r_{im})$ is the lottery offered to buyer $i$. The revenue obtained from buyer $i$ is not a function of $\boldsymbol{v}_{i'}^{(\ell)}$ or $\boldsymbol{r}_{i'}$ for any $i' \neq i$, so we can define $\mathrm{rev}_{\boldsymbol{r}_i} \left( \boldsymbol{v}_i^{(\ell)} \right)$ to be the revenue obtained from buyer $i$ under the lottery defined by $\boldsymbol{r}$. Of course, the overall revenue equals the sum of the revenues obtained from each buyer, so $\mathrm{rev}_{\boldsymbol{r}} \left( \boldsymbol{v}^{(\ell)} \right) = \sum_{i=1}^{n} \mathrm{rev}_{\boldsymbol{r}_i} \left( \boldsymbol{v}_i^{(\ell)} \right)$. Using this notation, we can write

$$\sum_{\ell=1}^{L} \mathrm{rev}_{\hat{\boldsymbol{r}}} \left( \boldsymbol{v}^{(\ell)} \right) = \max_{\boldsymbol{r} \in [0,1]^{n(m+1)}} \sum_{\ell=1}^{L} \sum_{i=1}^{n} \mathrm{rev}_{\boldsymbol{r}_i} \left( \boldsymbol{v}_i^{(\ell)} \right) = \max_{\boldsymbol{r} \in [0,1]^{n(m+1)}} \sum_{i=1}^{n} \sum_{\ell=1}^{L} \mathrm{rev}_{\boldsymbol{r}_i} \left( \boldsymbol{v}_i^{(\ell)} \right). \quad (8)$$

In Lemma B.1 in Appendix B, we prove that we can flip the order of the max and the sum in Equation (8):

$$\sum_{\ell=1}^{L} \mathrm{rev}_{\hat{\boldsymbol{r}}} \left( \boldsymbol{v}^{(\ell)} \right) = \sum_{i=1}^{n} \max_{\boldsymbol{r}_i \in [0,1]^{m+1}} \sum_{\ell=1}^{L} \mathrm{rev}_{\boldsymbol{r}_i} \left( \boldsymbol{v}_i^{(\ell)} \right) \quad (9)$$

and

$$\sum_{\ell=1}^{L} \mathrm{rev}_{\hat{\boldsymbol{r}}'} \left( \boldsymbol{w}^{(\ell)} \right) = \sum_{i=1}^{n} \max_{\boldsymbol{r}_i \in [0,1]^{m+1}} \sum_{\ell=1}^{L} \mathrm{rev}_{\boldsymbol{r}_i} \left( \boldsymbol{w}_i^{(\ell)} \right).$$

Fix an arbitrary bidder $i \in [n]$. Let $\hat{\boldsymbol{q}}_i \in \mathbb{R}^{m+1}$ be the lottery parameter vector that maximizes average revenue over $\boldsymbol{v}_i^{(1)}, \ldots, \boldsymbol{v}_i^{(L)}$:

$$\hat{\boldsymbol{q}}_i = \mathrm{argmax}_{\boldsymbol{r}_i \in [0,1]^{m+1}} \sum_{\ell=1}^{L} \mathrm{rev}_{\boldsymbol{r}_i} \left( \boldsymbol{v}_i^{(\ell)} \right).$$

Similarly, let $\hat{\boldsymbol{q}}_i' \in \mathbb{R}^{m+1}$ be the lottery parameter vector that maximizes average revenue over $\boldsymbol{w}_i^{(1)}, \ldots, \boldsymbol{w}_i^{(L)}$:

$$\hat{\boldsymbol{q}}_i' = \mathrm{argmax}_{\boldsymbol{r}_i \in [0,1]^{m+1}} \sum_{\ell=1}^{L} \mathrm{rev}_{\boldsymbol{r}_i} \left( \boldsymbol{w}_i^{(\ell)} \right).$$

Equation (9) implies that the empirically optimal parameter vector $\hat{\boldsymbol{r}}$ over $\mathcal{S}$ is simply the concatenation of the empirically optimal parameter vectors per bidder: $\hat{\boldsymbol{r}} = (\hat{\boldsymbol{q}}_1, \ldots, \hat{\boldsymbol{q}}_n)$. Similarly,

$\hat{\boldsymbol{r}}' = (\hat{\boldsymbol{q}}_1', \dots, \hat{\boldsymbol{q}}_n')$. From Lemma B.7, we know that the revenues of $\hat{\boldsymbol{q}}_i$ and $\hat{\boldsymbol{q}}_i'$ are close on average over $\mathcal{S}$. In other words, $\sum_{\ell=1}^{L} \mathrm{rev}_{\hat{\boldsymbol{q}}_i}\left(\boldsymbol{v}_i^{(\ell)}\right) - \mathrm{rev}_{\hat{\boldsymbol{q}}_i'}\left(\boldsymbol{v}_i^{(\ell)}\right) \leq L\epsilon$. Therefore,

$$\sum_{\ell=1}^{L} \mathrm{rev}_{\hat{\boldsymbol{r}}}\left(\boldsymbol{v}^{(\ell)}\right) - \mathrm{rev}_{\hat{\boldsymbol{r}}'}\left(\boldsymbol{v}^{(\ell)}\right) = \sum_{i=1}^{n} \sum_{\ell=1}^{L} \mathrm{rev}_{\hat{\boldsymbol{q}}_i}\left(\boldsymbol{v}_i^{(\ell)}\right) - \mathrm{rev}_{\hat{\boldsymbol{q}}_i'}\left(\boldsymbol{v}_i^{(\ell)}\right) \leq Ln\epsilon,$$

so the lemma statement holds. $\qquad\square$

**Lemma B.10.** *For any $L \geq 1$ and $\delta \in (0, 1)$, with probability $1 - \delta$ over the draw of a set $\mathcal{S} \sim \mathcal{D}^L$, for every lottery parameter vector $\boldsymbol{r} \in \mathbb{R}^{n(m+1)}$,*

$$\left| \frac{1}{L} \sum_{\boldsymbol{v} \in \mathcal{S}} \mathrm{rev}_{\boldsymbol{r}}(\boldsymbol{v}) - \mathbb{E}_{\boldsymbol{v} \sim \mathcal{D}}[\mathrm{rev}_{\boldsymbol{r}}(\boldsymbol{v})] \right| = O\left( \sqrt{\frac{nm \log(nm)}{L}} + \sqrt{\frac{1}{L} \log \frac{1}{\delta}} \right).$$

*Proof.* Given a valuation vector $\boldsymbol{v} \in \mathbb{R}^{nm}$, let $\mathrm{rev}_{\boldsymbol{v}}(\boldsymbol{r})$ equal revenue as a function of the lottery parameter vector $\boldsymbol{r} \in \mathbb{R}^{n(m+1)}$. We prove this result by relying on the notion of *delineability*, which was introduced by Balcan et al. [6]. A mechanism class is $(d, t)$-*delineable* if:

1. It is parameterized by vectors $\boldsymbol{r} \in \mathbb{R}^d$, and

2. For any valuation vector $\boldsymbol{v}$, there is a set $H$ of $t$ hyperplanes in $\mathbb{R}^d$ such that in any connected component $C$ of $\mathbb{R}^d \setminus H$, the function $\mathrm{rev}_{\boldsymbol{v}}(\boldsymbol{r})$ is linear over $C$.

Balcan et al. [6] prove that for any $(d, t)$-delineable mechanism class, with probability $1 - \delta$ over the draw of a set $\mathcal{S} \sim \mathcal{D}^L$, for every mechanism parameter vector $\boldsymbol{r} \in \mathbb{R}^d$,

$$\left| \frac{1}{L} \sum_{\boldsymbol{v} \in \mathcal{S}} \mathrm{rev}_{\boldsymbol{r}}(\boldsymbol{v}) - \mathbb{E}_{\boldsymbol{v} \sim \mathcal{D}}[\mathrm{rev}_{\boldsymbol{r}}(\boldsymbol{v})] \right| = O\left( \sqrt{\frac{d \log(dt)}{L}} + \sqrt{\frac{1}{L} \log \frac{1}{\delta}} \right). \tag{10}$$

Clearly, in the case of lottery mechanisms, $d = n(m + 1)$. We claim that $t = n$. To see why, for any valuation vector $\boldsymbol{v}$, bidder $i$ will choose to participate in the lottery if and only if $\sum_{j=1}^{m} v_{ij} r_{ij} - r_{i0} \geq 0$, which is a hyperplane in $\mathbb{R}^{n(m+1)}$. On one side of this hyperplane, the bidder will pay $r_{i0}$, and on the other side, he will pay nothing. Let $H$ be the set of $n$ such hyperplanes—one per bidder. In any connected component $C$ of $\mathbb{R}^{n(m+1)} \setminus H$, the function $\mathrm{rev}_{\boldsymbol{v}}(\boldsymbol{r})$ is a linear function of the parameters $r_{10}, \dots, r_{n0}$. Therefore, the class is $(n(m + 1), n)$-delineable, so the lemma statement follows from Equation (10). $\qquad\square$

**Proposition 3.11.** *Fix an arbitrary error term $\epsilon$. For any deterministic algorithm $\mathcal{A}$ that takes as input a training set $\mathcal{S} \subseteq \mathbb{R}^{nm}$ and returns a vector of lottery parameters $\mathcal{A}(\mathcal{S}) \in \mathbb{R}^{n(m+1)}$, there exists a distribution $\mathcal{D}$ such that with probability $1$ over the draw $\mathcal{S} = \{\boldsymbol{v}^{(1)}, \dots, \boldsymbol{v}^{(L)}\} \sim \mathcal{D}^L$, $\max_{\boldsymbol{r} \in \mathbb{R}^{n(m+1)}} \mathbb{E}[\mathrm{rev}_{\boldsymbol{r}}(\boldsymbol{v})] - \mathbb{E}[\mathrm{rev}_{\mathcal{A}(\mathcal{S}')}(\boldsymbol{v})] = \Omega(n\epsilon)$ for some noisy training set $\mathcal{S}' = \{\boldsymbol{w}^{(1)}, \dots, \boldsymbol{w}^{(L)}\}$ such that $\|\boldsymbol{v}^{(\ell)} - \boldsymbol{w}^{(\ell)}\|_\infty \leq \epsilon$ for all $\ell \in [L]$.*

*Proof.* We first prove this guarantee for a single buyer and a single item. These mechanisms are defined by a single price $r_{10}$ and probability $r_{11} \in [0, 1]$. We will work with a simple distribution where with probability $1$, $v_{11} = \rho$ for some $\rho \geq 0$ ($v_{11}$ is the buyer's value for the item).

Under this distribution, the revenue-maximizing lottery sets $r_{10} = \rho$ and $r_{11} = 1$. This is because the buyer will choose to participate in the lottery so long as their expected utility $v_{11} r_{11} = \rho r_{11}$ is at least the price $r_{10}$. Therefore, the optimal revenue is $\max\{r_{10} : \rho r_{11} \geq r_{10}\}$, which is maximized when $r_{11} = 1$ and $r_{10} = \rho$.

We now split our analysis into two cases.

**Case 1:** In the first case, there exists a constant $\bar{s} > 2\epsilon$ such that when $\mathcal{A}$ receives the training set $\mathcal{S}' = \{\bar{s}, \dots, \bar{s}\}$, it returns a lottery $\mathcal{A}(\mathcal{S}') = (r_{10}, r_{11})$ with $(\bar{s} - \epsilon) r_{11} < r_{10}$. In this case, we define $\rho = \bar{s} - \epsilon$. Then the lottery that $\mathcal{A}$ returns has revenue $0$, but the optimal lottery has revenue $\bar{s} - \epsilon > \epsilon$, so the proposition holds.

**Case 2:** Otherwise, in the second case, for every constant $\bar{s} > 2\epsilon$, when $\mathcal{A}$ receives the training set $\mathcal{S}' = \{\bar{s}, \ldots, \bar{s}\}$, it returns a lottery $\mathcal{A}(\mathcal{S}') = (r_{10}, r_{11})$ with $(\bar{s} - \epsilon) r_{11} \geq r_{10}$. In this case, we define $\rho = \bar{s}$. The optimal lottery has revenue $\bar{s}$ and the revenue of the lottery that $\mathcal{A}$ returns is at most $r_{10} \leq \bar{s} - \epsilon$, so the proposition holds. $\qquad\square$

## B.4 Additional proofs about item-price mechanisms with unit-demand buyers under dispersion

**Lemma B.11.** *Suppose there are two items for sale and a single unit-demand buyer with values $\boldsymbol{v} = (v_{11}, v_{12}) \in [0,1]^2$. For any $\epsilon > 0$, there exists a vector $\boldsymbol{w} \in [0,1]^2$ such that $\boldsymbol{v} - \boldsymbol{\epsilon} \leq \boldsymbol{w} \leq \boldsymbol{v}$ and $\max_{\boldsymbol{r}} \mathrm{rev}_{\boldsymbol{r}}(\boldsymbol{v}) - \mathrm{rev}_{\hat{\boldsymbol{r}}'}(\boldsymbol{v}) > |v_{11} - v_{12}|$, where $\hat{\boldsymbol{r}}' = \mathrm{argmax}_{\boldsymbol{r}} \mathrm{rev}_{\boldsymbol{r}}(\boldsymbol{w})$.*

Since $|v_{11} - v_{12}|$ can be as large as 1, Lemma B.11 implies that we are not able to obtain a non-trivial bound on the stability function $p$.

*Proof of Lemma B.11.* Without loss of generality, suppose that $v_{11} \geq v_{12} > 0$. Also, suppose that we break ties in favor of item 1: if $v_{11} - r_{11} = v_{12} - r_{12} \geq 0$, then the buyer will buy item 1. In this case, the optimal set of prices are $\hat{\boldsymbol{r}} = (v_{11}, v_{12})$ and $\mathrm{rev}_{\hat{\boldsymbol{r}}}(\boldsymbol{v}) = v_{11}$. Let $\boldsymbol{w} = (v_{11}, \langle v_{12} - \epsilon \rangle)$. In this case, $(v_{11}, \langle v_{12} - \epsilon \rangle) = \mathrm{argmax}_{\boldsymbol{r}} \mathrm{rev}_{\boldsymbol{r}}(\boldsymbol{w}) = \hat{\boldsymbol{r}}'$. Under this set of prices, the buyer with the values $\boldsymbol{v}$ will choose to buy item 2, so $\mathrm{rev}_{\hat{\boldsymbol{r}}'}(\boldsymbol{v}) < v_{12}$. Therefore, the lemma statement holds. $\qquad\square$

Before proving Theorem 3.13, we begin by restating Definition 3.12 more formally using mathematical notation.

**Definition B.12** (($\epsilon, k$)-dispersion [5])**.** Let $\boldsymbol{v}^{(1)}, \ldots, \boldsymbol{v}^{(L)}$ be $L$ valuation vectors. We say this set of vectors is ($\epsilon, k$)-dispersed if for any price vector $\boldsymbol{r} \in \mathbb{R}^m$, there are at most $k$ samples $\boldsymbol{v}^{(\ell)}$ such that for some buyer $i \in [n]$, either:

1. For some pair of items $j, j' \in [m]$, buyer $i$'s utility for item $j$ is within $\epsilon$ of her utility for item $j'$, i.e.,
$$\left| v_{i,j}^{(\ell)} - r_j - \left( v_{i,j'}^{(\ell)} - r_{j'} \right) \right| \leq \epsilon,$$
or

2. Buyer $i$'s utility for some item $j$ is between $0$ and $\epsilon$, i.e., $0 \leq v_{i,j}^{(\ell)} \leq \epsilon$.

We begin with the following helpful lemma which allows us to prove Theorem 3.13.

**Lemma B.13.** *Let $\mathcal{S} = \left\{ \boldsymbol{v}^{(1)}, \ldots, \boldsymbol{v}^{(L)} \right\}$ and $\mathcal{S}' = \left\{ \boldsymbol{w}^{(1)}, \ldots, \boldsymbol{w}^{(L)} \right\}$ be two sets of valuation vectors such that for all $\ell \in [L]$, $\left\| \boldsymbol{v}^{(\ell)} - \boldsymbol{w}^{(\ell)} \right\|_\infty \leq \epsilon$ and all $\boldsymbol{r} \in \mathbb{R}^m$, $\mathrm{rev}_{\boldsymbol{r}} \left( \boldsymbol{v}^{(\ell)} \right)$ and $\mathrm{rev}_{\boldsymbol{r}} \left( \boldsymbol{w}^{(\ell)} \right)$ are contained in the interval $[0,1]$. Suppose that the set $\mathcal{S}$ is $(2\epsilon, k)$-dispersed. Then for every parameter vector $\boldsymbol{r} \in \mathbb{R}^m$, $\sum_{\ell=1}^{L} \left| \mathrm{rev}_{\boldsymbol{r}} \left( \boldsymbol{v}^{(\ell)} \right) - \mathrm{rev}_{\boldsymbol{r}} \left( \boldsymbol{w}^{(\ell)} \right) \right| \leq k$.*

*Proof.* For each valuation vector $\boldsymbol{v}^{(\ell)} \in \mathcal{S}$, every pair of items $j, j' \in [m]$, and every bidder $i \in [n]$, let $h_{i,j,j'}^{(\ell)} : \mathbb{R}^m \to \{0,1\}$ be a hyperplane indicator function such that

$$h_{i,j,j'}^{(\ell)}(\boldsymbol{r}) = \begin{cases} 1 & \text{if } v_{i,j}^{(\ell)} - r_j \geq v_{i,j'}^{(\ell)} - r_{j'} \\ 0 & \text{otherwise.} \end{cases}$$

If $h_{i,j,j'}^{(\ell)}(\boldsymbol{r}) = 1$, then buyer $i$ with valuations defined by the vector $\boldsymbol{v}^{(\ell)}$ prefers item $j$ to $j'$ when presented with the prices $\boldsymbol{r}$. Otherwise, when $h_{i,j,j'}^{(\ell)}(\boldsymbol{r}) = 0$, buyer $i$ prefers item $j'$ to $j$. Similarly, for the set $\mathcal{S}'$, we define the same set of hyperplane indicator functions, which we denote as $\tilde{h}_{i,j,j'}^{(\ell)}$:

$$\tilde{h}_{i,j,j'}^{(\ell)}(\boldsymbol{r}) = \begin{cases} 1 & \text{if } w_{i,j}^{(\ell)} - r_j \geq w_{i,j'}^{(\ell)} - r_{j'} \\ 0 & \text{otherwise.} \end{cases}$$

Fix an arbitrary price vector $\boldsymbol{r} \in \mathbb{R}^m$. For any index $\ell \in [L]$, suppose that for all buyers $i \in [n]$ and item pairs $j, j' \in [m]$, $h_{i,j,j'}^{(\ell)}(\boldsymbol{r}) = \tilde{h}_{i,j,j'}^{(\ell)}(\boldsymbol{r})$. Then every buyer's preference ordering over items is

the same under the valuations $\boldsymbol{v}^{(\ell)}$ as it is under the valuations $\boldsymbol{w}^{(\ell)}$, so the items each buyer buys will be the same under both valuation vectors. Therefore, $\text{rev}_{\boldsymbol{r}}\left(\boldsymbol{v}^{(\ell)}\right) = \text{rev}_{\boldsymbol{r}}\left(\boldsymbol{w}^{(\ell)}\right)$.

We claim that due to dispersion, for at most $k$ of the indices $\ell \in [L]$, $h_{i,j,j'}^{(\ell)}(\boldsymbol{r}) \neq \tilde{h}_{i,j,j'}^{(\ell)}(\boldsymbol{r})$ for some buyer $i \in [n]$ and item pair $j, j' \in [m]$. To see why, suppose $h_{i,j,j'}^{(\ell)}(\boldsymbol{r}) \neq \tilde{h}_{i,j,j'}^{(\ell)}(\boldsymbol{r})$ for some $\ell \in [n]$ and $j, j' \in [m]$. There are two cases, either $h_{i,j,j'}^{(\ell)}(\boldsymbol{r}) = 0$ or $h_{i,j,j'}^{(\ell)}(\boldsymbol{r}) = 1$:

1. In the first case, $h_{i,j,j'}^{(\ell)}(\boldsymbol{r}) = 0$, so $v_{i,j}^{(\ell)} - r_j < v_{i,j'}^{(\ell)} - r_{j'}$. Since $h_{i,j,j'}^{(\ell)}(\boldsymbol{r}) \neq \tilde{h}_{i,j,j'}^{(\ell)}(\boldsymbol{r})$, it must be that $w_{i,j}^{(\ell)} - r_j \geq w_{i,j'}^{(\ell)} - r_{j'}$. Therefore,

$$v_{i,j'}^{(\ell)} - r_{j'} \leq w_{i,j'}^{(\ell)} + \epsilon - r_{j'} \leq w_{i,j}^{(\ell)} + \epsilon - r_j \leq v_{i,j}^{(\ell)} + 2\epsilon - r_j < v_{i,j'}^{(\ell)} + 2\epsilon - r_{j'}.$$

Rearranging terms, we have that

$$r_j - r_{j'} - 2\epsilon \leq v_{i,j}^{(\ell)} - v_{i,j'}^{(\ell)} < r_j - r_{j'}. \tag{11}$$

2. In the second case, $h_{i,j,j'}^{(\ell)}(\boldsymbol{r}) = 1$, so $v_{i,j}^{(\ell)} - r_j \geq v_{i,j'}^{(\ell)} - r_{j'}$. Since $h_{i,j,j'}^{(\ell)}(\boldsymbol{r}) \neq \tilde{h}_{i,j,j'}^{(\ell)}(\boldsymbol{r})$, it must be that $w_{i,j}^{(\ell)} - r_j < w_{i,j'}^{(\ell)} - r_{j'}$. Therefore,

$$v_{i,j}^{(\ell)} - r_j \leq w_{i,j}^{(\ell)} + \epsilon - r_j < w_{i,j'}^{(\ell)} + \epsilon - r_{j'} \leq v_{i,j'}^{(\ell)} + 2\epsilon - r_{j'} < v_{i,j}^{(\ell)} + 2\epsilon - r_j.$$

Rearranging terms, we have that

$$r_j - r_{j'} \leq v_{i,j}^{(\ell)} - v_{i,j'}^{(\ell')} < r_j - r_{j'} + 2\epsilon. \tag{12}$$

From the fact that the set $\mathcal{S}$ is $(2\epsilon, k)$-dispersed, we know that Equations (11) and (12) can hold for at most $k$ of the valuation vectors $\boldsymbol{v}^{(\ell)} \in \mathcal{S}$. Therefore, for at most $k$ indices $\ell \in [L]$, $\text{rev}_{\boldsymbol{r}}\left(\boldsymbol{v}^{(\ell)}\right) \neq \text{rev}_{\boldsymbol{r}}\left(\boldsymbol{w}^{(\ell)}\right)$. For these indices $\ell \in [L]$, $\left|\text{rev}_{\boldsymbol{r}}\left(\boldsymbol{v}^{(\ell)}\right) - \text{rev}_{\boldsymbol{r}}\left(\boldsymbol{w}^{(\ell)}\right)\right| \leq 1$ because $\text{rev}_{\boldsymbol{r}}\left(\boldsymbol{v}^{(\ell)}\right), \text{rev}_{\boldsymbol{r}}\left(\boldsymbol{w}^{(\ell)}\right) \in [0, 1]$. Therefore, the lemma statement holds. $\square$

**Theorem 3.13.** *Let* $\mathcal{S} = \left\{\boldsymbol{v}^{(1)}, \ldots, \boldsymbol{v}^{(L)}\right\} \sim \mathcal{D}^L$ *be a set of* $(2\epsilon, k)$-*dispersed vectors. Let* $\mathcal{S}' = \left\{\boldsymbol{w}^{(1)}, \ldots, \boldsymbol{w}^{(L)}\right\} \subset \mathbb{R}_{\geq 0}^{nm}$ *be another set such that for all* $\ell \in [L]$, $\left\|\boldsymbol{v}^{(\ell)} - \boldsymbol{w}^{(\ell)}\right\|_{\infty} \leq \epsilon$. *Let* $\hat{\boldsymbol{r}}'$ *be empirically optimal over* $\mathcal{S}'$: $\hat{\boldsymbol{r}}' = \text{argmax}_{\boldsymbol{r} \in \mathbb{R}^{nm}} \sum_{\ell=1}^{L} \text{rev}_{\boldsymbol{r}}\left(\boldsymbol{w}^{(\ell)}\right)$. *With probability* $1 - \delta$ *over the draw of* $\mathcal{S}$, $\max_{\boldsymbol{r} \in \mathbb{R}^{nm}} \mathbb{E}_{\boldsymbol{v} \sim \mathcal{D}}\left[\text{rev}_{\boldsymbol{r}}(\boldsymbol{v})\right] - \mathbb{E}\left[\text{rev}_{\hat{\boldsymbol{r}}'}(\boldsymbol{v})\right] = O\left(\frac{k}{L} + \sqrt{\frac{1}{L}\left(nm\log(nm) + \log\frac{1}{\delta}\right)}\right)$.

*Proof.* From prior research by Morgenstern and Roughgarden [46], we know that with probability $1 - \delta$, for all price vectors $\boldsymbol{r} \in \mathbb{R}^{nm}$, the average revenue of $\boldsymbol{r}$ over $\mathcal{S}$ is close to its expected revenue:

$$\left|\frac{1}{L}\sum_{\ell=1}^{L}\text{rev}_{\boldsymbol{r}}\left(\boldsymbol{v}^{(\ell)}\right) - \mathbb{E}_{\boldsymbol{v} \sim \mathcal{D}}\left[\text{rev}_{\boldsymbol{r}}(\boldsymbol{v})\right]\right| = O\left(\sqrt{\frac{1}{L}\left(nm\log(nm) + \log\frac{1}{\delta}\right)}\right). \tag{13}$$

Let $\boldsymbol{r}^* = \operatorname{argmax}_{\boldsymbol{r} \in \mathbb{R}^m} \mathbb{E}_{\boldsymbol{v} \sim \mathcal{D}} \left[ \operatorname{rev}_{\boldsymbol{r}}(\boldsymbol{v}) \right]$. Then

$$\max_{\boldsymbol{r} \in \mathbb{R}^m} \mathbb{E}_{\boldsymbol{v} \sim \mathcal{D}} \left[ \operatorname{rev}_{\boldsymbol{r}}(\boldsymbol{v}) \right] - \mathbb{E}_{\boldsymbol{v} \sim \mathcal{D}} \left[ \operatorname{rev}_{\hat{\boldsymbol{r}}'}(\boldsymbol{v}) \right]$$

$$= \mathbb{E}_{\boldsymbol{v} \sim \mathcal{D}} \left[ \operatorname{rev}_{\boldsymbol{r}^*}(\boldsymbol{v}) \right] - \frac{1}{L} \sum_{\ell=1}^{L} \operatorname{rev}_{\boldsymbol{r}^*}\left( \boldsymbol{v}^{(\ell)} \right) + \frac{1}{L} \sum_{\ell=1}^{L} \operatorname{rev}_{\boldsymbol{r}^*}\left( \boldsymbol{v}^{(\ell)} \right) - \mathbb{E}_{\boldsymbol{v} \sim \mathcal{D}} \left[ \operatorname{rev}_{\hat{\boldsymbol{r}}'}(\boldsymbol{v}) \right]$$

$$\leq \frac{1}{L} \sum_{\ell=1}^{L} \operatorname{rev}_{\boldsymbol{r}^*}\left( \boldsymbol{v}^{(\ell)} \right) - \mathbb{E}_{\boldsymbol{v} \sim \mathcal{D}} \left[ \operatorname{rev}_{\hat{\boldsymbol{r}}'}(\boldsymbol{v}) \right] + O\left( \sqrt{\frac{1}{L}\left( nm \log(nm) + \log \frac{1}{\delta} \right)} \right) \qquad (14)$$

$$\leq \frac{1}{L} \sum_{\ell=1}^{L} \operatorname{rev}_{\boldsymbol{r}^*}\left( \boldsymbol{w}^{(\ell)} \right) + \frac{k}{L} - \mathbb{E}_{\boldsymbol{v} \sim \mathcal{D}} \left[ \operatorname{rev}_{\hat{\boldsymbol{r}}'}(\boldsymbol{v}) \right] + O\left( \sqrt{\frac{1}{L}\left( nm \log(nm) + \log \frac{1}{\delta} \right)} \right) \qquad (15)$$

$$\leq \frac{1}{L} \sum_{\ell=1}^{L} \operatorname{rev}_{\hat{\boldsymbol{r}}'}\left( \boldsymbol{w}^{(\ell)} \right) + \frac{k}{L} - \mathbb{E}_{\boldsymbol{v} \sim \mathcal{D}} \left[ \operatorname{rev}_{\hat{\boldsymbol{r}}'}(\boldsymbol{v}) \right] + O\left( \sqrt{\frac{1}{L}\left( nm \log(nm) + \log \frac{1}{\delta} \right)} \right) \qquad (16)$$

$$\leq \frac{1}{L} \sum_{\ell=1}^{L} \operatorname{rev}_{\hat{\boldsymbol{r}}'}\left( \boldsymbol{v}^{(\ell)} \right) + \frac{2k}{L} - \mathbb{E}_{\boldsymbol{v} \sim \mathcal{D}} \left[ \operatorname{rev}_{\hat{\boldsymbol{r}}'}(\boldsymbol{v}) \right] + O\left( \sqrt{\frac{1}{L}\left( nm \log(nm) + \log \frac{1}{\delta} \right)} \right) \qquad (17)$$

$$\leq \frac{2k}{L} + O\left( \sqrt{\frac{1}{L}\left( nm \log(nm) + \log \frac{1}{\delta} \right)} \right), \qquad (18)$$

where Equations (14) and (18) follow from Equation (13), Equations (15) and (17) follow from Lemma B.13, and Equation (16) follows from the fact that $\hat{\boldsymbol{r}}' = \operatorname{argmax} \sum_{\ell=1}^{L} \operatorname{rev}_{\boldsymbol{r}}\left( \boldsymbol{w}^{(\ell)} \right)$. $\qquad \square$

When does dispersion hold? One example, also observed in prior research [5, 7, 34], is when the distribution over buyers' values is relatively "smooth." More formally, for any distribution over $[0,1]$ with probability density function $f : [0,1] \to \mathbb{R}_{\geq 0}$, we say the density function is $\kappa$-*bounded* if $\max_{x \in [0,1]} f(x) \leq \kappa$.

**Lemma 3.15.** *Suppose that for every buyer $i \in [n]$ and every pair of items $j, j' \in [m]$, buyer $i$'s values for items $j$ and $j'$ have a $\kappa$-bounded joint density function. Then for any $\epsilon > 0$, with probability $1 - \delta$ over the draw $\mathcal{S} = \left\{ \boldsymbol{v}^{(1)}, \ldots, \boldsymbol{v}^{(L)} \right\} \sim \mathcal{D}^L$, the set $\mathcal{S}$ is $(2\epsilon, k)$-dispersed with $k = 4Lnm^2 \kappa \epsilon + O\left( nm^2 \sqrt{L \log \frac{nm}{\delta}} \right)$.*

*Proof.* Fix a buyer $i \in [n]$ and pair of items $j, j' \in [m]$. For any price vector $\boldsymbol{r} \in \mathbb{R}^m$, we want to bound the number of samples $\boldsymbol{v}^{(\ell)}$ such that buyer $i$'s utility for item $j$ is within $2\epsilon$ of her utility for item $j'$: $\left| v_{i,j}^{(\ell)} - r_j - \left( v_{i,j'}^{(\ell)} - r_{j'} \right) \right| \leq 2\epsilon$. Since buyer $i$'s values for items $j$ and $j'$ have a $\kappa$-bounded joint density function, we know that $\Pr\left[ \left| v_{i,j}^{(\ell)} - r_j - \left( v_{i,j'}^{(\ell)} - r_{j'} \right) \right| \leq 2\epsilon \right] \leq 4\kappa\epsilon$. Therefore,

$$\mathbb{E}\left[ \left| \left\{ \ell : \left| v_{i,j}^{(\ell)} - r_j - \left( v_{i,j'}^{(\ell)} - r_{j'} \right) \right| \leq 2\epsilon \right\} \right| \right] \leq 4N\kappa\epsilon.$$

This is because since $v_{i,j}$ and $v_{i,j'}$ are both random variables in $[0,1]$ with a $\kappa$-bounded joint density function, the difference $v_{i,j} - v_{i,j'}$ also has a $\kappa$-bounded density function [5].

Next, we can write

$$\mathbb{E}\left[ \left| \left\{ \ell : \left| v_{i,j}^{(\ell)} - r_j - \left( v_{i,j'}^{(\ell)} - r_{j'} \right) \right| \leq 2\epsilon \right\} \right| \right] = \mathbb{E}\left[ \sum_{\ell=1}^{L} \mathbf{1}_{\left\{ \left| v_{i,j}^{(\ell)} - r_j - \left( v_{i,j'}^{(\ell)} - r_{j'} \right) \right| \leq 2\epsilon \right\}} \right].$$

Since the VC dimension on intervals is 2, we know that with probability $1 - \frac{\delta}{nm^2}$, for any price vector $\boldsymbol{r} \in \mathbb{R}^m$,

$$\left|\left\{\ell : \left|v_{i,j}^{(\ell)} - r_j - \left(v_{i,j'}^{(\ell)} - r_{j'}\right)\right| \leq 2\epsilon\right\}\right| = \sum_{\ell=1}^{L} \mathbf{1}_{\left\{\left|v_{i,j}^{(\ell)} - r_j - \left(v_{i,j'}^{(\ell)} - r_{j'}\right)\right| \leq 2\epsilon\right\}}$$

$$\leq \mathbb{E}\left[\sum_{\ell=1}^{L} \mathbf{1}_{\left\{\left|v_{i,j}^{(\ell)} - r_j - \left(v_{i,j'}^{(\ell)} - r_{j'}\right)\right| \leq 2\epsilon\right\}}\right] + O\left(\sqrt{L \log \frac{nm}{\delta}}\right)$$

$$\leq 4L\kappa\epsilon + O\left(\sqrt{L \log \frac{nm}{\delta}}\right). \tag{19}$$

Applying a union bound, we have show that with probability $1 - \delta$, for any price vector $\boldsymbol{r} \in \mathbb{R}^m$, any buyer $i \in [n]$, and any pair of items $j, j' \in [m]$ there are at most $4L\kappa\epsilon + O\left(\sqrt{L \log \frac{nm}{\delta}}\right)$ samples $\boldsymbol{v}^{(\ell)}$ such that buyer $i$'s utility for item $j$ is within $2\epsilon$ of her utility for item $j'$, i.e.:

$$\left|\left\{\ell : \left|v_{i,j}^{(\ell)} - r_j - \left(v_{i,j'}^{(\ell)} - r_{j'}\right)\right| \leq 2\epsilon\right\}\right| \leq 4L\kappa\epsilon + O\left(\sqrt{L \log \frac{nm}{\delta}}\right).$$

If we union these sets over all $n$ buyers and all $m^2$ pairs of items, we have that with probability $1 - \delta$, for all price vector $\boldsymbol{r} \in \mathbb{R}^m$,

$$\left|\left\{\ell : \left|v_{i,j}^{(\ell)} - r_j - \left(v_{i,j'}^{(\ell)} - r_{j'}\right)\right| \leq \epsilon \text{ for some } i \in [n] \text{ and } j, j' \in [m]\right\}\right| = \tilde{O}\left(Lnm^2\kappa\epsilon\right). \tag{20}$$

$\square$

**Corollary B.14.** *Suppose that for every buyer $i \in [n]$ and every pair of items $j, j' \in [m]$, buyer $i$'s values for items $j$ and $j'$ have a $\kappa$-bounded joint density function. For any $\delta \in (0, 1)$, let $\mathcal{S} = \left\{\boldsymbol{v}^{(1)}, \ldots, \boldsymbol{v}^{(L)}\right\} \sim \mathcal{D}^L$ be a set of $L$ valuation vectors and let $\mathcal{S}' = \left\{\boldsymbol{w}^{(1)}, \ldots, \boldsymbol{w}^{(L)}\right\}$ be another set of vectors such that for all $\ell \in [L]$, $\left\|\boldsymbol{v}^{(\ell)} - \boldsymbol{w}^{(\ell)}\right\|_\infty \leq \epsilon$. Finally, let $\hat{\boldsymbol{r}} = \operatorname{argmax}_{\boldsymbol{r} \in \mathbb{R}^m} \sum_{\ell=1}^{L} \operatorname{rev}_{\boldsymbol{r}}\left(\boldsymbol{w}^{(\ell)}\right)$. With probability $1 - \delta$,*

$$\max_{\boldsymbol{r} \in \mathbb{R}^m} \mathbb{E}_{\boldsymbol{v} \sim \mathcal{D}}\left[\operatorname{rev}_{\boldsymbol{r}}(\boldsymbol{v})\right] - \mathbb{E}_{\boldsymbol{v} \sim \mathcal{D}}\left[\operatorname{rev}_{\hat{\boldsymbol{r}}}(\boldsymbol{v})\right] \leq O\left(nm^2\kappa\epsilon + nm^2\sqrt{\frac{1}{L} \log \frac{nm}{\delta}}\right).$$

## C  Additional results about multi-task mechanism design (Section 4)

Our multi-task learning approach is inspired by the following two observations, Observations 1 and 2.
*Observation* 1 (known vectors $\boldsymbol{b}^{(t)}$ and $\boldsymbol{z}^{(\ell,t)}$). Suppose that rather than only being able to observe the valuation vectors $\boldsymbol{v}^{(\ell,t)}$, we were also able to observe the vectors $\boldsymbol{b}^{(t)}$ and $\boldsymbol{z}^{(\ell,t)}$, where $\boldsymbol{v}^{(\ell,t)} = \boldsymbol{b}^{(t)} + \boldsymbol{z}^{(\ell,t)}$. This would allow us to generate a total of $LT$ samples from each distribution $\mathcal{D}^{(t)}$, namely, $\left\{\boldsymbol{b}^{(t)} + \boldsymbol{z}^{(\ell,\tau)} : \ell \in [L], \tau \in [T]\right\}$. If the mechanism class we optimize over is $(p, q)$-robust, we would be able to provide strong learnability guarantees, as we summarize below.
*Theorem* C.1. *Fix a $(p, q)$-robust mechanism class. With probability $1 - \delta$ over the draw of $LT$ vectors $\left\{\boldsymbol{z}^{(\ell,\tau)} : \ell \in [L], \tau \in [T]\right\} \sim \mathcal{D}^{LT}$, for every task $t \in [T]$ and every parameter vector $\boldsymbol{r}$,*

$$\left|\frac{1}{LT} \sum_{\tau=1}^{T} \sum_{\ell=1}^{L} \operatorname{rev}_{\boldsymbol{r}}\left(\boldsymbol{b}^{(t)} + \boldsymbol{z}^{(\ell,\tau)}\right) - \mathbb{E}_{\boldsymbol{v} \sim \mathcal{D}^{(t)}}\left[\operatorname{rev}_{\boldsymbol{r}}(\boldsymbol{v})\right]\right| \leq q\left(\frac{\delta}{T}, LT, n, m\right).$$

Before proving Theorem C.1, we highlight the following implication: if $\boldsymbol{r}^{(t)} \in \mathbb{R}^d$ maximizes average revenue over the $LT$ samples from $\mathcal{D}^{(t)}$, then the expected revenue of $\boldsymbol{r}^{(t)}$ over $\mathcal{D}^{(t)}$ is nearly optimal. Formally, if $\boldsymbol{r}^{(t)} = \operatorname{argmax}_{\boldsymbol{r} \in \mathbb{R}^d} \sum_{\tau=1}^{T} \sum_{\ell=1}^{L} \operatorname{rev}_{\boldsymbol{r}}\left(\boldsymbol{b}^{(t)} + \boldsymbol{z}^{(\ell,\tau)}\right)$, then with probability $1 - \delta$, for every task $t$, $\max_{\boldsymbol{r} \in \mathbb{R}^d} \mathbb{E}_{\boldsymbol{v} \sim \mathcal{D}^{(t)}}\left[\operatorname{rev}_{\boldsymbol{r}}(\boldsymbol{v})\right] - \mathbb{E}_{\boldsymbol{v} \sim \mathcal{D}^{(t)}}\left[\operatorname{rev}_{\boldsymbol{r}^{(t)}}(\boldsymbol{v})\right] \leq 2 \cdot q\left(\frac{\delta}{T}, LT, n, m\right)$. For example, in the case of multi-item non-anonymous second-price auctions, Corollary 3.7 implies that

$$\max_{\boldsymbol{r} \in \mathbb{R}^d} \mathbb{E}_{\boldsymbol{v} \sim \mathcal{D}^{(t)}}\left[\operatorname{rev}_{\boldsymbol{r}}(\boldsymbol{v})\right] - \mathbb{E}_{\boldsymbol{v} \sim \mathcal{D}^{(t)}}\left[\operatorname{rev}_{\boldsymbol{r}^{(t)}}(\boldsymbol{v})\right] = O\left(\sqrt{\frac{nm \log(nm)}{LT}} + \sqrt{\frac{1}{LT} \log \frac{T}{\delta}}\right). \tag{21}$$

Meanwhile, if we did not bootstrap the $L(T-1)$ additional samples per task and only used the $L$ samples $\boldsymbol{v}^{(1,t)}, \ldots, \boldsymbol{v}^{(L,t)}$ to select a reserve vector $\boldsymbol{r}^{(t)}$, then $T$ would not appear in the denominator of Equation (21), so it would be worse by a multiplicative factor of $\sqrt{T}$. (The same would be true in the case of lottery mechanisms, as is evident from our bound on $q$ from Theorem 3.9.)

*Proof of Theorem C.1.* Fix a task $t$. By definition, $\left\{\boldsymbol{b}^{(t)} + \boldsymbol{z}^{(\ell,\tau)} : \ell \in [L], \tau \in [T]\right\}$ are samples from the distribution $\mathcal{D}^{(t)}$. Since the mechanism class is $(p,q)$-robust, we know that with probability $1 - \frac{\delta}{T}$ over the draw of the vectors $\left\{\boldsymbol{z}^{(\ell,\tau)} : \ell \in [L], \tau \in [T]\right\} \sim \mathcal{D}^{LT}$, for every parameter vector $\boldsymbol{r}$,

$$\left| \frac{1}{LT} \sum_{\tau=1}^{T} \sum_{\ell=1}^{L} \mathrm{rev}_{\boldsymbol{r}}\left(\boldsymbol{b}^{(t)} + \boldsymbol{z}^{(\ell,\tau)}\right) - \underset{\boldsymbol{v} \sim \mathcal{D}^{(t)}}{\mathbb{E}}\left[\mathrm{rev}_{\boldsymbol{r}}(\boldsymbol{v})\right] \right| \leq q\left(\frac{\delta}{T}, LT, n, m\right).$$

The theorem statement therefore holds by a union bound over the $T$ tasks. $\qquad\square$

*Observation* 2 (known vectors $\boldsymbol{b}^{(t)} - \boldsymbol{b}^{(\tau)}$)). Observation 1 applied to the hypothetical scenario where we knew all vectors $\boldsymbol{b}^{(t)}$ and $\boldsymbol{z}^{(\ell,t)}$. Instead, suppose we knew only the differences $\boldsymbol{b}^{(t)} - \boldsymbol{b}^{(\tau)}$ for every pair of tasks $t, \tau \in [T]$, along with the true training set $\left\{\boldsymbol{v}^{(\ell,t)} : \ell \in [L], t \in [T]\right\}$. In this case, we could recover the vectors $\boldsymbol{b}^{(t)} + \boldsymbol{z}^{(\ell,\tau)}$ for every index $\ell \in [L]$ and pair of tasks $t, \tau \in [T]$, which is all we need to apply Theorem C.1. After all,

$$\boldsymbol{b}^{(t)} + \boldsymbol{z}^{(\ell,\tau)} = \boldsymbol{b}^{(t)} - \boldsymbol{b}^{(\tau)} + \boldsymbol{b}^{(\tau)} + \boldsymbol{z}^{(\ell,\tau)} = \boldsymbol{b}^{(t)} - \boldsymbol{b}^{(\tau)} + \boldsymbol{v}^{(\ell,\tau)}. \tag{22}$$

In reality, we do not know the differences $\boldsymbol{b}^{(t)} - \boldsymbol{b}^{(\tau)}$, but we can estimate them from data. We then use these estimates to bootstrap training instances as in Observation 2. We may not be able to estimate these differences exactly—there may be additive noise in our estimates—but from our results from the previous section (in particular, Fact 3.2), we know that this noise is not problematic in our final revenue guarantees so long as its magnitude is not too large.

We provide two results in this vein. In Appendix C.1, we show that the differences $\boldsymbol{b}^{(t)} - \boldsymbol{b}^{(\tau)}$ can be estimated approximately, and we show exactly how the resulting noise appears in our learning guarantees. For the mechanism classes we studied in the previous section, our per-task sample complexity guarantees are significantly better than the best-known single-task sample complexity bounds. In Appendix C.2, we show that in some cases, the differences $\boldsymbol{b}^{(t)} - \boldsymbol{b}^{(\tau)}$ can be estimated exactly, in which case our learning guarantees follow directly from Theorem C.1.

### C.1 Approximate estimation

For each pair of tasks $t$ and $\tau$ and bidder $i$, we define the following natural estimate of $b_i^{(t)} - b_i^{(\tau)}$:

$$\hat{b}_i^{(t,\tau)} = \frac{1}{Lm} \sum_{\ell=1}^{L} \sum_{j=1}^{m} v_{ij}^{(\ell,t)} - v_{ij}^{(\ell,\tau)},$$

where $v_{ij}^{(\ell,t)}$ is bidder $i$'s value for item $j$ under the $\ell^{th}$ sample $\boldsymbol{v}^{(\ell,t)}$ of task $t$. This estimate makes sense because in expectation, each summand equals $b_i^{(t)} - b_i^{(\tau)}$. A Hoeffding bound implies that this estimate's error is small, as we prove in the following lemma.

**Lemma C.2.** *With probability $1 - \delta$ over the draw of the $LT$ samples $\left\{\boldsymbol{v}^{(\ell,t)} : \ell \in [L], t \in [T]\right\}$, for all bidders $i \in [n]$ and pairs of tasks $t, \tau \in [T]$, $\left| \hat{b}_i^{(t,\tau)} - \left(b_i^{(t)} - b_i^{(\tau)}\right) \right| \leq \sqrt{\frac{1}{2Lm} \log \frac{2nT^2}{\delta}}.$*

*Proof.* By definition, $v_{ij}^{(\ell,t)} - v_{ij}^{(\ell,\tau)} = b_i^{(t)} + z_{ij}^{(\ell,t)} - \left(b_i^{(\tau)} + z_{ij}^{(\ell,\tau)}\right)$, where $z_{ij}^{(\ell,t)}$ and $z_{ij}^{(\ell,\tau)}$ are both i.i.d. draws from the distribution $\mathcal{D}_{ij}$. Therefore, $\mathbb{E}\left[v_{ij}^{(\ell,t)} - v_{ij}^{(\ell,\tau)}\right] = b_i^{(t)} - b_i^{(\tau)}$, so the lemma follows from a Hoeffding bound and a union bound over all $n$ bidders and $\binom{T}{2}$ pairs of tasks. $\qquad\square$

We now show how to combine this estimate with Fact 3.2 and Theorem C.1 to provide strong multi-task learning guarantees. We use the notation $\hat{\boldsymbol{b}}^{(t,\tau)} = \left(\hat{b}_1^{(t,\tau)}, \ldots, \hat{b}_n^{(t,\tau)}\right)$ and $\hat{\boldsymbol{b}}^{(t,\tau)} + \boldsymbol{v}^{(\ell,\tau)}$ to denote the vector in $\mathbb{R}^{nm}$ whose components equal $\hat{b}_i^{(t,\tau)} + v_{ij}^{(\ell,\tau)}$ for all bidders $i \in [n]$ and items $j \in [m]$. Fix an arbitrary task $t \in [T]$. Let $\mathcal{S}_t = \left\{\boldsymbol{b}^{(t)} + \boldsymbol{z}^{(\ell,\tau)} : \ell \in [L], \tau \in [T]\right\}$. This is a set of $LT$ samples drawn from the distribution $\mathcal{D}^{(t)}$. We do not have access to this set of samples, but we do have access to a noisy version $\hat{\boldsymbol{b}}^{(t,\tau)} + \boldsymbol{v}^{(\ell,\tau)}$ of each sample $\boldsymbol{b}^{(t)} + \boldsymbol{z}^{(\ell,\tau)}$. From Lemma C.2 and Equation (22), we know that with probability $1 - \delta$, for all pairs of tasks $t, \tau \in [T]$ and all $\ell \in [L]$, our estimate $\hat{\boldsymbol{b}}^{(t,\tau)} + \boldsymbol{v}^{(\ell,\tau)}$ is close to the true sample $\boldsymbol{b}^{(t)} + \boldsymbol{z}^{(\ell,\tau)}$: $\left\|\boldsymbol{b}^{(t)} + \boldsymbol{z}^{(\ell,\tau)} - \left(\hat{\boldsymbol{b}}^{(t,\tau)} + \boldsymbol{v}^{(\ell,\tau)}\right)\right\|_\infty \leq \sqrt{\frac{1}{2Lm} \log \frac{2nT^2}{\delta}}$. To apply Fact 3.2, we must underestimate the buyers' values. Therefore, we define our noisy samples as $\mathcal{S}_t' = \left\{\boldsymbol{w}^{(\ell,t,\tau)} : \ell \in [L], \tau \in [T]\right\}$ where for each bidder $i$, each item $j$, and each $\ell \in [L]$, $w_{ij}^{(\ell,t,\tau)} = \hat{b}_i^{(t,\tau)} + v_{ij}^{(\ell,\tau)} - \sqrt{\frac{1}{2Lm} \log \frac{2nT^2}{\delta}}$. Lemma C.2 and Equation (22) imply the following corollary.

**Corollary C.3.** *Let* $\epsilon = \sqrt{\frac{2}{Lm} \log \frac{2nT^2}{\delta}}$. *With probability* $1 - \delta$ *over the draw of the $LT$ samples* $\left\{\boldsymbol{v}^{(\ell,t)} : \ell \in [L], t \in [T]\right\}$, *for every bidder* $i \in [n]$, *every item* $j \in [m]$, *every pair of tasks* $t, \tau \in [T]$, *and every index* $\ell \in [L]$, $\boldsymbol{b}^{(t)} + \boldsymbol{z}^{(\ell,\tau)} - \boldsymbol{\epsilon} \leq \boldsymbol{w}^{(\ell,t,\tau)} \leq \boldsymbol{b}^{(t)} + \boldsymbol{z}^{(\ell,\tau)}$.

We now provide strong multi-task learning guarantees for lotteries and second-price auctions.

### C.1.1 Lotteries

Corollaries 3.10 and C.3 together with Theorem C.1 imply that optimizing the reserves over the set of perturbed training instances $\mathcal{S}_t'$ results in a nearly optimal lottery.

**Theorem 4.2.** *For each task $t$, let $\hat{\boldsymbol{r}}_t'$ be the empirically optimal lottery parameter vector over the set* $\mathcal{S}_t'$: $\hat{\boldsymbol{r}}_t' = \operatorname{argmax}_{\boldsymbol{r} \in \mathbb{R}^{n(m+1)}} \sum_{\ell=1}^L \sum_{\tau=1}^T \operatorname{rev}_{\boldsymbol{r}}\left(\boldsymbol{w}^{(\ell,t,\tau)}\right)$. *With probability* $1 - \delta$, *for every* $t \in [T]$,
$$\max_{\boldsymbol{r} \in \mathbb{R}^{n(m+1)}} \mathbb{E}_{\boldsymbol{v} \sim \mathcal{D}^{(t)}}\left[\operatorname{rev}_{\boldsymbol{r}}(\boldsymbol{v}) - \operatorname{rev}_{\hat{\boldsymbol{r}}_t'}(\boldsymbol{v})\right] = \tilde{O}\left(\sqrt{\frac{1}{LT}\left(nm + \log \frac{T}{\delta}\right) + \frac{n^2}{Lm} \log \frac{nT}{\delta}}\right).$$

*Proof.* This follows from Corollary 3.10 with $\epsilon = O\left(\sqrt{\frac{1}{Lm} \log \frac{nT}{\delta}}\right)$, as per Corollary C.3. $\square$

### C.1.2 Multi-item second-price auctions with non-anonymous reserves

Corollaries 3.7 and C.3 together with Theorem C.1 imply the following guarantee for second-price auctions with non-anonymous reserves.

**Theorem 4.1.** *For each task $t \in [T]$, let $\hat{\boldsymbol{r}}_t'$ be the empirically optimal reserve vector over the set* $\mathcal{S}_t'$: $\hat{\boldsymbol{r}}_t' = \operatorname{argmax}_{\boldsymbol{r} \in \mathbb{R}^{nm}} \sum_{\ell=1}^L \sum_{\tau=1}^T \operatorname{rev}_{\boldsymbol{r}}\left(\boldsymbol{w}^{(\ell,t,\tau)}\right)$. *With probability* $1 - \delta$, *for every task $t \in [T]$*,
$$\max_{\boldsymbol{r} \in \mathbb{R}^{nm}} \mathbb{E}_{\boldsymbol{v} \sim \mathcal{D}^{(t)}}\left[\operatorname{rev}_{\boldsymbol{r}}(\boldsymbol{v}) - \operatorname{rev}_{\hat{\boldsymbol{r}}_t'}(\boldsymbol{v})\right] = \tilde{O}\left(\sqrt{\frac{1}{LT}\left(nm + \log \frac{T}{\delta}\right) + \frac{m}{L} \log \frac{nT}{\delta}}\right).$$

*Proof.* This result follows from Corollary 3.7 by the same logic as Theorem 4.2. $\square$

## C.2 Exact estimation

We now show that in some settings, we can, in fact, estimate the differences $\boldsymbol{b}^{(t)} - \boldsymbol{b}^{(\tau)}$ exactly, thus reducing the error in our revenue guarantees. This exact estimation is possible when there exists a quantal $\kappa \in \mathbb{R}$ such that for every bidder $i$ and every pair of tasks $t, \tau \in [T]$, $b_i^{(t)} - b_i^{(\tau)} = s\kappa$ for some $s \in \mathbb{Z}$. In this case, we use the following estimate $\hat{b}_i^{(t,\tau)}$ for the difference $b_i^{(t)} - b_i^{(\tau)}$:

$$\hat{b}_i^{(t,\tau)} = \left[\frac{1}{Lm} \sum_{\ell=1}^L \sum_{j=1}^m v_{ij}^{(\ell,t)} - v_{ij}^{(\ell,\tau)}\right]_\kappa,$$

where the notation $[x]_\kappa$ denotes the value $x$ rounded to the nearest multiple of $\kappa$. We prove that this estimate is exact so long as the number of samples $L$ per task is sufficiently large.

**Lemma C.4.** *Suppose that $L = \Omega\left(\frac{1}{\kappa^2 m} \ln \frac{Tn}{\delta}\right)$. With probability $1 - \delta$ over the draw of the LT samples $\left\{\boldsymbol{v}^{(\ell,t)} : \ell \in [L], t \in [T]\right\}$, for every bidder $i$ and pair of tasks $t, \tau \in [T]$, $\hat{b}_i^{(t,\tau)} = b_i^{(t)} - b_i^{(\tau)}$.*

*Proof.* From Lemma C.2, we know with probability $1 - \delta$, for every bidder $i$ and pair $t, \tau \in [T]$,

$$\left| \frac{1}{Lm} \sum_{\ell=1}^{L} \sum_{j=1}^{m} v_{ij}^{(\ell,t)} - v_{ij}^{(\ell,\tau)} - \left(b_i^{(t)} - b_i^{(\tau)}\right) \right| \leq \sqrt{\frac{1}{2Lm} \ln \frac{2T^2 n}{\delta}}. \tag{23}$$

Therefore, when $L = \Omega\left(\frac{1}{\kappa^2 m} \ln \frac{Tn}{\delta}\right)$, the right-hand-side of Equation (23) is at most $\frac{\kappa}{2}$. As a result, when we round $\frac{1}{Lm} \sum_{\ell=1}^{L} \sum_{j=1}^{m} v_{ij}^{(\ell,t)} - v_{ij}^{(\ell,\tau)}$ to the nearest multiple of $\kappa$ in order to formulate the estimate $\hat{b}_i^{(t,\tau)}$, we round it to exactly $b_i^{(t)} - b_i^{(\tau)}$, so the theorem statement holds. $\qquad\square$

From Lemma C.4 and Equation (22), we know that with probability $1 - \delta$, for every pair of tasks $t, \tau \in [T]$ and every index $\ell \in [L]$, $\hat{\boldsymbol{b}}^{(t,\tau)} + \boldsymbol{v}^{(\ell,\tau)} = \boldsymbol{b}^{(t)} + \boldsymbol{z}^{(\ell,\tau)}$, which is a sample from the distribution $\mathcal{D}^{(t)}$. As in Appendix C.1, we use the notation $\boldsymbol{w}^{(\ell,t,\tau)} = \hat{\boldsymbol{b}}^{(t,\tau)} + \boldsymbol{v}^{(\ell,\tau)}$ and $\mathcal{S}_t' = \left\{\boldsymbol{w}^{(\ell,t,\tau)} : \ell \in [L], \tau \in [T]\right\}$. We now provide learning guarantees similar to those in Appendix C.1. In the case of lottery mechanisms, Corollary 3.10, Theorem C.1, and Lemma C.4 imply that the empirically optimal lottery over the bootstrapped samples $\mathcal{S}_t'$ is nearly optimal overall.

**Theorem C.5.** *Suppose that $L = \Omega\left(\frac{1}{\kappa^2 m} \ln \frac{Tn}{\delta}\right)$. For each task $t \in [T]$, let $\hat{\boldsymbol{r}}_t'$ be the empirically optimal lottery parameter vector over the set $\mathcal{S}_t'$: $\hat{\boldsymbol{r}}_t' = \operatorname{argmax}_{\boldsymbol{r} \in [0,1]^{n(m+1)}} \sum_{\ell=1}^{L} \sum_{\tau=1}^{T} \operatorname{rev}_{\boldsymbol{r}}\left(\boldsymbol{w}^{(\ell,t,\tau)}\right)$. With probability $1 - \delta$, for every task $t \in [T]$,*

$$\max_{\boldsymbol{r} \in [0,1]^{n(m+1)}} \mathop{\mathbb{E}}_{\boldsymbol{v} \sim \mathcal{D}^{(t)}} \left[\operatorname{rev}_{\boldsymbol{r}}(\boldsymbol{v})\right] - \mathop{\mathbb{E}}_{\boldsymbol{v} \sim \mathcal{D}^{(t)}} \left[\operatorname{rev}_{\hat{\boldsymbol{r}}_t'}(\boldsymbol{v})\right] = O\left(\sqrt{\frac{nm \log(nm)}{LT}} + \sqrt{\frac{1}{LT} \log \frac{T}{\delta}}\right).$$

*Proof.* This theorem follows directly from Theorem C.1 because with probability $1 - \delta$, for every index $\ell \in [L]$ and ever pair of tasks $t, \tau \in [T]$, $\boldsymbol{w}^{(\ell,t,\tau)} = \boldsymbol{b}^{(t)} + \boldsymbol{z}^{(\ell,\tau)}$. $\qquad\square$

Theorem C.5 implies that for any $\gamma \in (0,1)$, when $LT = \tilde{\Omega}\left(\frac{nm}{\gamma^2}\right)$, the expected revenue of the lottery defined by the parameter vector $\hat{\boldsymbol{r}}_t'$ is within $\gamma$ of optimal. Meanwhile, as we mentioned in Section 4, the best-known single-task sample complexity guarantee requires $L = \tilde{\Omega}\left(\frac{nm}{\gamma^2}\right)$. Therefore, using our approach, the multi-task sample complexity $LT$ is equal to the best-known single-task sample complexity.

In the case of multi-item second-price auctions with non-anonymous reserves, Theorem C.1, Lemma C.4, and Corollary 3.7 imply the following guarantee.

**Theorem C.6.** *Suppose that $L = \Omega\left(\frac{1}{\kappa^2 m} \ln \frac{Tn}{\delta}\right)$. For each task $t \in [T]$, let $\hat{\boldsymbol{r}}_t'$ be the empirically optimal reserve vector over the set $\mathcal{S}_t'$: $\hat{\boldsymbol{r}}_t' = \operatorname{argmax}_{\boldsymbol{r} \in [0,1]^{nm}} \sum_{\ell=1}^{L} \sum_{\tau=1}^{T} \operatorname{rev}_{\boldsymbol{r}}\left(\boldsymbol{w}^{(\ell,t,\tau)}\right)$. With probability $1 - \delta$, for every task $t \in [T]$,*

$$\max_{\boldsymbol{r} \in [0,1]^{nm}} \mathop{\mathbb{E}}_{\boldsymbol{v} \sim \mathcal{D}^{(t)}} \left[\operatorname{rev}_{\boldsymbol{r}}(\boldsymbol{v}) - \operatorname{rev}_{\hat{\boldsymbol{r}}_t'}(\boldsymbol{v})\right] = O\left(\sqrt{\frac{nm \log(nm)}{LT}} + \sqrt{\frac{1}{LT} \log \frac{T}{\delta}}\right).$$

*Proof.* This theorem follows by the same logic as Theorem C.5. $\qquad\square$

This theorem implies that for any $\gamma \in (0,1)$, when $LT = \tilde{\Omega}\left(\frac{nm}{\gamma^2}\right)$, the revenues of the reserve vectors $\hat{\boldsymbol{r}}_t'$ are within $\gamma$ of optimal. On the other hand, as we mentioned in Appendix C.1.2, the best-known single-task sample complexity guarantee is $L = \tilde{\Omega}\left(\frac{nm}{\gamma^2}\right)$ [46], which is equal to our multi-task sample complexity $LT$.

| Category pairs | KS test statistic | Estimated base value difference |
|---|---|---|
| (27261, 27264) | 0.06 | -17.672 |
| (27261, 27268) | 0.072 | 1.757 |
| (27261, 27269) | 0.072 | -30.594 |
| (27264, 27268) | 0.065 | 19.429 |
| (27264, 27269) | 0.052 | -12.922 |
| (27268, 27269) | 0.06 | -32.351 |

Table 2: For several category pairs $(i, j)$, we report the KS test statistic measuring the similarity between the original dataset $\widehat{S}_i$ and the shifted dataset $\widehat{S}'_j$. We also report the estimated base-value difference mean $\left( \widehat{S}_j \right) - \text{mean} \left( \widehat{S}_i \right)$.

## D Testing the common base value assumption

Since the preceding theory relies on the common value assumption, it is important verify that this assumption holds up in real world setting. To verify this, we analyze the Ebay bidding data for sports merchandise [44] and test whether the common-base-value model [9, 24] reflects real-world value distributions for similar items.

**Setup:** Each element of the Ebay sports merchandise dataset describes the price an item was sold for on the platform, which equals the maximum of the reserve and the second-highest bid. We use this value as a proxy for a bid, since the individual bids are not reported in the dataset. The items are sorted into a number of categories including: baseball, basketball, football trading cards, and so on. Let $\widehat{S}_i \subset \mathbb{R}$ denote the set of all prices an item in category $i$ was sold for in the dataset. We assume that each set $\widehat{S}_i$ is drawn i.i.d. from an unknown distribution $\mathcal{D}_i$. Intuitively, the distributions defined by similar categories should be similar.

**Procedure:** If distributions $\mathcal{D}_i$ and $\mathcal{D}_j$ fall under the common-base-value model, then the two distributions are related by an additive shift. This shift can be estimated by taking the difference of the estimated means. That is, we calculate shifted bids $\widehat{S}'_i = \left\{ s + \text{mean} \left( \widehat{S}_j \right) - \text{mean} \left( \widehat{S}_i \right) \mid s \in \widehat{S}_i \right\}$. If the common-base-value model holds, then $\widehat{S}'_i$ and $\widehat{S}_j$ should be roughly from the same distribution. To test this, we use the well-known two-sample Kolmogorov–Smirnov test (KS) to test if $\widehat{S}'_i$ and $\widehat{S}_j$ are from the same distribution. KS tests the null hypothesis that the two sets of samples are from the same distribution, returning a test statistic that is the maximum absolute distance between the empirical CDFs of the two sets of samples. If this statistic is very small, it tells us that the two sets of samples come from very similar distributions.

**Results:** We observe that the KS test statistic is small for several distributions corresponding to different categories. As we show in Table 2, we find that the empirical buyer value distribution for the cluster of categories $\{27261, 27264, 27268, 27269\}$ falls closely inline with the common-base-value model: the KS test statistic computed for any pair of those distributions lie between $0.05$ to $0.07$. These are inline with the actual eBay categories that the IDs correspond to, which are baseball and basketball merchandise. Intuitively, it does make sense that buyer distributions for different sports merchandises have similar distributions.

We also observe sizable variance in the estimated base-value difference mean $\left( \widehat{S}_j \right) - \text{mean} \left( \widehat{S}_i \right)$. These results suggest that the common-base-value model can be useful for understanding similarities between buyers' value distributions. Lastly, the average bids in each category range from $8$ to $109$ with a high standard deviation of $18$ to $60$. This high variance rules out the possibility that the distributions are very concentrated, in which case the CBV assumption would be trivially satisfied.