# OpenReview forum: "Revenue maximization via machine learning with noisy data"
_NeurIPS.cc/2021/Conference — NeurIPS 2021 Poster_

### Official Review · Reviewer_dk54 · 2021-07-15

**Rating:** 8
**Confidence:** 3

**Summary:**

The paper studies the problem of learning revenue-maximizing auction mechanisms from samples when the observed samples may be adversarially corrupted. It shows that just applying ERM will still work well in terms of performance and sample complexity. It also shows that a particular multi-task version of the auction learning problem can be reduced to the problem of learning under adversarial noise, and so ERM will also work for this problem.


**Limitations And Societal Impact:**

This is a theory-heavy paper that is unlikely to have much direct social impact, and the authors are very clear about the scope of their results.

**Main Review:**

Longer summary:

The paper considers the problem of learning an auction from samples when the samples may be corrupted by decreasing valuations by epsilon. (Not quite the same thing as $\ell_\infty$ adversarial corruption, but close). The core claim of the paper is that ordinary ERM still works well for many classes of auctions.

The general quantity of concern is difference between mean revenue of the auction learned on true data, and mean revenue of the auction learned on noisy data. Various techniques are presented for bounding this quantity with high probability. In particular, the authors show that if a class of mechanisms is p-stable (on average over true samples, the mechanisms learned on corrupted and true data have revenue bounded by p) and q-convergent (simply a normal error bound on empirical vs. mean revenue). The classes of mechanism studied in the paper are already known to be q-convergent for various q, so by showing this result and then providing a bound of p-stability for each mechanism, it's shown that ERM works well.

In one additional case (item-price mechanisms with unit-demand buyers), p-stability does not hold. However, if it is assumed that the bidder's valuation vectors are well-dispersed (which holds with high probability if the joint valuation distribution is smooth in the sense of never having a peak density larger than $\kappa$), ERM again works well.

Finally, a limited but plausible multi-task version of the auction learning problem is defined, inspired by the common-base value model of auctions. Given multiple datasets with different, unknown base values, one can compile a larger training set which has only small errors with high probability. Learning on this new dataset is equivalent to learning under noise, so the results above showing that ERM works well also work.

Originality:

The paper leans heavily on existing theoretical results and techniques for learning auctions from samples (e.g. $\kappa$-boundedness of valuation distributions, dispersion). However, it is definitely still original.

Quality:

The paper is high quality and well argued.

Clarity:

The NeurIPS page limit has produced some unavoidable fragmentation but there's nothing to be done about this.

Significance:

The problem is an important one. It's rare to have a large supply of truthful valuations to learn an auction, and even if they are available those valuations might shift over time. So understanding how to learn auctions robust to corrupted data is an important part of the overall project of learning auctions. The results don't present any new techniques to use, but rather show that naively training on the corrupted dataset should still work well, which is surprising and interesting.

Other comments:

There is a large body of research on learning auctions in the face of distributional adversaries. Two arbitrary examples:

Koçyiğit et al., Distributionally Robust Mechanism Design

Bergemann et al., Robust Monopoly Pricing

This is a different problem than what is addressed in this paper (which is particularly concerned with learning from samples, and does not present new learning algorithms, but rather shows that the naive approach is already fairly robust). But it is related, and I was surprised not to see any of this work even cited in the appendix.

Given the discussion of ICML 21 reviews, it's possible this material was cut to prevent confusion, but I think it might be worth mentioning some of it.

There is also some adversarial robustness literature briefly cited, but only work focusing on test-time evasion attacks. I don't think it's essential to the message of the paper at all, but it couldn't hurt to cite some of the work on poisoning, where the training dataset is corrupted, as this is closer to the setting studied in this paper.

Final update: I enjoyed reading this paper and my opinion remained positive after the discussion period.

**Time Spent Reviewing:**

5

---

> ### Author Response · Authors · 2021-08-10
> **Response to Reviewer dk54**
>
> Thank you for your thoughtful feedback on the paper!
> We are glad to hear that you found the problem to be important, and the paper and its results to be well argued and surprising.
>
> We will be sure to include the two references mentioned and as well as some references to data poisoning papers, which are indeed very relevant. Please let us know if you have any further questions.

---

### Official Review · Reviewer_e5fx · 2021-07-16

**Rating:** 7
**Confidence:** 1

**Summary:**

The authors provide guarantees for optimizing revenue using a learning-based framework over several common mechanism classes when an imperceptible adversarial noise is added to the training set. They then use the guarantees for multi-task mechanism design.



**Limitations And Societal Impact:**

Yes

**Main Review:**

- The authors make two original contributions - First, they study learning-based mechanism design under adversarial noise. Despite the volatility of the revenue functions, they conclude through a sensitivity analysis that there isn't a catastrophic loss in revenue when imperceptible adversarial noise is added. They also study the problem of multi-task mechanism design (which unlike existing literature doesn't assume that there is one single distribution defining the buyer's values).
- The paper is well written and easy to read.
- The problem that the authors' study and the results presented in the paper are of interest to both the fields of robust ML and mechanism design.

**Time Spent Reviewing:**

3 hours

---

> ### Author Response · Authors · 2021-08-10
> **Response to e5fx**
>
> Thank you for your feedback on the paper!
> We appreciate your recognition of the problem's significance and the relevance of the results presented to both the field of robust ML and mechanism design.
>
> Please let us know if you have any further questions.

---

### Official Review · Reviewer_7Ngd · 2021-07-16

**Rating:** 6
**Confidence:** 3

**Summary:**

This paper provides robustness analysis for several existing types of auctions when the empirical bids can be perturbed within an L1 ball. The authors proposed two conditions: p-stable and q-convergent, under which the expected revenue of the empirically
optimal mechanism over the noisy samples is close to the expected revenue of the optimal mechanism in the class. The authors further show that several existing mechanisms satisfy these conditions to various degree.

**Limitations And Societal Impact:**

The authors have addressed the limitations and potential negative societal impact of this work.

**Main Review:**

Overall I find the paper well-written and the structure is relatively clear. I agree with the authors that robust auction design under noisy empirical data is an important problem with direct practical applications, and the authors did a good job in motivating the importance of auction design with noisy data.
1. One main concern I have is about the bounded noise assumption, and I would like to see more justification for it in the auction setting. Based on my understanding, this assumption describes the scenario where each bid might be perturbed by a small epsilon distance w.r.t. the original true bids. However, as the authors also mentioned (in Cai and Daskalakis [18]), a seemingly more natural noisy data setting is to consider distributional changes. For example, a small fraction of the bids might be arbitrarily perturbed. The bounded noise assumption seems to be a bit restrictive in the sense that each bid can only be off by a small value.
2. From the results (e.g. theorem 3.6 and 3.9), the revenue loss scales linearly in n / m and epsilon, even for the two classes of mechanisms that are shown to be p-stable and q-convergent. This seems to be a bit unsatisfying especially when n / m and epsilon is large. I would be curious to see if this is the optimal guarantee.
3. Based on my understanding, some other commonly studied auctions, e.g. second price auction with anonymous price, do not satisfy the p-stable and q-convergent condition. I wonder if the "sensitivity analysis" that the authors presented can provide insights on adapting auctions to be p-stable and q-convergent.
4, As the authors focused on ERM, I would be interested in seeing how the robustness guaranteed by p-stable and q-convergent is reflected in experiments, e.g. with simulated data.

--------------------------
Considering the authors' response, I decide to change my score from 5 to 6.

**Time Spent Reviewing:**

6

---

> ### Author Response · Authors · 2021-08-10
> **Response to Reviewer 7Ngd**
>
>
> Thank you for your thoughtful feedback on the paper!
> We agree that the paper's central problem---robust auction design under noisy empirical data---is an important problem with direct practical applications, and we value the feedback that the paper is well written.
>
> ---
>
> We’re confident that we can address all the concerns raised, and we kindly ask to have our scores re-evaluated.
>
> ```
> I would like to see more justification for [the bounded noise assumption] in the auction setting.
> ```
>
> 1. As we describe in lines 29-31 and 72-73, our motivation for this paper comes from numerous papers in adversarial learning showing that ERM for deep learning can be severely affected by minute noise. We thus looked to answer: is it true that ERM for revenue maximization can also be severely affected by minute noise?
>
>     Naturally, it made the most sense to use the same bounded noise model as in the adversarial learning literature and derive results accordingly. Indeed, the bounded noise model is the canonical noise model in adversarial learning, and for its justification, we defer to the many papers that study this noise model (e.g., Goodfellow et al., ICLR'15; Madry et al., ICLR'18; Szegedy et al., ICLR'14).
>
> 2. For the auction setting specifically, we provide a few more motivations for this noise model:
>
>    a. The noise may arise when one is using *estimated* bids to learn auction parameters. For example, in our multi-task mechanism design analysis from Section 4, we estimate bids for one task using bids from other tasks (lines 336-344).
>
>    b. There is precedence for the use of bounded noise models in auction literature; one example is the work of Golrezaei et al (NeurIPS '19).
>
>    c. The noise may be due to bounded rationality on the part of the bidders, leading to small, seemingly-innocuous differences between the buyers' true values and reported values.
>
> 3. Finally, we note that our results can immediately be extended to cover unbounded, sub-Gaussian noise, a standard noise model in machine learning. In particular, let $S = \\left\\{\vec{v}^{(1)}, \dots, \vec{v}^{(L)}\\right\\} \subset R^{nm}$ be a set of $L$ valuation vectors, where $n$ is the number of buyers and $m$ is the number of items. Suppose that each element of each vector $\vec{v}^{(i)}$ is perturbed by subG$(\sigma^2)$ to obtain the noisy vector $\vec{w}^{(i)}$. Then with probability $1-\delta$, for all $i \in [L]$, $||\vec{w}^{(i)} - \vec{v}^{(i)}||_{\infty} \leq \sigma \sqrt{2 \log \frac{2Lnm}{\delta}}$. Note that the noise need not be independent. Therefore, all of our results hold with probability $1-2\delta$ over the draw of the true values $S$ and the noise for $\epsilon = 2\sigma \sqrt{2 \log \frac{2Lnm}{\delta}}$.
>
> We will add these examples to Section 1 in the revision.
>
> ```
> As the authors also mentioned (in Cai and Daskalakis [18]), a seemingly more natural noisy data setting is to consider distributional changes.
> ```
>
> 1. As we write in lines 76-79, the paper by Cai and Daskalakis (FOCS'17) requires that the distance between the true distribution over buyers' values and the noisy distribution is small according to the KS distance. In our noise model, the KS distance between the true distribution and the empirical distribution over the noisy samples may be arbitrarily large.
>
>     For example, if a constant fraction of the probability mass falls in an interval $[c,c+\epsilon]$, our adversary can perturb all bids from this interval to equal $c$, which will cause the KS distance between the two distributions to be large. Therefore, there are settings where one noise model is more applicable than the other and vice versa, so we believe that the two noise models are not comparable.
>
> 2. Regarding the restrictiveness of the assumptions, we note that the paper by Cai and Daskalakis (FOCS'17) makes a stronger assumption that the buyers' values follow a product distribution, whereas we are able to handle arbitrarily correlated distributions.
>
> 3. Lastly, since research on mechanism design via machine learning with noisy data is only in its earliest stage, there is no "canonical" noise model. Thus, we believe it would be beneficial to consider a diverse set of noise models, and as the bounded noise model is commonly used in robust ML, it is a natural candidate to study in the context of auctions.
>
> We will be sure expand on these distinctions in Section 1.2 of the revision.
>
>
> ```
> From the results (e.g. theorem 3.6 and 3.9), the revenue loss scales linearly in n / m and epsilon, even for the two classes of mechanisms that are shown to be p-stable and q-convergent. [...] I would be curious to see if this is the optimal guarantee.
> ```
>
> In fact, we do provide such lower bounds. In Proposition 3.8, we prove that the dependence on $m\epsilon$ is tight for non-anonymous second-price auctions, where $m$ is the number of items. In Proposition 3.11, we prove that the dependence on $n\epsilon$ is tight for lotteries, where $n$ is the number of buyers.
>
> ```
> Based on my understanding, some other commonly studied auctions, e.g. second price auction with anonymous price, do not satisfy the p-stable and q-convergent condition.
> ```
>
> In fact, we do show that the class of second-price auctions with anonymous reserves satisfies $p$-stability and $q$-convergence. Please see Table 1, lines 158-159, and Appendix B.1.
>
> ```
> I wonder if the "sensitivity analysis" that the authors presented can provide insights on adapting auctions to be p-stable and q-convergent.
> ```
>
> This is a great direction for future research that we will be sure to include in the conclusion.
>
> Please let us know if you have any further questions.

---

### Official Review · Reviewer_6knT · 2021-07-16

**Rating:** 6
**Confidence:** 3

**Summary:**

The authors study revenue maximization for auctions from samples corrupted by bounded (adversarial) noise. They propose a condition called stability which, together with convergence bounds, guarantees that empirical risk minimization on the sample achieves near-optimal performance on the true distribution. They demonstrate that simple auction formats such as the second-price auctions with reserves satisfy stability.

Finally, they use stability to help design auctions for multi-task settings where the distribution over values in each task is a translation of the base distribution over values (and the translation is unknown). The main insight is that the samples from other tasks can be appropriately transferred to improve the revenue approximation guaranteed for each task.



**Limitations And Societal Impact:**

The authors adequately discussed the societal impact of their work.

**Main Review:**

This paper builds on a line of recent work on revenue maximization from iid samples. The problem of revenue maximization from samples corrupted by adversarial noise is a natural extension. I found the setup for the multi-task setting and the insight that stability is helpful in this setting to be particularly interesting, and I would encourage the authors to add more detail about these results to the main body of the paper. The paper was well-written and seemed technically sound.

While the proofs that the auction formats were stable were clearly written and seemed technically sound, I did not find the fact that these auctions were stable to be surprising considering the robustness properties proven in previous work (e.g. [29] and (*) below).

It is not clear that stability holds for revenue-optimal auction formats (i.e. the Myerson auction in the single item setup). Since the goal is ultimately revenue maximization, it seems important to design auctions from corrupted samples that achieve near-optimal revenue across all possible auction formats.

Another weakness is the assumption that the adversarial noise is bounded, which seems necessary for the proposed stability property to guarantee a high revenue. However, this assumption seems very strong, and it would be interesting to design variants of stability that are suitable for more realistic noise models.

Minor comments:
-	Parenthesis on last line of Def. 3.1 would help with clarity.

(*) Huang, Liu, and Wang. Learning Optimal Reserve Price against Non-myopic Bidders, 2018.

——-
Update: Thanks to the authors for their detailed responses. Although investigating the stability of a richer class of mechanisms would provide a more comprehensive picture, the author response has convinced me that their results on multi-item auctions are sufficiently important for acceptance. I have thus decided to raised my score.


**Time Spent Reviewing:**

4

---

> ### Author Response · Authors · 2021-08-10
> **Response to Reviewer 6knT**
>
>
> Thank you for your thoughtful feedback on the paper! We appreciate your recognition that the paper is technically sound, and we are glad that you found stability's connection to multi-task mechanism design to be particularly insightful.
>
> We will be sure to further highlight the link between multi-task learning and stability in Section 1 as well as include the Huang et al reference as suggested.
>
> ---
>
> We’re confident that we can address all the concerns raised, and we kindly ask to have our scores re-evaluated.
>
> ```
> I did not find the fact that these auctions were stable to be surprising considering the robustness properties proven in previous work (e.g. [29] and Huang, Liu, and Wang. Learning Optimal Reserve Price against Non-myopic Bidders, 2018).
> ```
>
> In a nutshell, the key difference is that Huang et al. and Guo et al. [29] study mechanism design for *single-item* settings, whereas we study a wider and different set of *multi-item* mechanism classes including those not considered by Guo et al. and Huang et al.: multi-item lotteries and multi-item item-pricing mechanisms. We are not aware of any implications that the papers by Guo et al. and Huang et al. have for these multi-item mechanism classes.
>
> In fact, the past few decades of research on revenue-maximizing multi-item mechanism design has demonstrated that results and intuition from the single-item setting do not always carry over to the multi-item setting. And we see that this is the case in our paper as well.
>
> For example, an item-pricing mechanism for a single item would have a straight-forward stability analysis (similar to the analysis of a single-item anonymous second-price auction that we include for intuition in Appendix B). With multiple items and unit-demand buyers, however, stability does not even hold, as we discuss in lines 269-271. Therefore, intuition from the single-item case does not carry over to the multi-item case, and we must use completely different analysis techniques (namely, the use of *dispersion*, which is the focus of Section 3.4). We will be sure to add this discussion to Section 1.1 of the revision.
>
> ```
> It is not clear that stability holds for revenue-optimal auction formats (i.e. the Myerson auction in the single item setup). Since the goal is ultimately revenue maximization, it seems important to design auctions from corrupted samples that achieve near-optimal revenue across all possible auction formats.
> ```
>
> We emphasize that the focus of this paper is *multi-item* mechanism design, where even with full knowledge of the distribution over buyers' values, the form of the revenue-maximizing auction is still *unknown*.
> While we agree that the problem of designing multi-item auctions from corrupted samples that achieve near-optimal revenue is important, it likely cannot be resolved before we understand what the optimal, multi-item mechanisms are in the first place.
>
> Fortunately, prior research has shown that in some settings, the mechanism families we optimize over contain mechanisms that obtain a constant fraction of optimal revenue. In these settings, our approach can be used to learn a mechanism with approximately optimal revenue. For example, Chawla et al. (EC'2007, STOC'2010) prove that under unit-demand buyers, item-pricing mechanisms provide a constant fraction of optimal revenue. This setting is the focus of Section 3.4. Hart and Nisan (EC'2012, JET'2017) and Li and Yao (PNAS'2013) prove that under a single additive buyer, item-pricing mechanisms provide a $\Theta(\log m)$ fraction of optimal revenue, where $m$ is the number of items. This setting is the focus of Section 3.3 (where for a single additive buyer, a multi-item non-anonymous second-price auction is equivalent to an item-pricing mechanism). We will add these references to Section 1.2.
>
> Indeed, it would be exciting to comprehensively characterize the stability of all mechanisms. As the reviewer suggests, analyzing the stability of single-item Myerson auctions is an especially interesting direction for future research. Interestingly, we note that Guo et al.'s paper, which is referenced in he previous paragraph of the review, is on Myerson's auction and could provide insights. We will add this direction to Section 5.
>
> ```
> [The bounded noise] assumption seems very strong, and it would be interesting to design variants of stability that are suitable for more realistic noise models.
> ```
>
> 1. As we describe in lines 29-31 and 72-73, our motivation for this paper comes from numerous papers in adversarial learning showing that ERM for deep learning can be severely affected by minute noise. We thus looked to answer: is it true that ERM for revenue maximization can also be severely affected by minute noise?
>
>     Naturally, it made the most sense to use the same bounded noise model as in the adversarial learning literature and derive results accordingly. Indeed, the bounded noise model is the canonical noise model in adversarial learning, and for its justification, we defer to the many papers that study this noise model (e.g., Goodfellow et al., ICLR'15; Madry et al., ICLR'18; Szegedy et al., ICLR'14).
>
> 2. For the auction setting specifically, we provide a few more motivations for this noise model:
>
>    a. The noise may arise when one is using *estimated* bids to learn auction parameters. For example, in our multi-task mechanism design analysis from Section 4, we estimate bids for one task using bids from other tasks (lines 336-344).
>
>    b. There is precedence for the use of bounded noise models in the auction literature; one example is the work of Golrezaei et al (NeurIPS '19).
>
>    c. The noise may be due to bounded rationality on the part of the bidders, leading to small, seemingly-innocuous differences between the buyers' true values and reported values.
>
> 3. Finally, we note that our results can immediately be extended to cover unbounded, sub-Gaussian noise, a standard noise model in machine learning. In particular, let $S = \\left\\{\vec{v}^{(1)}, \dots, \vec{v}^{(L)}\\right\\} \subset R^{nm}$ be a set of $L$ valuation vectors, where $n$ is the number of buyers and $m$ is the number of items. Suppose that each element of each vector $\vec{v}^{(i)}$ is perturbed by subG$(\sigma^2)$ to obtain the noisy vector $\vec{w}^{(i)}$. Then with probability $1-\delta$, for all $i \in [L]$, $||\vec{w}^{(i)} - \vec{v}^{(i)}||_{\infty} \leq \sigma \sqrt{2 \log \frac{2Lnm}{\delta}}$. Note that the noise need not be independent. Therefore, all of our results hold with probability $1-2\delta$ over the draw of the true values $S$ and the noise for $\epsilon = 2\sigma \sqrt{2 \log \frac{2Lnm}{\delta}}$.
>
> We will be sure to add these examples to Section 1 in the revision.
>
> Please let us know if you have any further questions.

---

> ### Author Response · Authors · 2021-08-25
> **Response to Reviewer 6knT part II**
>
> Hope this note finds you well! We're just checking in since there is a week to go until the end of the rebuttal period and we were wondering if we have addressed your main concerns.
>
> To recap, we believe the concerns are: 1) Whether the robustness results are surprising based on previous literature, 2) Whether the multi-item mechanism classes that we study contain revenue-optimal mechanisms, and 3) Justification for the bounded noise model.
>
> In short, our focus is on *multi-item* mechanisms which differ from the cited literature (on single-item mechanisms)---the nuance here is that the intuition doesn't carry forward from single-item settings (one example is addressed in Section 3.4, as we discuss in our response). Furthermore, it is not yet known what the revenue-optimal mechanisms are in the multi-item case, though the mechanisms we analyze can provide approximately optimal revenue, as we describe in the response. This addresses 1) and 2). Lastly, for 3), we have provided several justifications of the noise model for your reference.
>
> Please let us know if you have any other questions and thanks for your time!

---

### Decision · Program_Chairs · 2021-09-27

**Decision:**

Accept (Poster)

**Comment:**

The paper studies the revenue-optimal mechanism design when the bidder value distribution can be corrupted by bounded adversarial noise. This is a nice contribution and a good fit to NeurIPS.